

# Toward a multivariate formulation of the PKF assimilation: application to a simplified chemical transport model

Antoine Perrot[1], Olivier Pannekoucke[1,2,3], and Vincent Guidard[1]

[1]CNRM, Université de Toulouse, Météo-France, CNRS, Toulouse, France
[2]CERFACS, Toulouse, France
[3]INPT-ENM, Toulouse, France

**Correspondence:** Olivier Pannekoucke (olivier.pannekoucke@meteo.fr)

**Abstract.** This contribution explores a new approach to forecast multivariate covariances for atmospheric chemistry through the use of the parametric Kalman filter (PKF). In the PKF formalism, the error covariance matrix is modelized by a covariance model relying on parameters, for which the dynamics is then computed. The PKF has been formulated in univariate cases, and a multivariate extension for chemical transport models is explored here. To do so, a simplified two-species chemical transport model over a 1D domain is introduced, based on the nonlinear Lotka-Volterra equations, which allows to propose a multivariate pseudo covariance model. Then, the multivariate PKF dynamics is formulated and its results are compared with a large ensemble Kalman filter (EnKF) in several numerical experiments. In these experiments, the PKF accurately reproduces the EnKF. Eventually, the PKF is formulated for a more complex chemical model composed of six chemical species (Generic Reaction Set). Again, the PKF succeeds at reproducing the multivariate covariances diagnosed on the large ensemble.

## 1 Introduction

Data assimilation aims to provide an estimation of the true state of a system. This estimation, called the analysis, is a compromise between the forecast of the state and the available observations. The optimal combination of the forecast and the observations relies on their respective error covariance matrices as given by the Kalman filter equations Kalman (1960). The accuracy of the analysis is directly related to quality of these two matrices.

In atmospheric chemistry applications, the system to study is the concentration of multiple chemical species in the atmosphere. In most cases, chemical transport models (CTMs) are used to forecast the concentrations, as the operational model MOCAGE used in Meteo-France Josse et al. (2004). CTMs make predictions based on the transport by the wind (the fields are provided by NWP models) and the chemical interactions of the species (Hauglustaine et al., 1998).

In this context, the forecast-error covariance matrix features the correlations of the forecast errors within and between the chemical species. These correlations are respectively denoted by *auto-correlations* and *cross-correlations*. Accurately de-





scribing the auto and cross correlation is a key component for improving the overall quality of the analysis. Indeed, strong correlations exist between different chemical species, and the analysis could benefit from them: an observation for a given species might also correct others concentrations and reducing their error amplitude at the same time.

However, the estimation and the modelling of multivariate covariances in air quality is a complex topic (Emili et al., 2016). But this is not specific to air quality, and two main approaches are found in data assimilation. The first one relies on balance operators and has been introduced in variationnal data assimilation. These balance operators establish a relation between the state variables and allow for the modelling of cross-covariances from the design of univariate covariances. Such operators exist in numerical weather prediction (Derber and Bouttier, 1999; Fisher, 2003) as well as for the ocean (Weaver et al., 2006),

but as far as we know, no balance operators are used in atmospheric chemistry applications. The second approach relies on the ensemble method (Evensen, 2009) where an ensemble of forecasts is used to estimate the multivariate covariance matrix (Coman et al., 2012). The ensemble method offers a flow dependent estimation of the error statistics and leads to a practical implementation of the Kalman filter, that is the ensemble Kalman filter (EnKF) (Evensen, 1994). The EnKF applies to a wide range of problems, from a simple Lorenz 63 model (Lorenz, 1963) to the numerical prediction of the atmosphere or the ocean.

On the other hand, this advantage may be seen as a limitation: the EnKF not necessarily takes advantage of the particular set of equations of a problem *e.g.* the continuity of physical fields which leads to simplification not available in the usual matrix formulation of the EnKF equations. Moreover the ensemble method presents some drawbacks. For instance, since the estimation often relies on a small ensemble, the statistical estimations are polluted by a spurious sampling noise which needs to introduce filtering (Berre et al., 2007) and localization (Houtekamer and Mitchell, 1998, 2001). In air quality, it may be

preferable to set them to zero to avoid polluting the resulting analysis state (Tang et al., 2011; Gaubert et al., 2014), except at the globe surface (Eben et al., 2005) or when the chemical species are strongly correlated (Miyazaki et al., 2012). Note that additionnal treatments can be required as inflation of the variance so to represent effects of model errors (Anderson and Anderson, 1999; Whitaker and Hamill, 2003). As another drawback, the numerical computation of the EnKF is costly since it relies on the several time integrations of a numerical model, but most of time the integrations are made at a lower spatial

resolution and in parallel.

Recently, a new approximation of the KF has been introduced, the parametric Kalman filter (PKF), where the error covariance matrices are approximated by a covariance model fitted with a set of parameters *e.g.* the grid-point variance and the local anisotropy (Pannekoucke et al., 2016). In the PKF, the dynamics of the parameters are described all along the forecast and analysis steps of the assimilation cycle (Pannekoucke, 2021). This approach does not rely on ensembles, and the dynamics of

the parameters is deduced from the partial differential equations that govern the physical system. Hence, the PKF opens the way to understanding the physics of uncertainties. However, the construction of the parameter dynamics is the most difficult part for the design of the PKF. When the parameters are the variance and the local error-correlation anisotropy, a systematic formalism for deducing PKF's equations based on a Reynold decomposition has been introduced associated with a Python package, SymPKF (Pannekoucke and Arbogast, 2021), and leveraged on the Python computer algebra system Sympy (Meurer et al.,

2017). But, modeling the physics of uncertainties often comes with closure problems. To alleviate this issue, an other numerical





framework, PDE-Netgen has been introduced to be able to close problems using a deep learning approach (Pannekoucke and Fablet, 2020).

Applying the PKF approach for CTMs is attractive because the parametric dynamics is known for the transport equations (Cohn, 1993; Pannekoucke et al., 2018), and this leads to a better understanding of the forecast-error covariance dynamics

*e.g.* a better understanding of the model-error covariance due to the numerical integration (Pannekoucke et al., 2021), and the loss of variance which appears in the EnKF (Ménard et al., 2021). While the PKF has been formulated for univariate statistics, a first attempt in multivariate statistics has been proposed, based on the balance operator approach (Pannekoucke, 2021). However, applying such a balance operator is a challenge for chemical reactions where no simple relation exists as the geostrophic balance in weather forecasting. Hence, the aim of this contribution is to explore how to extend the univariate PKF

into a multivariate formulation adapted to CTMs. To do so, a multivariate covariance model adapted to air quality prediction is first proposed and then it is validated from a twin experiment based on an EnKF using a large ensemble.

The paper is organized as follows. Section 2 reminds basic concepts in data assimilation with the formalism of the Kalman Filter and its parametric approximation in univariate statistics. Then, in Section 3, a simplified two species multivariate CTM is introduced for which a multivariate parametric assimilation is first proposed then validated based on a comparison with an

ensemble approach. A six-species chemical scheme is considered in Section 4 to evaluate the PKF multivariate forecast in a more complex context. A discussion of the results is proposed in Section 5 before to conclude in Section 6.

## 2   Background on the Parametric Kalman Filter

The parametric Kalman filter (PKF) is a recent implementation of the Kalman filter where the covariance matrices are approximated by some covariance model. For the sake of consistency, this section first recaps the basics of the Kalman filter, then

it reminds the diagnosis of covariance matrix in large dimension and covariance models so to introduce the formalism of the PKF in univariate statistics. The section ends with a numerical example of interest for air quality that illustrates the PKF.

### 2.1   Analysis and forecast step in the Kalman filter

Here we consider a system whose state is denoted by $\mathcal{X}$ and governed by the evolution equation

$$\partial_t \mathcal{X} = \mathcal{M}(\mathcal{X}). \tag{1}$$

Time integration from a time $t_q$ to a time $t_{q+1}$ of the dynamics Eq. (1) defines the propagator $\mathcal{M}_{t_{q+1} \leftarrow t_q}$, that maps a state $\mathcal{X}(t_q)$ to the prediction of Eq. (1), $\mathcal{X}(t_{q+1}) = \mathcal{M}_{t_{q+1} \leftarrow t_q} \mathcal{X}(t_q)$. In geophysics, $\mathcal{X}$ stands for the multivariate fields that represent the state of the ocean, the atmosphere or chemical species concentration for air quality. The dynamics $\mathcal{M}$ is then given by a system of partial differential equations. After spatial discretization, $\mathcal{M}$ becomes a system of ordinary differential equations, and $\mathcal{X}$ is a vector of dimension $n$.

Because of the spatio-temporal sparsity of the observations as well as the error of modelling, the exact true state at a time $t = t_q$, $\mathcal{X}_q^t$, is unknown.



Data assimilation aims to provide the analysis state, $\mathcal{X}_q^a$, that is an estimation of $\mathcal{X}_q^t$ performed from the observations and estimation of $\mathcal{X}_q^t$ coming from the past. The analysis state is decomposed into $\mathcal{X}_q^a = \mathcal{X}_q^t + \varepsilon_q^a$ where $\varepsilon_q^a$ is the analysis error, which is modeled as a random error of zero mean and covariance matrix $\mathbf{P}_q^a = \mathbb{E}\left(\varepsilon_q^a(\varepsilon_q^a)^{\mathrm{T}}\right)$, with $\mathbb{E}$ (or its shorthand $\overline{\cdot}$) the

expectation operator, and $^{\mathrm{T}}$ the transpose operator. This analysis state $X_q^a$ can be obtained by combining the forecast state $\mathcal{X}_q^f$ and the observations $\mathcal{Y}_q^{obs}$. Similarly, to the analysis state, the forecast and the observations can be written as $\mathcal{X}_q^f = X_q^t + \varepsilon_q^f$ and $\mathcal{Y}_q^{obs} = Y_q^t + \varepsilon_q^{obs}$ introducing the forecast (the observation) error $\varepsilon_q^f$ ($\varepsilon_q^{obs}$), both modelled as random errors of zero mean and covariance matrices $\mathbf{P}_q^f = \mathbb{E}\left(\varepsilon_q^f(\varepsilon_q^f)^{\mathrm{T}}\right)$ and $\mathbf{R}_q = \mathbb{E}\left(\varepsilon_q^{obs}(\varepsilon_q^{obs})^{\mathrm{T}}\right)$ respectively. In the case where the dynamic of $\mathcal{X}^t$ is assumed linear, replacing $\mathcal{M}$ by its matrix version $\mathbf{M}$ in Eq. (1); and when the errors are Gaussian and uncorrelated in time,

the Kalman filter's equations (KF) describe the evolution of the uncertainty over time (Kalman, 1960).

The process of estimating the analysis state from a forecast and some observations is called the analysis step. The forecast error covariance matrix denoted by $\mathbf{P}_q^f$ and the observation error covariance matrix $\mathbf{R}_q$ associated respectively with $\mathcal{X}_q^f$ and $\mathcal{Y}_q^{obs}$, are used to produce the optimal estimation (*analysis*) $\mathcal{X}_q^a$ of $\mathcal{X}_q^t$, and the associated analysis-error covariance matrix $\mathbf{P}_q^a$. The equations of this procedure are:

$$\mathcal{X}_q^a = \mathcal{X}_q^f + \mathbf{K}_q\left(\mathcal{Y}_q^{obs} - \mathbf{H}_q\mathcal{X}_q^f\right), \tag{2a}$$

$$\mathbf{P}_q^a = (\mathbf{I}_n - \mathbf{K}_q\mathbf{H}_q)\,\mathbf{P}_q^f, \tag{2b}$$

where $\mathbf{K}_q = \mathbf{P}_q^f\mathbf{H}_q^{\mathrm{T}}\left(\mathbf{H}_q\mathbf{P}_q^f\mathbf{H}_q^{\mathrm{T}} + \mathbf{R}_q\right)^{-1}$ is the Kalman gain matrix with $\mathbf{H}_q$ the linear observation operator that maps the state vector into the observation space; $\mathbf{P}_q^a$ is the analysis error covariance matrix ; and $\mathbf{I}_n$ the identity matrix in dimension $n$.

Next, the forecast step pushes the uncertainty forward in time. The analysis state $X_q^a$ is propagated using the linear dynamics

$\mathbf{M}$, so to obtain the forecast $\mathcal{X}_{q+1}^f$ at time $t_{q+1}$ leading to an estimation of the true state system $\mathcal{X}^t(t_{q+1})$. The Gaussian error statistics for this forecast are given by the Kalman filter forecast step

$$\mathcal{X}_{q+1}^f = \mathbf{M}_{q+1 \leftarrow q}\mathcal{X}_q^a,, \tag{3a}$$

$$\mathbf{P}_{q+1}^f = \mathbf{M}_{q+1 \leftarrow q}\mathbf{P}_q^a\left(\mathbf{M}_{q+1 \leftarrow q}\right)^{\mathrm{T}} + \mathbf{Q}_q, \tag{3b}$$

where $\mathbf{Q}_q$ is the model error covariance matrix. Thereafter, no model error is considered *i.e.* $\mathbf{Q}$ is zero.

While the Kalman filter formalism is based on simple vector algebra equations, it is not easy to understand the statistical content of the error covariances, which would require representing each covariance function and exploring their temporal evolution. Fortunately, simple diagnosis can be introduced to summarize the statistical relationship between points in the geographic domain. In turn, these diagnostics can be used as parameters of covariance models, as detailed now.

## 2.2 Diagnosis and modelling of covariance matrix in large dimension

In data assimilation, two diagnosis for the error covariance matrices are often introduced: the variance field, and the anisotropy of the correlation functions which corresponds to the principal axes of the spatial correlation. These diagnosis are recalled here for the forecast-error covariance matrix.





The forecast error variance field, $V^f$, is defined by $V^f(\mathbf{x}) = \mathbb{E}\left((\varepsilon^f(\mathbf{x}))^2\right)$ where $\mathbf{x}$ denotes the coordinate of a grid point. The variance field also corresponds to the diagonal of $\mathbf{P}^f$. The field of variance characterizes the magnitude of the error at a given position.

When the forecast-error is a differential random field, the anisotropy of the correlation is characterized by the so-called local forecast-error metric tensor $\mathbf{g}^f(\mathbf{x})$, that appears in the Taylor expansion of the correlation function (Daley, 1991)

$$\rho^f(\mathbf{x}, \mathbf{x} + \delta\mathbf{x}) \approx 1 - \frac{1}{2}||\delta\mathbf{x}||^2_{\mathbf{g}^f(\mathbf{x})}. \tag{4}$$

The local metric tensor $\mathbf{g}^f(\mathbf{x})$ is a symmetric positive-definite matrix that prevents the correlation value from being larger than one. The metric tensor is related to the statistics of the random field $\varepsilon^f$ according to the formula (Berre et al., 2007):

$$\mathbf{g}^f_{ij}(\mathbf{x}) = \mathbb{E}\left[\partial_{\mathbf{x}^i}\left(\frac{\varepsilon^f}{\sigma^f}\right)\partial_{\mathbf{x}^j}\left(\frac{\varepsilon^f}{\sigma^f}\right)\right](\mathbf{x}), \tag{5}$$

where $\sigma^f = \sqrt{V^f}$ is the forecast-error standard deviation.

In practice, the direction of the largest correlation anisotropy corresponds to the principal axe of the smallest eigenvalue for the metric tensor: the metric tensor is *contravariant*. It it thus useful to introduce the local aspect tensor (Purser et al., 2003) whose geometry goes as the correlation, and is defined as the inverse of the metric tensor:

$$\mathbf{s}^f(\mathbf{x}) = \left(\mathbf{g}^f(\mathbf{x})\right)^{-1}, \tag{6}$$

where the superscript $^{-1}$ denotes the matrix inverse.

One of the motivations behind the diagnosis of the variance and the local anisotropy tensor is that they can be used as parameters of covariance models, the VLATcov models (Pannekoucke, 2021). For instance, the anisotropy tensor has been used as a proxy for setting the heterogeneous diffusion tensor field of the covariance model based on a heterogeneous diffusion equation (Pannekoucke and Massart, 2008; Mirouze and Weaver, 2010). The covariance model based on a heterogeneous diffusion equation is an example of covariance model used in variational data assimilation and introduced to build heterogeneous covariance model, that is a covariance model for which the correlation functions vary from one geographical point to another. While there is no analytical expression for the covariance functions based on the diffusion operator, analytical heterogeneous VLATcov models exist, for instance the heterogeneous Gaussian-like covariance model

$$\mathbf{P}^{\mathrm{he\cdot gauss}}(V, \mathbf{s})(\mathbf{x}, \mathbf{y}) = \sqrt{V_{\mathbf{x}}V_{\mathbf{y}}}\frac{|\mathbf{s}_{\mathbf{x}}|^{1/4}|\mathbf{s}_{\mathbf{y}}|^{1/4}}{|\frac{1}{2}(\mathbf{s}_{\mathbf{x}} + \mathbf{s}_{\mathbf{y}})|^{1/2}}\exp\left(-\frac{1}{2}||\mathbf{x} - \mathbf{y}||^2_{[\frac{1}{2}(\mathbf{s}_{\mathbf{x}} + \mathbf{s}_{\mathbf{y}})]^{-1}}\right), \quad (7)$$

with $|\cdot|$ denoting the matrix determinant (Paciorek and Schervish, 2006).

Heterogeneous covariance models are important because they provide a way to produce non obvious correlation functions from a set a parameters. Hence, approximating a covariance matrix, as the forecast-error covariance at a given time, by a covariance model leads to sum up the statistical content into a set of parameters. The parameteric Kalman filter takes advantage of this kind of approximation so to reproduce the Kalman filter dynamics as now explained.





### 2.3 Formalism of the parametric Kalman filter

150 A covariance model is first considered, $\mathbf{P}(\mathcal{P})$, where $\mathcal{P}$ denotes a set of parameters. For instance, when the PKF is designed from a VLATcov models, the set of parameters $\mathcal{P}$ is given by the field of variance and of the local anisotropic tensors *i.e.* $\mathcal{P} = (V, \mathbf{s})$ or $\mathcal{P} = (V, \mathbf{g})$.

To describe the sequential evolution of error covariance matrices along the assimilation cycles we assume that the forecast error-covariance matrix at a time $t_q$, $\mathbf{P}_q^f$, is approximated by the covariance model, $\mathbf{P}(\mathcal{P}_q^f)$, where $\mathcal{P}_q^f$ denotes a set of 155 parameters so that $\mathbf{P}(\mathcal{P}_q^f) \approx \mathbf{P}_q^f$.

At an abstract level, the parametric Kalman filter consists of the following sequential steps (Pannekoucke, 2021). The PKF analysis step, equivalent to Eq. (2), consists to determine the analysis state $\mathcal{X}_q^a$ and the parameters $\mathcal{P}_q^a$ from $\mathcal{X}_q^f$, $\mathcal{P}_q^f$ and the observations. The sketch of this step consists in a sequential processing of observations, similar to the one often encountered in EnKF (Houtekamer and Mitchell, 2001), that is a sequential assimilation of single observations based on Eq. (2a) for the mean 160 accompanied with an update of the covariance parameters so that, at the end of the analysis step, $\mathbf{P}(\mathcal{P}_q^a)$ approximates the analysis error covariance of the Kalman filter Eq. (2b) *i.e.* $\mathbf{P}(\mathcal{P}_q^a) \approx \mathbf{P}_q^a$. Note that this sequential assimilation of observations can be performed in parallel as for the EnKF, with the difference that the EnKF often assimilates a batch of observations in place of a single observation. Of course, for the PKF this step only relies on the update of the parameters, with no ensemble. For instance, when considering a VLATcov model $\mathbf{P}(V, \mathbf{g})$, the PKF analysis of a single observation at position $\mathbf{x}_l$, of value $y^o$ 165 and observation-error variance $V^o$, writes (at time $t_q$)

$$\mathcal{X}^a(\mathbf{x}) = \mathcal{X}^f(\mathbf{x}) + \sigma^f(\mathbf{x})\rho_{\mathbf{x}_l}^f(\mathbf{x})\frac{\sigma^f(\mathbf{x}_l)}{V^f(\mathbf{x}_l) + V^o}(y^o - \mathcal{X}^f(\mathbf{x}_l)), \tag{8a}$$

$$V^a(\mathbf{x}) = V^f(\mathbf{x})\left(1 - [\rho_{\mathbf{x}_l}^f(\mathbf{x})]^2\frac{V^f(\mathbf{x}_l)}{V^f(\mathbf{x}_l) + V^o}\right), \tag{8b}$$

$$\mathbf{g}^a(\mathbf{x}) \approx \frac{V^f(\mathbf{x})}{V^a(\mathbf{x})}\mathbf{g}^f(\mathbf{x}), \tag{8c}$$

where the function $\rho_{\mathbf{x}_l}^f(\mathbf{x}) = \rho(\mathbf{g}^f)(\mathbf{x}_l, \mathbf{x})$ is the correlation function associated with the covariance matrix $\mathbf{P}(V^f, \mathbf{g}^f)$ ; $\sigma^f = 170 \sqrt{V^f}$ is the field of forecast-error standard deviation ; and where Eq. (8c) is the leading order approximation of the anisotropy update.

Then, the forecast step of the PKF, equivalent to Eq. (3), consists of finding the dynamics of the parameters so to predict $\mathcal{P}_{q+1}^f$ from $\mathcal{P}_q^a$, so that $\mathbf{P}(\mathcal{P}_{q+1}^f)$ approximates the forecast-error covariance matrix of the Kalman filter *i.e.* $\mathbf{P}(\mathcal{P}_{q+1}^f) \approx \mathbf{P}_{q+1}^f$. The equation for the mean is the Eq. (3a) of the KF.

175 To put some flesh on the bone, an illustration of the PKF is now proposed for an univariate advection problem, with a focus on the forecast step. This introduction of an intermediate problem aims to give the reader a good understanding of the PKF, its advantages and difficulties, which will be necessary to address the more complex problem encountered in multivariate CTM.



## 2.4 Advection of a passive tracer with the PKF

For a one-dimensional (1D) and periodic domain, of coordinate $x$, the conservative advection of a tracer, $\mathcal{X}(t, x)$, by a stationary
heterogeneous wind field $u(x)$, can be described by the partial differential dynamics

$$\partial_t \mathcal{X} + \partial_x (u\mathcal{X}) = 0, \tag{9a}$$

or equivalently by

$$\partial_t \mathcal{X} + u\partial_x \mathcal{X} = -\mathcal{X}\partial_x u. \tag{9b}$$

The forecast step of the PKF is illustrated for the conservative dynamics where the covariance matrices are approximated by
a VLATcov model. In what follows, the PKF dynamics for the variance and the anisotropy is first presented. Then, a numerical
test-bed shows the ability of the PKF to predict the uncertainty dynamics, the latter being estimated from an ensemble method
introduced to provide a reference. This example ends by highlighting some of the limitations of the numerical validation of the
PKF from an ensemble method in presence of model error.

### 2.4.1 Formulation of the forecast step of PKF

In this 1D univariate context based on VLATcov model, the PKF dynamics for the forecast step is composed of three equations:
one for the mean state $\overline{\mathcal{X}}$, and two for the parameters of the VLATcov model, that is the variance field $V(t, x)$ and the anisotropy
field $s(t, x)$. Note that in 1D domain, the anisotropy is a scalar.

To obtain the dynamics of PKF's parameters, we proceed using a Reynold's decomposition. A Reynold's decomposition
consists in rewriting a random field $\mathcal{X}$ as a mean field plus a perturbation, that is $\mathcal{X} = \overline{\mathcal{X}} + \varepsilon$ with $\overline{\varepsilon} = 0$. Then, by using the
definition of the variance field $V_x = \overline{\varepsilon_x^2}$, and plugging it into the problem equation Eq. (9b), one can obtain its dynamics. An
equivalent process leads to the dynamics of the metric tensor $g_x$ and of the aspect tensor $s_x$, but its hand computation requires
long expressions that can be difficult to handle. To facilitate the computation of the VLATcov PKF dynamics, a computer
algebra tool, the Python package SymPKF (Pannekoucke and Arbogast, 2021), has been specifically design to derive the PKF
system dynamics. Note that a splitting strategy can be introduced so to simplify the computation of the full PKF dynamics
(Pannekoucke and Arbogast, 2021). For nonlinear dynamics, SymPKF compute the PKF dynamics from the tangent-linear
evolution.

Leveraging on SymPKF, the PKF system for the advection Eq. (9b) reads as

$$\partial_t \mathcal{X} + u\partial_x \mathcal{X} = -\mathcal{X}\partial_x u, \tag{10a}$$

$$\partial_t V + u\partial_x V = -2V\partial_x u, \tag{10b}$$

$$\partial_t s + u\partial_x s = 2s\partial_x u. \tag{10c}$$

where the overline of the mean state $\overline{\mathcal{X}}$ has been dropped for the sake of simplicity. Note that the PKF system Eq. (10),
which is decoupled, corresponds to the true uncertainty dynamics for the advection problem (Cohn, 1993; Pannekoucke et al.,



2016, 2018). This is not true in general where closure issue can appear *e.g.* for a diffusion equation, because of the second-order derivative, an unknown term appears in the dynamics of the metric and has to be closed (Pannekoucke et al., 2018).

A numerical experiment is now conducted to evaluate the PKF ability to forecast the error statistics.

### 2.4.2    Numerical validation of the PKF

The numerical experiment studies of the time propagation of an uncertainty at time $t = 0$, featured by a mean state $\mathcal{X}^0$ and an error covariance $\mathbf{P}^0$, to an arbitrary time $T$. Here, the initial error covariance is defined as the covariance $\mathbf{P}^0 = \mathbf{P}(V^0, s^0)$, where $\mathbf{P}(V, s)$ is the VLATcov model based on the heterogeneous Gaussian like model Eq. (7), for $(V^0, s^0)$ given.

To assess the PKF ability to forecast the error statistics, we compare its results with diagnoses obtained from the forecast of a large ensemble, $\{\mathcal{X}_k^f\}_{1 \le k \le N_e}$, of size $N_e = 6400$, which implies a relative error of $1.25\%$, according to the central limit theorem. At $t = 0$, the ensemble is populated for each $k$ as $\mathcal{X}_k^f(0) = \mathcal{X}^0 + \mathbf{P}_0^{1/2} \zeta_k$, where $\mathbf{P}_0^{1/2}$ is the square-root of the initial covariance matrix $\mathbf{P}_0$, and $\zeta_k$ a Gaussian sample with zero mean and covariance matrix $\mathbf{I}_n$ where $n$ is the dimension of the vector $\mathcal{X}$ *i.e.* $\zeta_k \sim \mathcal{N}(0, \mathbf{I}_n)$. Then, each member $\mathcal{X}_k^f$ is computed from the time integration of Eq. (9b) starting from $\mathcal{X}_k^f(0)$.

Note that, for the linear dynamics Eq. (9a), the full computation of the KF covariance prediction could have been considered, but the ensemble approximation has been preferred since it introduces the methodology adapted to the nonlinear setting explored for the multivariate situation in Section 3.

Hence, from the ensemble, the variance at a given time is then estimated from its unbiased estimator

$$\widehat{V^f}(x) = \frac{1}{N_e - 1} \sum_{k=1}^{N_e} \varepsilon_k^2, \tag{11}$$

with $\varepsilon_k = \mathcal{X}_k^f(x) - \widehat{\mathcal{X}^f}(x)$ and where $\widehat{\mathcal{X}^f} = \frac{1}{N_e} \sum_{k=1}^{N_e} \mathcal{X}_k^f$ is the empirical mean. The metric tensor, defined from Eq. (5), is estimated by

$$\widehat{g^f}(x) = \frac{1}{N_e} \sum_{k=1}^{N_e} (\partial_x \widetilde{\varepsilon}_k^f(x))^2, \tag{12}$$

where $\widetilde{\varepsilon}_k^f = \frac{1}{\sqrt{\widehat{V^f}}} (\mathcal{X}_k^f - \widehat{\mathcal{X}^f})$ is the normalized error.

The numerical framework used to forecast both the ensemble and the PKF system is now described. The periodic domain
is $[0, D)$ with $D = 1000$km. It is regularly discretized with $N_x = 241$ grid points, which corresponds to a meshsize $\Delta x$ of size $4.15$km. The dynamics Eq. (9b) and Eq. (10) are discretized with a finite difference method, where spatial derivatives are approximated using a centred scheme of order 2. The time integration is done using a fourth-order Runge-Kutta (RK4) scheme of time step $\Delta t$ verifying the Courant-Friedrichs-Lewy condition (CFL) (Weisstein, 2002) $\Delta t = \Delta x / U_{max}$, where $U_{max}$ is the maximum wind speed magnitude of $u$.

For this experiment, the mean state $\mathcal{X}$, the variance field $V$ and the aspect-tensor field $s$ are initialized homogeneously with values $\mathcal{X}^0 = 1$, $V^0 = (\sigma^0)^2$ where $\sigma^0 = 0.1$, and $s^0 = (l_h^0)^2$ where $l_h^0 = 15\Delta x \simeq 62.2$km. This initial setting also corresponds to the initial state of the PKF dynamics Eq. (10). In regards of the domain chosen, this setting for the length-scale is in agreement



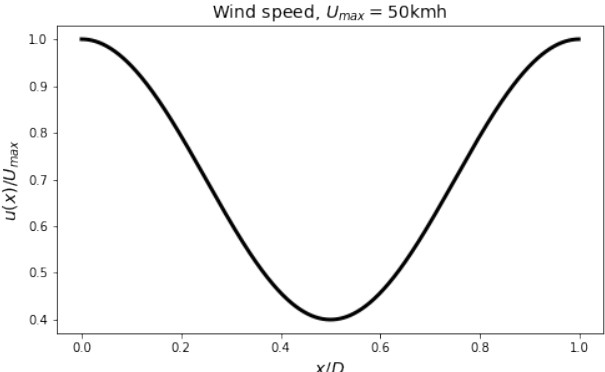

**Figure 1.** Wind field $u$.

with practical estimations often encountered (Ménard et al., 2016). The wind field considered, shown in Fig. 1, is defined by $u(x) = (35 + 15 \cos(2\pi x))/D$, and modelizes a wind of average intensity $35 \mathrm{kmh}^{-1}$ and of max speed $U_{max} = 50 \mathrm{km h}^{-1}$. The characteristic time $\tau_{adv}$ is defined by $\tau_{adv} = 1/\overline{u} \simeq 28.5\mathrm{h}$, and approximately corresponds to the time of a revolution of the tracer around the periodic domain. The simulation time horizon $T = t_{end}$ is set to $t_{end} = 3\tau_{adv}$.

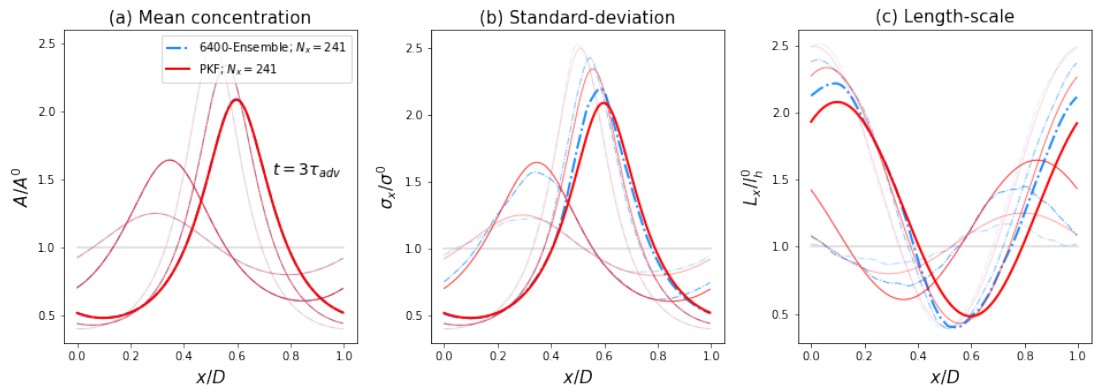

**Figure 2.** Comparison of the low resolution forecasts ($N_x = 241$) at times $t = [0.6, 1.2, 1.8, 2.4, 3.0]\tau_{adv}$ from the PKF (red lines) with the diagnoses on the ensemble forecast (cyan dash-dotted lines). The more transparent the curve, the closer it is to $t = 0$. The horizontal grey lines represent the initial conditions.

The general behaviour of the error statistics regardless of the method employed is first addressed, then the performances of the PKF and EnKF are compared.

The experiment shows that the tracer tends to concentrate in the deceleration zones (see Fig. 1 from $x = 0$ to $x = 0.5$), and to dilute in the acceleration zones (from $x = 0.5$ to $x = 1.0$). This observation also applies to the standard-deviation field (panel b), as it is governed by the same dynamic as the tracer's concentration (it is straightforward to calculate the dynamics of $\sigma$ using





the dynamics of the variance Eq. (10b)). On panel (c), the length-scales (1D equivalent of the anisotropy) are subject to two processes: a pure transport term, (l.h.s. of Eq. (10c)), and a production term related to the wind sheer (r.h.s. of Eq. (10c)). This production term is positive (negative) when the wind field is accelerating (decelerating), indicating an increase (decrease) of

the length-scales in the accelerating (decelerating) wind regions. In contrary to the concentrations and standard-deviation fields (governed by a conservative transport), the average value of the length-scales varies in time, however numerical experiments (not shown here) have shown that it oscillates around the initial value.

   Regarding the performances of the two methods, the PKF forecast results for the error statistics are quite similar to the one diagnosed from the ensemble *i.e.* the EnKF for this test-bed. The forecasts of the concentrations (panel a) are identical for

both methods. Although the dynamics for the variance Eq. (10b) and the anisotropy Eq. (10c) are exact in the PKF system, a significant difference is observed between the forecasts of the two methods (panels b and c). We justify this gap in the next paragraph 2.4.3. This numerical experiment shows that the PKF is able to produce high quality forecasts of the diagnoses of the forecast-error statistics, a result that is confirmed by looking at the forecast-error correlation functions.

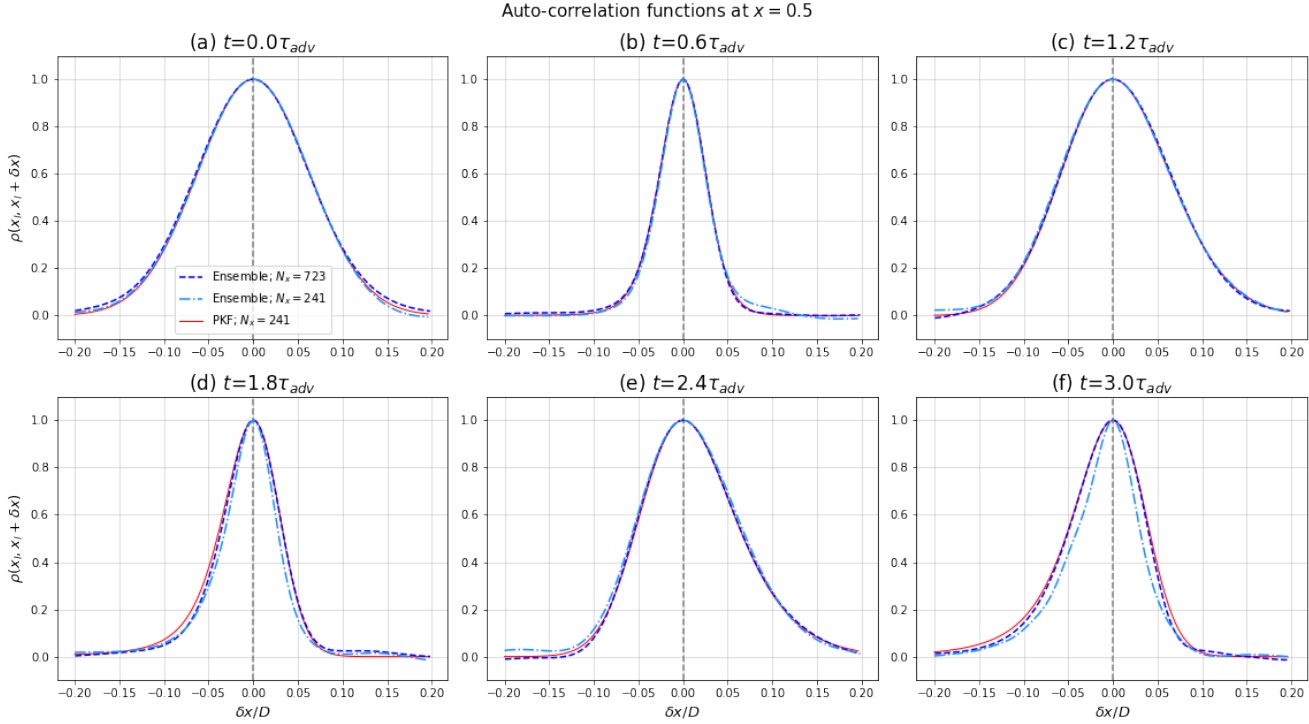

**Figure 3.** Correlation functions at location $x_l = 0.5$ and times $t = [0.0, 0.6, 1.2, 1.8, 2.4, 3.0]\tau_{adv}$, computed with PKF correlation model fitted with low resolution ($N_x = 241$) PKF forecast for error statistics (red lines) and diagnosed on the low-resolution ($N_x = 241$) ensemble (cyan dash-dotted lines) and high resolution ($N_x = 723$) ensemble (blue dashed lines).





Figure 3 compares the correlation functions at position $x_l = 0.5$, estimated from the ensemble for the EnKF (dash-dotted
cyan lines) and modeled from the predicted parameters for the PKF (solid red lines) when using Eq. (7), at different times. At
a qualitative level, the PKF is able to approximate the correlation functions, the latter being only known to within a sampling
noise because of the ensemble estimation which is assumed low due to the ensemble size. In particular, the PKF is able to
reproduce the large (the small) spread of the symmetric correlations present in panel (a) (panel b). But the PKF is also able
to represent the anisotropy of the correlations as the one shown *e.g.* in panel (e) where the correlation function at that time
appears broader on its right part (corresponding to $x$ larger than $x_l$) than on its left part (corresponding to $x$ smaller than $x_l$).

This example shows the motivation behind the PKF: it is able to predict the covariance error with a good skill and at a low
numerical cost. This low numerical cost first concerns the computer memory: the information contained in a covariance matrix
of size $\mathcal{O}(N_x^2)$ in the ensemble case, is resumed by the covariance model Eq. (7) which only needs a few parameters of size of
order $\mathcal{O}(N_x)$ (with $\mathcal{O}$ being the Big O notation, meaning "proportional to"). But the low numerical cost concerns also the time
consumed to predict the uncertainty: the PKF only relies on the single time integration of Eq. (10), that represents the cost of 3
time integrations of the initial dynamics Eq. (9b), compared to the 6400 time integrations required for the ensemble used here.

As another advantage, the PKF provides informations about the physics of the uncertainty: when ensemble diagnosis only
observes the time evolution of the statistics without any explications, the PKF provides a simplified proxy that details the origin
of these statistical evolutions with only three equations and by thus the PKF improves our knowledge of uncertainty dynamics.

Next, we would like to warn the reader about the exploration of the uncertainty dynamics from numerical experiments, as
made here to validate the PKF from an ensemble method, that faces some limits.

### 2.4.3 Limits of the numerical validation of the PKF in presence of model error

Figure 2 has shown a gap between the PKF and EnKF regarding the forecast of the error statistics (standard deviation and
length-scales, panels b and c). We now justify this observation, relating it to a model error.

As the problem is discretized for numerical simulations, the actual equation that is simulated is not exactly Eq. (9a), but
rather an implicit modified equation induced by the use of finite differences for the spatial and the temporal discretisation.
Focusing on the spatial discretization, the modified equation writes

$$\partial_t \mathcal{X} = -u\partial_x \mathcal{X} - \mathcal{X}\partial_x u - \frac{\Delta x^2}{6}u\partial_x^3 \mathcal{X} - \frac{\Delta x^2}{6}\mathcal{X}\partial_x^3 u + \mathcal{O}(\Delta x^3), \qquad (13)$$

which shows additional dispersive terms not present in the initial dynamics (Eq. 9a). Note that Eq. (13) is not the full modified
equation of the discretized model, in particular it does not represent the effect of the RK4 time scheme, but the error associated
to fourth-order time scheme should be negligible compared with the spatial numerical error (second-order). Hence, Eq. (13)
should be close to the true modified equation, and the presence of additional processes may explain the significant differences
observed in Fig. 2-(b) and (c): the dispersive term $-\frac{\Delta x^2}{6}u\partial_x^3 \mathcal{X}$ contributes to reduce the speed of the transport to a value lower
than $u$, while the term $-\frac{\Delta x^2}{6}\mathcal{X}\partial_x^3 u$ implies a local exponential growing (damping) of $\mathcal{X}(t,x)$ where $\partial_x^3 u$ is negative (positive).
This exponential evolution only contributes to the magnitude of the forecast-error *i.e.* it modifies the variance field but it has
no influence on the length-scale (Pannekoucke et al., 2018). At the opposite, the dispersive term influences both the variance




and the length-scale as it can be observed in Fig. 2-(c): the EnKF curves appear slightly late behind the PKF ones (the wind transports the curves toward the right), presenting a negative shift in the amplitude.

Since the magnitude of the dispersive term scale as $\mathcal{O}(\Delta x^2)$, a simulation at high resolution could damp this term and would
lead to attribute the gap observed in Fig. 2 to the model error. Note that only the error statistics are significantly affected by the numerical model error, the estimation for the means coinciding for the two methods on Fig. 2-(a). Therefore, as with the PKF the numerical forecast of any error statistic is treated equivalently as a state vector forecast, that is a direct time-integration, this points out the sensitivity of the EnKF to numerical model error.

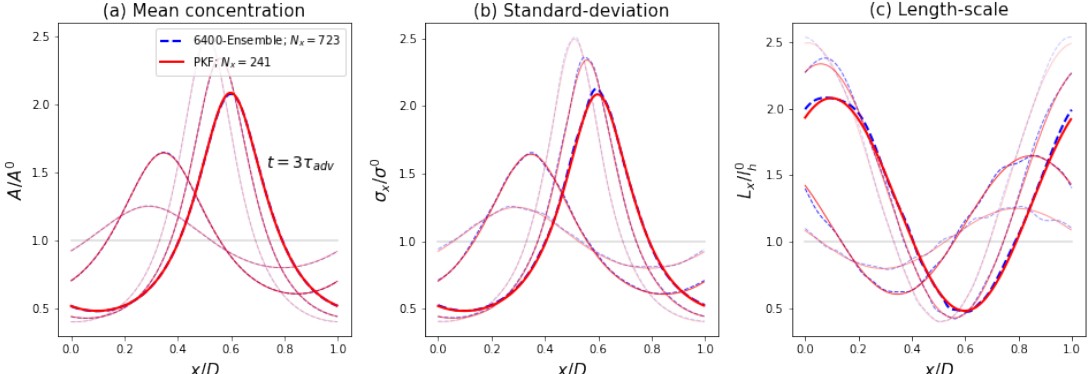

**Figure 4.** Same experiment as Fig. 2, except the EnKF forecast has been simulated using a higher grid definition ($N_x = 723$) to reduce numerical model error.

Thus, a high resolution forecast of the EnKF is now performed, with a grid of three times the original resolution *i.e.* $N_x = 3 \times$
$241 = 723$ grid points. To be consistent with the initial low resolution experiment, the initial length-scale of the high resolution is set to $l_h^0 = 3 \times 15\Delta x = 45\Delta x \simeq 62.2$km. The time step has been adapted in consequence to match the CFL condition. The results of this new simulation, in Fig. 4, show that predicting the ensemble at high resolution leads to the same variance and length-scale fields as the ones predicted by the PKF, while the latter is computed at low resolution. This demonstrates the quality of the forecasted error statistics for the PKF, even at a low resolution. Figure 3 also shows the correlation functions (blue
dashed lines) computed from the high resolution EnKF forecast. The correlation functions represented are in better accordance with the PKF modelled correlation functions than for the low resolution ensemble forecast (cyan dash-dottes lines), see *e.g.* panels (d) to (f). This shows that the PKF is little subject to numerical model error as the error statistics forecasts directly results from their time-integration.

Hence, this numerical exploration of the PKF, applied to the conservative dynamics, has shown the ability of the PKF to
provide an accurate prediction of univariate error statistics. The exploration of the multivariate extension is now addressed.



## 3 Toward a multivariate formulation of the PKF

For multivariate problems, a modelization of the cross-correlation functions (or inter-species correlation functions) is needed. Moreover, it would be convenient to introduce a multivariate covariance model that extends the univariate VLATcov model, as the heterogeneous Gaussian model (Eq. 7), so to leverage on the PKF dynamics of univariate statistics.

Because multivariate modelling is a difficult topic, a simplified test-bed dynamics is first introduced in Section 3.1. Then, to determine a multivariate covariance model and its parameters, a data-driven modeling is considered in Section 3.2. Next the mutivariate PKF is formulated, detailing the prediction step in Section 3.3 and the analysis step in Section 3.4. Finally, two numerical assimilation experiments are conducted in Section 3.5.

### 3.1 Introduction of the simplified chemical transport model

To explore a multivariate formulation of the PKF, a simplified chemical transport model is introduced. This simplified CTM contains the essential features of what can be found in a more realistic CTM, that is advection, multiple chemical species and non-linearities.

To do so, a 1D periodic domain of coordinate $x$ is considered, where two non-linearly reactive chemical species, $A(t,x)$ and $B(t,x)$, are advected in a conservative way by a heterogeneous and stationary wind field $u(x)$. The non-linear reaction is given

by the Lotka-Volterra (LV) equations (see Appendix A), which leads to the coupled dynamics

$$\partial_t A + u\partial_x A = -A\partial_x u + k_1 A - k_2 AB, \tag{14a}$$

$$\partial_t B + u\partial_x B = -B\partial_x u + k_2 AB - k_3 B. \tag{14b}$$

where the transport is written following the univariate 1D example Eq. (9b), and where the LV reaction appears as the last two terms in the right hand side of each prognostic equations. The constants $k_1$, $k_2$ and $k_3$ characterize the reaction rates: $k_1$

corresponds to the rate at which $A$ is produced; constant $k_2$ represents the rate at which the chemical reactions between $A$ and $B$ produces $2B$; and $k_3$ describes the decay rate for specie $B$. Note that at a formal level, the state vector associated with Eq. (14) is then $\mathcal{X}(t,x) = (A,B)(t,x)$.

Considered as a dynamical system of ordinary equations and represented in the phase space $(A,B)$, the solutions of the Lotka-Volterra's dynamics are periodical orbits flowing around the critical point of coordinates $(A_c, B_c) = \left(\frac{k_3}{k_1}, \frac{k_1}{k_2}\right)$, as shown

in Fig. 5. This is the kind of time evolution observed at each grid point when there is no wind ($u = 0$).

Thanks to this simplified multivariate framework, a proxy for the cross-covariance is now proposed.

### 3.2 Development of a proxy multivariate covariance model

In this multivariate framework, the error-covariance matrix $\mathbf{P} = \mathbb{E}\left(\varepsilon_{\mathcal{X}}(\varepsilon_{\mathcal{X}})^{\mathrm{T}}\right)$ associated with the state $\mathcal{X} = (A,B)$, of error $\varepsilon_{\mathcal{X}} = (\varepsilon_A, \varepsilon_B)$, reads as a block matrix

$$\mathbf{P} = \begin{pmatrix} \mathbf{P}_A & (\mathbf{P}_{AB})^{\mathrm{T}} \\ \mathbf{P}_{AB} & \mathbf{P}_B \end{pmatrix}, \tag{15}$$



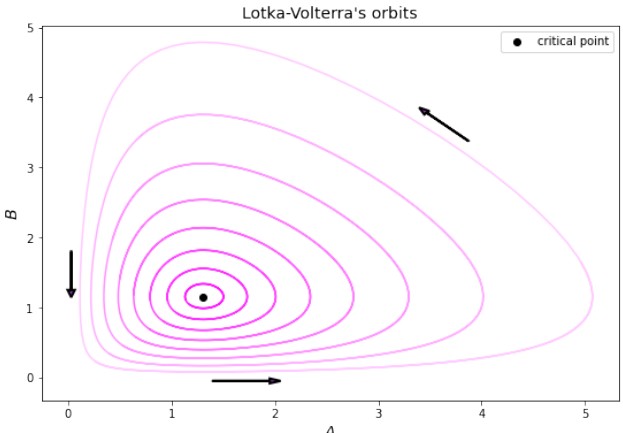

**Figure 5.** Numerical simulations of the Lotka-Volterra dynamical system whose solutions are periodical orbits (purple curves, with one orbit by level of purple transparency magnitude), flowing counter clock-wisely around the critical point $(A_c, B_c) = \left( \frac{k_3}{k_1}, \frac{k_1}{k_2} \right)$ (black dot).

where $\mathbf{P}_A$ and $\mathbf{P}_B$ are the auto-covariance matrices of the errors, and $\mathbf{P}_{AB}$ the cross-covariance, or the inter-species covariance, of the errors. Note that, in general, $\mathbf{P}_{AB}$ is not symmetric *i.e.* $(\mathbf{P}_{AB})^{\mathrm{T}} \neq \mathbf{P}_{AB}$. The two-point cross covariance between grid points of coordiante $x$ and $y$ writes

$$\mathbf{P}_{AB}(x,y) = \sqrt{V_A(x)}\sqrt{V_B(y)}\rho_{AB}(x,y), \tag{16}$$

where

$$\rho_{AB}(x,y) = \frac{V_{AB}(x,y)}{\sqrt{V_A(x)}\sqrt{V_B(y)}}, \tag{17}$$

is the cross-correlation function, with $V_{AB}(x,y) = \overline{\varepsilon_A(x)\varepsilon_B(y)}$ the two-point cross covariance. The cross-correlation function is not symmetric in general *i.e.* $\rho_{AB}(x,y) \neq \rho_{AB}(y,x)$. In particular, if $\mathbf{C}_{AB}$ denotes the associated cross-correlation matrix, then $\mathbf{C}_{AB} \neq (\mathbf{C}_{AB})^{\mathrm{T}}$.

At a covariance modelling point of view, and in the perspective of the PKF, the univariate covariances $\mathbf{P}_A$ and $\mathbf{P}_B$ could be approximated by a VLATcov model *e.g.* $\mathbf{P}(V_A, s_A)$. Morover, the cross-covariance field $V^{AB}(x) = \overline{\varepsilon_x^A \varepsilon_x^B}$ will appear in the dynamics of $V_A$ and $V_B$ because of the coupling due to LV equations, and should be considered as a natural parameter for a multivariate PKF. At this stage, the question is whether it is possible to approximate the cross-covariance functions $V^{AB}(x,y)$ knowing the parameters $(\overline{A}, \overline{B}, V_A, V_B, V_{AB}, s_A, s_B)$ (where in this notation $V_{AB}$ denotes the field $V_{AB}(x)$).

Since no multivariate modelling extending the VLATcov model is available. A numerical exploration of the dynamics of multivariate statistics is performed for the LV-CTM, so then to guess a proxy for the cross-covariance functions.





### 3.2.1 Ensemble of multivariate forecasts

Compared to the univariate experiment described in Section 2.4.2, without a multivariate covariance model, it is not possible to sample a multivariate ensemble. For this reason, the error for the two chemical species are assumed decorrelated at the initial time $t = 0$, so that the error-covariance matrix, $\mathbf{P}^0$, is the block diagonal

$$\mathbf{P}^0 = \begin{pmatrix} \mathbf{P}_A^0 & 0 \\ 0 & \mathbf{P}_B^0 \end{pmatrix}, \tag{18}$$

where $\mathbf{P}_A^0 \ (\mathbf{P}_B^0)$ is the univariate covariance associated with error on $A$ ($B$). Following the ensemble generation of Section 2.4.2, the univariate covariance matrices are chosen as the two VLATcov $\mathbf{P}_A^0 = \mathbf{P}(V_A^0, s_A^0)$ and $\mathbf{P}_B^0 = \mathbf{P}(V_B^0, s_B^0)$. Then, an ensemble of $N_e = 6400$ initial conditions $(\mathcal{X}_k^0)_{k \in [1, N_e]}$ is sampled, with for each $k$, $\mathcal{X}_k^0 = \mathcal{X}^0 + (\mathbf{P}^0)^{1/2} \zeta_k$, where $\mathcal{X}^0 = (A^0, B^0)$ and $(\mathbf{P}^0)^{1/2}$ is the block diagonal matrix $(\mathbf{P}^0)^{1/2} = \mathrm{diag}\left(\mathbf{P}(V_A^0, s_A^0)^{1/2}, \mathbf{P}(V_B^0, s_B^0)^{1/2}\right)$. This time, $\zeta_k$ is a sample of $\mathcal{N}(0, \mathbf{I}_n)$ with $n = 2N_x$. The domain is discretized in $N_x = 723$ grid points.

For the simulation, the fields $A^0$ and $B^0$ are set to the constants $A^0 = 1.2$ and $B^0 = 0.8$. The univariate parameters are set to $\sigma_A^0 = 0.1 \cdot A^0$, $\sigma_B^0 = 0.1 \cdot B^0$, $s_A^0 = s_A^0 = l_h^2$ with $l_h = 45\Delta x \simeq 62$km. The reaction rates of LV are set to $(k_1, k_2, k_3) = (0.075, 0.065, 0.085)$. The time integration follows the numerical setting used for the univariate simulation presented in Section 2.4.2, and leads to an ensemble of $N_e = 6400$ multivariate forecasts.

While there is no cross-correlation at the initial condition, the coupling provided by the LV equations should introduce a non-zero cross-correlation between errors on $A$ and $B$, and this can be diagnosed from the computation of the cross-covariance of the forecast-errors $V^{AB}(x, y)$ at time $t$, estimated by

$$\widehat{V}^{AB}(t, x, y) = \frac{1}{N_e - 1} \sum_{k=1}^{N_e} \varepsilon_{A,k}(t, x) \varepsilon_{B,k}(t, y), \tag{19}$$

with $\varepsilon_{A,k}(t, x) = A_k(t, x) - \widehat{A}(t, x)$ and $\varepsilon_{B,k}(t, y) = B_k(t, y) - \widehat{B}(t, y)$, where $\widehat{A}$ and $\widehat{B}$ are the empirical means of the ensemble of forecasts $(A_k)$ and $(B_k)$, from which an estimation of the cross-correlation functions $\widehat{\rho}_{AB}(t, x, y)$ and matrix $\widehat{\mathbf{C}}_{AB}(t)$ can be deduced.

Figure 6 shows the time evolution of the cross-correlation with respect to the grid point $x_l = 0.5$ *i.e.* the function $\rho_{AB}(x_l, \cdot)$ (blue dashed line). As it has been specified, the cross-correlation is zero at $t = 0$ (panel a). Then, as it is expected, the cross-correlation evolves along the time, presenting an anti cross-correlation at $t = 0.6\tau_{adv}$ (panel b), then a positive one at $t = 1.8\tau_{adv}$ (panel d). At $t = 2.4\tau_{adv}$ (panel e), the cross-correlation appears clearly asymmetric, while reaching its maximum value at a $y$ strictly lower than $x_l$.

Now, a proxy for the cross-correlation is introduced from the data set of multivariate forecasts.





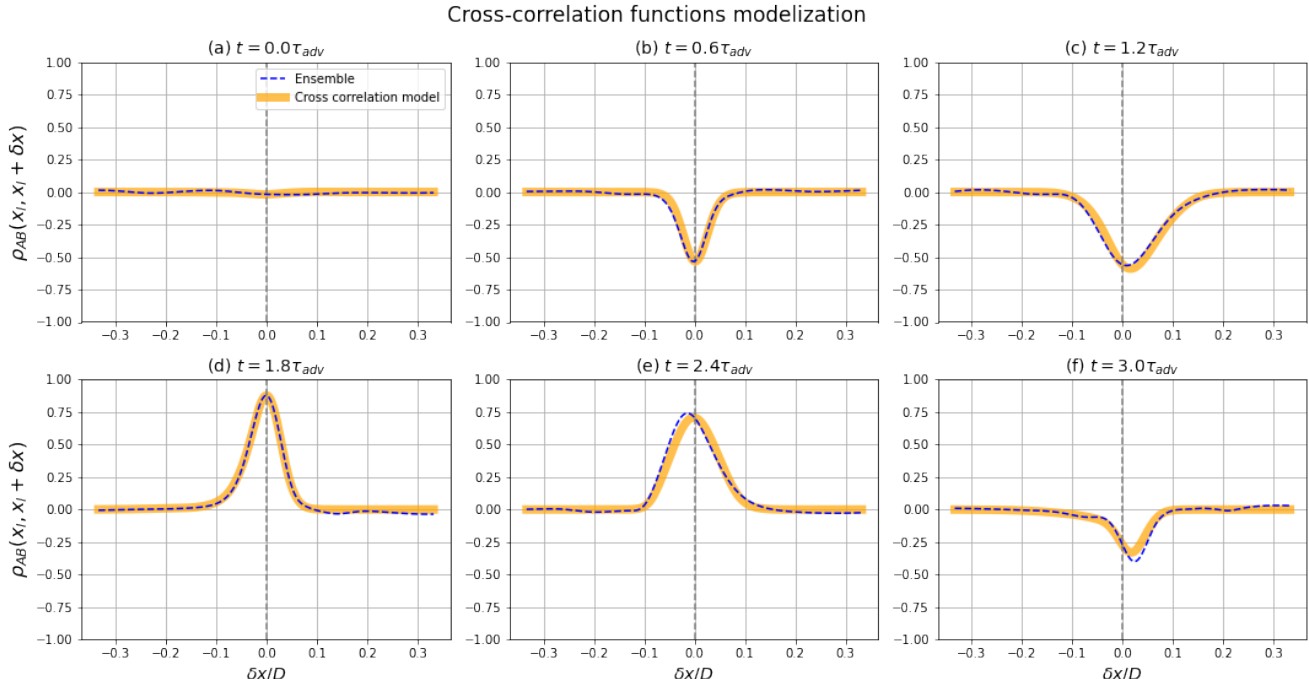

**Figure 6.** Evaluation of the cross-correlation model at location $x_l = 0.5$ and times $t = [0.0, 0.6, 1.2, 1.8, 2.4, 3.0]\tau_{adv}$.

### 3.2.2 Formulation of a proxy for the cross-correlation

After a trial-and-error process, and inspired from the VLATcov model Eq. (7), the following expression

$$r_{AB}(\mathbf{x}, \mathbf{y}) = \frac{1}{2} \left( \frac{V_{\mathbf{x}}^{AB}}{\sigma_{\mathbf{x}}^A \sigma_{\mathbf{x}}^B} + \frac{V_{\mathbf{y}}^{AB}}{\sigma_{\mathbf{y}}^A \sigma_{\mathbf{y}}^B} \right) \exp\left( -||\mathbf{x} - \mathbf{y}||^2_{[\frac{1}{4}(s_{\mathbf{x}}^A + s_{\mathbf{x}}^B + s_{\mathbf{y}}^A + s_{\mathbf{y}}^B)]^{-1}} \right), \quad (20)$$

function of the known parameters $\mathcal{P} = (V_A, V_B, V_{AB}, s_A, s_B)$, has been proposed as a proxy for the cross-correlation $\rho_{AB}$ *i.e.* $r_{AB}(x,y) \approx \rho_{AB}(x,y)$. It consists in a interpolation by the mean of the cross-correlation values at location $\mathbf{x}$ and $\mathbf{y}$, multiplied

by a gaussian kernel, where the univariate aspect-tensor has been substituted by the mean of the aspect-tensors of all chemical species. The resulting proxy for the cross-correlation matrix is denoted by $\mathbf{C}_{AB}^{\text{proxy}}(\mathcal{P})$.

One of the main advantages of considering a simple analytic formula is its can be extended to a problem with more chemical species and for a domain of higher dimension.

Note that formulation Eq. (20) is symmetric ($r_{AB}(x,y) = r_{AB}(y,x)$), while cross-correlation are not symmetric in general

($\rho_{AB}(x,y) \neq \rho_{AB}(y,x)$), but this expression leverages on all the parameters known at locations $\mathbf{x}$ and $\mathbf{y}$. However, the function $r_{AB,x}(\delta x) = r_{AB}(x, x + \delta x)$ is not necessarily symmetric in $\delta x$, where in general $r_{AB,x}(\delta x) \neq r_{AB,x}(-\delta x)$.

To assess the skill of the proxy, Fig. 6 shows the functions $r_{AB}(x_l, \cdot)$ deduced from the ensemble-estimated parameters $\widehat{\mathcal{P}}(t) = (\widehat{V_A}, \widehat{V_B}, \widehat{V_{AB}}, \widehat{s_A}, \widehat{s_B})(t)$ (bold orange line), that can be compared with the ensemble estimated cross-correlation





$\rho_{AB}(x_l, \cdot)$ (blue dashed line). At a qualitative level, the functions $r_{AB}$ are in accordance with the cross-correlation $\rho_{AB}$ of
reference for all the panels. Note that, while $r_{AB}$ is symmetric, the functions $r_{AB}(x_l, \cdot)$ can be asymmetric as it appears in
panel (c) and (f).

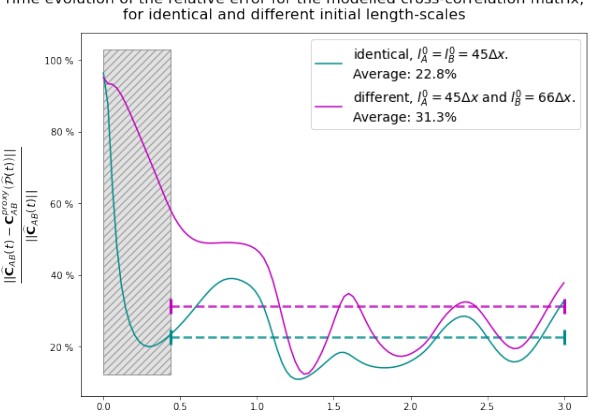

**Figure 7.** Time evolutions of the relative errors between the empirical cross-correlation matrix (EnKF) and the proxy-generated cross-correlation matrix fitted with EnKF-diagnosed parameters, for two different settings of the initial length scales: equal length-scales with $l_0^A = l_0^B = 45\Delta x \approx 66$km (turquoise line) and different length-scales with $l_0^A = 45\Delta x$ and $l_0^B = 66\Delta x \approx 91$km (mauve line). The results being dominated by sampling noise for $t < 0.45$, they are not retained (grey hatching) for the computing of the temporal averages (dashed segments).

At a quantitative level, Fig. 7 shows the time evolution of the relative error $\frac{||\widehat{\mathbf{C}}_{AB}(t) - \mathbf{C}_{AB}^{\text{proxy}}(\widehat{\mathcal{P}}(t))||}{||\widehat{\mathbf{C}}_{AB}(t)||}$, where $||\mathbf{U}|| = \sqrt{\text{Tr}(\mathbf{U}\mathbf{U}^{\text{T}})}$ is the Frobenius matrix norm where $\text{Tr}$ is the trace operator; $\widehat{\mathbf{C}}_{AB}(t)$ is the ensemble estimation of the cross-correlation matrix ; and $\mathbf{C}_{AB}^{\text{proxy}}(\widehat{\mathcal{P}}(t))$ is the proxy for the cross-correlation matrix fitted with ensemble-estimated parameters $\widehat{\mathcal{P}}(t)$. Two differ-
ent experiments are shown depending on whether the initial length-scale for a $A$ and $B$ are equal, $l_0^A = l_0^B = 45\Delta x \approx 66$km
(turquoise lines) ; or different, $l_0^A \approx 66$km but $l_0^B = 66\Delta x \approx 91$km (purple lines).

As the two multivariate error fields are uncorrelated at the initial time, the true cross-correlation matrix $\mathbf{C}_{AB}(t = 0)$ is zero. However, the ensemble used in the estimation of $\widehat{\mathbf{C}}_{AB}(t = 0)$ being finite, this produces spurious non-zero cross-correlation leading to a non-zero matrix and to a relative error larger than $80\%$. Then, the first instants of the simulation are dominated by
the sampling noise, and they are exclude for the analysis of the results (grey hatching). After $t \simeq 0.45$, the experiments offer valid results and lead to temporal averages of $22.8\%$ when $l_0^A = l_0^B$ (turquoise dashed line) and $31.3$ when $\%$ $l_0^A \neq l_0^B$ (purple dashed line). Note that the effect of the sampling noise leads to overestimate the true value of the averages by an amount of 8 points of percent for this kind of experiment (Pannekoucke, 2021).

We don't know if any formula similar to Eq. (20) has been already introduced as a possible proxy of cross-correlations. As
mentioned above, $r_{AB}$ does not share the same property of the cross-correlation (*e.g.* $r_{AB}$ is symmetric while $\rho_{AB}$ is not), and




thus, there is no guaranty that a multivariate covariance model based on the proxy $r_{AB}$ leads to a true covariance matrix: such a multivariate covariance model is symmetric because $r_{AB}$ is symmetric, but not necessarily positive definite.

Despite of the limitations of the proxy, a multivariate extension of the univariate VLATcov model is explored below, where the cross correlation is approximated by the proxy Eq. (20). This leads to a multivariate VLATcov model of parameters the fields $(V_{AB}, V_A, V_B, s_A, s_B)$ for which we can formulate a PKF.

### 3.3 Formulation and simplification of the parameters dynamics

The computation of the PKF dynamics leverages on the SymPKF package which applied to the dynamics Eq. (14), provides the following system of coupled equations

$$\partial_t A + u\partial_x A = -A\partial_x u + k_1 A - k_2 AB - k_2 V_{AB} \tag{21a}$$

$$\partial_t B + u\partial_x B = -B\partial_x u - k_3 B + k_2 AB + k_2 V_{AB} \tag{21b}$$

$$\partial_t V_{AB} + u\partial_x V_{AB} = -2V_{AB}\partial_x u + V_{AB}(k_1 - k_2 B - k_3 + k_2 A) + k_2 V_A B - k_2 V_B A \tag{21c}$$

$$\partial_t V_A + u\partial_x V_A = -2V_A\partial_x u + 2[V_A(k_1 - k_2 B) - k_2 A V_{AB}] \tag{21d}$$

$$\partial_t V_B + u\partial_x V_B = -2V_B\partial_x u + 2[V_B(-k_3 + k_2 A) + k_2 B V_{AB}] \tag{21e}$$

$$\partial_t s_A = -\underbrace{\frac{2k_2 A V_{AB} s_A}{V_A}}_{T^A_{chem-1}} + \underbrace{\frac{2k_2 A \sigma_B s_A^2 \overline{\partial_x \tilde{\varepsilon}_A \partial_x \tilde{\varepsilon}_B}}{\sigma_A}}_{T^A_{chem-2}} + \underbrace{\frac{k_2 A s_A^2 \overline{\tilde{\varepsilon}_B \partial_x \tilde{\varepsilon}_a} \partial_x V_B}{\sigma_A \sigma_B}}_{T^A_{chem-3}} - \underbrace{\frac{k_2 A \sigma_B s_A^2 \overline{\tilde{\varepsilon}_B \partial_x \tilde{\varepsilon}_A} \partial_x V_B}{V_A^{\frac{3}{2}}}}_{T^A_{chem-4}}$$

$$+ \underbrace{\frac{2k_2 \sigma_B s_A^2 \overline{\tilde{\varepsilon}_B \partial_x \tilde{\varepsilon}_A} \partial_x A}{\sigma_A}}_{T^A_{chem-5}} - \underbrace{u\partial_x s_A}_{T^A_{adv-1}} + \underbrace{2s_A \partial_x u}_{T^A_{adv-2}} \tag{21f}$$

$$\partial_t s_B = \underbrace{\frac{2k_2 B V_{AB} s_B}{V_B}}_{T^B_{chem-1}} - \underbrace{\frac{2k_2 B \sigma_A s_B^2 \overline{\partial_x \tilde{\varepsilon}_A \partial_x \tilde{\varepsilon}_B}}{\sigma_B}}_{T^B_{chem-2}} - \underbrace{\frac{k_2 B s_B^2 \overline{\tilde{\varepsilon}_A \partial_x \tilde{\varepsilon}_B} \partial_x V_A}{\sigma_A \sigma_B}}_{T^B_{chem-3}} + \underbrace{\frac{k_2 B \sigma_A s_B^2 \overline{\tilde{\varepsilon}_A \partial_x \tilde{\varepsilon}_B} \partial_x V_B}{V_B^{\frac{3}{2}}}}_{T^B_{chem-4}}$$

$$- \underbrace{\frac{2k_2 s_B^2 \overline{\tilde{\varepsilon}_A \partial_x \tilde{\varepsilon}_B} \partial_x B}{\sigma_B}}_{T^B_{chem-5}} - \underbrace{u\partial_x s_B}_{T^B_{adv-1}} + \underbrace{2s_B \partial_x u}_{T^B_{adv-2}} \tag{21g}$$

where the overline of the mean states $\overline{A}$ and $\overline{B}$ have been discarded for the sake of simplicity. The PKF is a second order filter in which the variance of the fluctuations modify the time evolution of the mean states *e.g.* by the term $-k_2 V_{AB}$ of Eq. (21a).

For the dynamics of the anisotropy, Eq. (21f) and Eq. (21g), the contributions due to the transport (to the chemistry) are labeled as $T^{(\cdot)}_{adv-(\cdot)}$ ($T^{(\cdot)}_{chem-(\cdot)}$) so to be identified. Hence, each term is labeled as $T^Z_j$, where $Z$ stands for the chemical species and $j$ for the index of the term including the processes from which the term comes from *e.g.* $T^A_{chem-5}$ denotes in Eq. (21f) the fifth term due to the chemistry in the dynamics of the anisotropy of $A$.

 

Note that the dynamics induced by the transport process is exact as mentioned in paragraph 2.4.1. In the PKF system Eq. (21)
the dynamics of the mean concentrations $A$ and $B$, variances $V_A$ and $V_B$ and covariance $V_{AB}$, Eq. (21a) to Eq. (21e), are
independant from those of anistropy fields Eq. (21f) and Eq. (21g). The reciprocal is not true: the anisotopy fields dynamics
(Eq. (21f)-Eq. (21g)) are forced by the means, the variances, covariances and their spatial heterogeneity. Eq. (21a) and Eq. (21b)
also indicate an interaction between the covariance and the mean concentrations.

The dynamics of the aspect tensors, Eq. (21f) and Eq. (21g), are not closed: some terms are expressed as expectations of
the normalized errors $\tilde{\varepsilon}_A = \varepsilon_A/\sqrt{V_A}$ and $\tilde{\varepsilon}_B = \varepsilon_B/\sqrt{V_B}$. These open terms can not be directly expressed using the available
parameters, preventing the forecast of the error statistics. The role and magnitude of these terms is studied in the following
paragraphs (3.3.1-3.3.3).

Several experiments are conducted in the following paragraphs to better understand the impact of this closure problem. In
paragraph 3.3.1, following the splitting approach (Pannekoucke et al., 2021), the transport terms are removed so to focus on
the contribution of the chemistry process for observing its influence on the uncertainty dynamics, in the case of homogeneous
statistical initial conditions. A comparison between the uncertainty dynamics in LV equations with the one for the harmonic
oscillator (HO) problem is also carried out. Then, in paragraph 3.3.2, the transport is rehabilitated so to quantify which of
the two processes at play is dominant in the anisotropy dynamics. Eventually a simplification is made in the dynamics of the
anistropy to close the PKF dynamics.

**3.3.1    Impact of the chemistry alone on the dynamics of the anisotropies for homogeneous statistical initial conditions**

Regarding the dynamics of the anisotropy fields presented in the prognostic equations (Eq. (21f)-Eq. (21g)), the part due to
transport in $T_{adv-(\cdot)}^{(\cdot)}$, is already well understood, as it comes down to the univariate case presented in paragraph 2.4.2. However,
the role of the chemistry in $T_{chem-(\cdot)}^{(\cdot)}$, is unclear at this time. The transport process is removed so to focus on the dynamics of
the anisotropy due to the chemistry.

In the PKF dynamics Eq. (21), when there is no transport, and when the variance fields are homogeneous at the initial
condition, the homogeneity is preserved during the time evolution. Hence, the spatial derivatives of the variance and of the
cross-variance fields are null, which leads to simplify the dynamics of the anisotropy (Eq. (21f)-Eq. (21g)) as

$$\partial_t s_A = \frac{2k_2 A s_A}{\sigma_A}\left(\sigma_B s_A \overline{\partial_x \tilde{\varepsilon}_A \partial_x \tilde{\varepsilon}_B} - \frac{V_{AB}}{\sigma_A}\right), \tag{22a}$$

$$\partial_t s_B = \frac{2k_2 B s_B}{\sigma_B}\left(\frac{V_{AB}}{\sigma_B} - \sigma_A s_B \overline{\partial_x \tilde{\varepsilon}_A \partial_x \tilde{\varepsilon}_B}\right). \tag{22b}$$

To focus on the contribution of the chemistry on the dynamics of the anisotropies, an ensemble of high resolution forecasts
is performed ($N_x = 723$), with only the chemistry part. Hence, the transport terms are set to zero in Eq. (14). Two numerical
experiments are conducted: first, the initial length-scales are equal for both species with $l_0^A = l_0^B = 45\Delta x \simeq 62$km (results are
shown in Fig. 8), then different with $l_0^A = 45\Delta x$ and $l_0^B = 66\Delta x \simeq 91$km (results in Fig. 9). The initial conditions for the





concentrations, and the multivariate statistics are chosen homogeneous over the domain in both cases. Therefore, only the time
series of the spatial average are shown for the variance, the cross-correlation, the length-scale and the open term $\overline{\partial_x \tilde{\varepsilon}_A \partial_x \tilde{\varepsilon}_B}$.

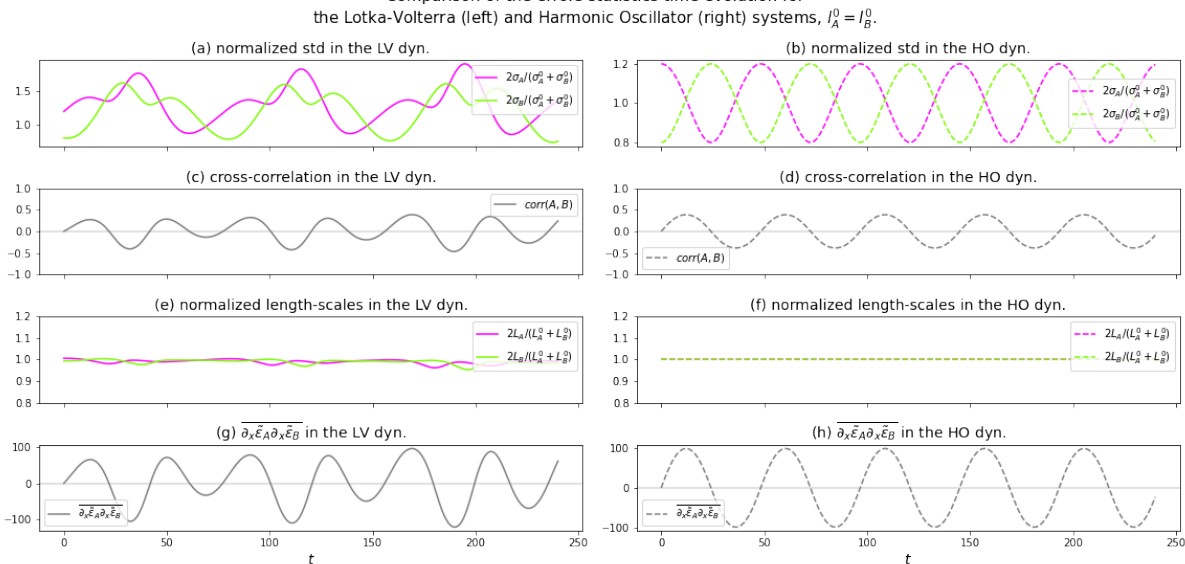

**Figure 8.** Time series of the spatial average of the error statistics: from the ensemble forecast with $N_e = 1600$ for Lotka-Volterra (LV, left panels) and Harmonic Oscillator analytical solutions (HO, right panels). Equal initial length-scales: $l_0^A = l_0^B = 45\Delta x$.

In the first experiment, Fig. 8, the magnitude of the error, given by the standard deviations (panel a), oscillates with a phase shift where the magnitude of the error in $A$ advances the one of $B$. The cross-correlation (panel c) and the unclosed term $\overline{\partial_x \tilde{\varepsilon}_A \partial_x \tilde{\varepsilon}_B}$ (panel g) oscillate in a similar way. In this experiment, where the initial length-scales are identical for $A$ and $B$, there is no time evolution of the length-scales, except the fluctuations that are due to the sampling noise (see panel e). The
second experiment, Fig. 9, shows roughly the same picture, except that this time, with initial length-scales of different values, oscillations are appearing (panel e). Since, *a priori*, it is not easy to track the reason for the change of behaviour observed on the length-scale dynamics, an analytical investigation of the harmonic oscillator (HO)

$$\partial_t A(t, \mathbf{x}) = -kB(t, \mathbf{x}), \tag{23a}$$

$$\partial_t B(t, \mathbf{x}) = kA(t, \mathbf{x}), \tag{23b}$$

is introduced, with $k = k_2$. The comparison with HO is relevant since it is an example of analytical multivariate dynamics and also because it mimics the periodic oscillations of LV, explaining the numerical results. For HO, it is possible to calculate the



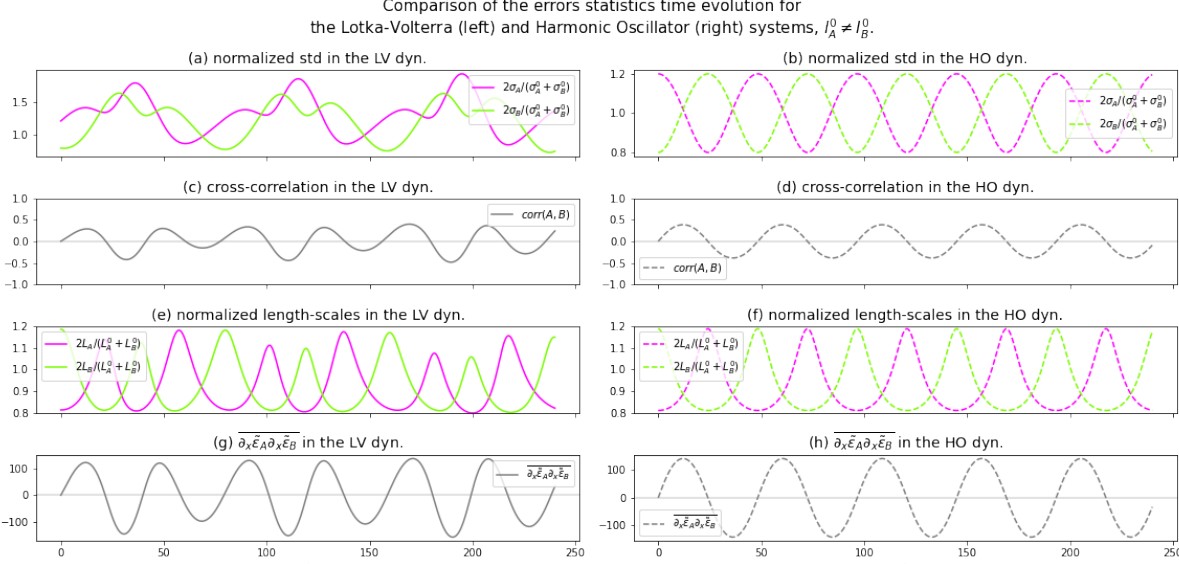

**Figure 9.** Time series of the spatial average of the error statistics: from the ensemble forecast with $N_e = 1600$ for Lotka-Volterra (LV, left panels) and Harmonic Oscillator analytical solutions (HO, right panels). Different initial length-scales: $l_0^A = 45\Delta x$ and $l_0^B = 66\Delta x$.

time evolution of the statistics (see Appendix B for details), which writes as

$$V_A(t) = \cos(kt)^2 V_A^0 + \sin(kt)^2 V_B^0, \tag{24a}$$

$$V_B(t) = \sin(kt)^2 V_A^0 + \cos(kt)^2 V_B^0, \tag{24b}$$

$$V_{AB}(t) = \cos(kt)\sin(kt)\left(V_A^0 - V_B^0\right), \tag{24c}$$

$$g_A(t) = \frac{1}{V_A(t)}\left[\cos(kt)^2 V_A^0 g_A^0 + \sin(kt)^2 V_B^0 g_B^0\right], \tag{24d}$$

$$g_B(t) = \frac{1}{V_B(t)}\left[\sin(kt)^2 V_A^0 g_A^0 + \cos(kt)^2 V_B^0 g_B^0\right], \tag{24e}$$

$$\mathbb{E}[\partial_x \tilde{\varepsilon}_A \partial_x \tilde{\varepsilon}_B](t) = \frac{\cos(kt)\sin(kt)}{\sigma_A(t)\sigma_B(t)}\left[V_A^0 g_A^0 - V_B^0 g_B^0\right]. \tag{24f}$$

Numerical results computed for the HO are represented in Fig. 8 and Fig. 9, and show some of the behaviour encountered
for the nonlinear LV equations. For instance, the oscillations of the variance are visible. Moreover, the length-scales oscillate
depending on the initial condition: when the initial length-scales are equal, there is no oscillations (see Fig. 8-(d)) that appear
from the analytical computation of $g_A$ and $g_B$ ; at the opposite for different values of the initial length-scales, oscillations appear
(see Fig. 9-(d)). These different behaviours of the anisotropy based on the initial settings of the length-scales are explained by
the analytical solutions of the error statistics for the harmonic oscillator. For instance, when plugging the identical initial
condition for the length-scales $g_A^0 = g_B^0$ and the analytical solution of $V_A(t)$ (Eq. 24a) into the r.h.s. of Eq. (24d), it simplifies





to $g_A(t) = g_A^0$. The same result applies for $g_B(t)$. This simplification no longer holds when $g_0^A \neq g_B^0$, leading to non constants length-scales which is effectively observed.

Note that for equal initial length-scales, the anisotropy appears stationnary (see Fig. 8e), which suggests a closure for the open term $\overline{\partial_x \tilde{\varepsilon}_A \partial_x \tilde{\varepsilon}_B}$: since the anisotropy are equal and constant, $s_A(t) = s_B(t) = \frac{s_A(t)+s_B(t)}{2} = s_A^0 = s_B^0 = \frac{s_A^0+s_B^0}{2}$, then

from the stationnarity of the anisotropy, $\partial_t s_A = \partial_t s_B = 0$, the right-hand side of Eqs. (22) leads to the expression

$$\overline{\partial_x \tilde{\varepsilon}_A \partial_x \tilde{\varepsilon}_B} = \frac{V_x^{AB}}{\sigma_x^A \sigma_x^B} \frac{2}{s_x^A + s_x^B}. \tag{25}$$

This closure indicates that the term $\overline{\partial_x \tilde{\varepsilon}_A \partial_x \tilde{\varepsilon}_B}$ is proportional to the cross-correlation in this particular case. This is confirmed in Fig. 8, where $\overline{\partial_x \tilde{\varepsilon}_A \partial_x \tilde{\varepsilon}_B}$ (panel g) appears to evolve as the cross-correlation (panel c). For this specific case, Eq. (25) also applies for the error statistics of the harmonic oscillator: using $g_A^0 = g_B^0$ and the time evolution of the covariance $V_{AB}$ (Eq. 24c)

allows to solve for the open term in Eq. (24f), obtaining the same expression as in Eq. (25).

The time evolution of the HO error statistics makes appear a swing of the error statistics between the two components $A$ and $B$, which qualitatively reproduces the evolution observed in the LV dynamics. The transfer of uncertainty from one component to the other is provided by the cross-covariance $V_{AB}$ when the magnitude of the uncertainty is different along each specie.

After this focus on the dynamics of the anisotropy due to the chemistry alone, the study of the contribution of the chemistry

and the transport is conducted, assessing the magnitude of each terms and processes on the dynamics of the anisotropy.

### 3.3.2    Detailed contribution of each processes in the dynamics of the anisotropy

What follows aim to identify the dominant terms, or processes in the dynamics of the anisotropy, Eq. (21f) and Eq. (21g).

Two different evaluations are performed. The first one evaluates the relative contribution $W_j^Z$ of the term $T_j^Z$ among all other terms in the dynamics of the anisotropy of $Z$, which reads as

$$W_j^Z(t) = \frac{||T_j^Z(t)||_1}{\sum_k ||T_k^Z(t)||_1}, \tag{26}$$

where $||v||_1 = \frac{1}{N_x} \sum_{j=1,...,N_x} |v_j|$ is the $L^1$ norm on the discretized domain $[0, D)$. The second one evaluates the relative contribution of each physical processes in the dynamics of the anisotropy *e.g.* the relative contribution of the advection in the dynamics of the anisotropy of $Z$, $W_{adv}^Z$, reads as

$$W_{adv}^Z(t) = \frac{||\sum_{k=1}^2 T_{adv-k}^Z(t)||_1}{||\sum_{k=1}^2 T_{adv-k}^Z(t)||_1 + ||\sum_{k=1}^5 T_{chem-k}^Z(t)||_1}, \tag{27}$$

from which the relative contribution of the chemistry writes $W_{chem}^Z(t) = 1 - W_{adv}^Z(t)$.

The computation of these relative contributions will rely on ensemble of forecasts. They will be used to diagnose *a posteriori* the PKF parameters $(A, B, V_A, V_B, V_{AB}, s_A, s_B)$ as well as the three open terms $(\overline{\partial_x \tilde{\varepsilon}_A \partial_x \tilde{\varepsilon}_B}, \overline{\tilde{\varepsilon}_A \partial_x \tilde{\varepsilon}_B}, \overline{\tilde{\varepsilon}_B \partial_x \tilde{\varepsilon}_A})$ to then reconstruct all the terms in anisotropy dynamics (Eq. (21f)-Eq. (21g)).



The quantifications of the relative contribution by term and by process will be performed for equal and different initial length-scales for $A$ and $B$, as it leads to different dynamics for the anisotropy. Thus, two ensembles are forecasted, with initial length-scales set to $l_0^A = l_0^B = 45\Delta x$ in the first, and $l_0^A = 45\Delta x$ and $l_0^B = 66\Delta x$ in the second. A high resolution grid is considered ($N_x = 723$) to reduce numerical model error; the time step has been adapted in consequence to match the CFL. The other settings as well as the numerical configuration for this experiment are unchanged from previous ensemble forecast performed in paragraph 3.2.1.

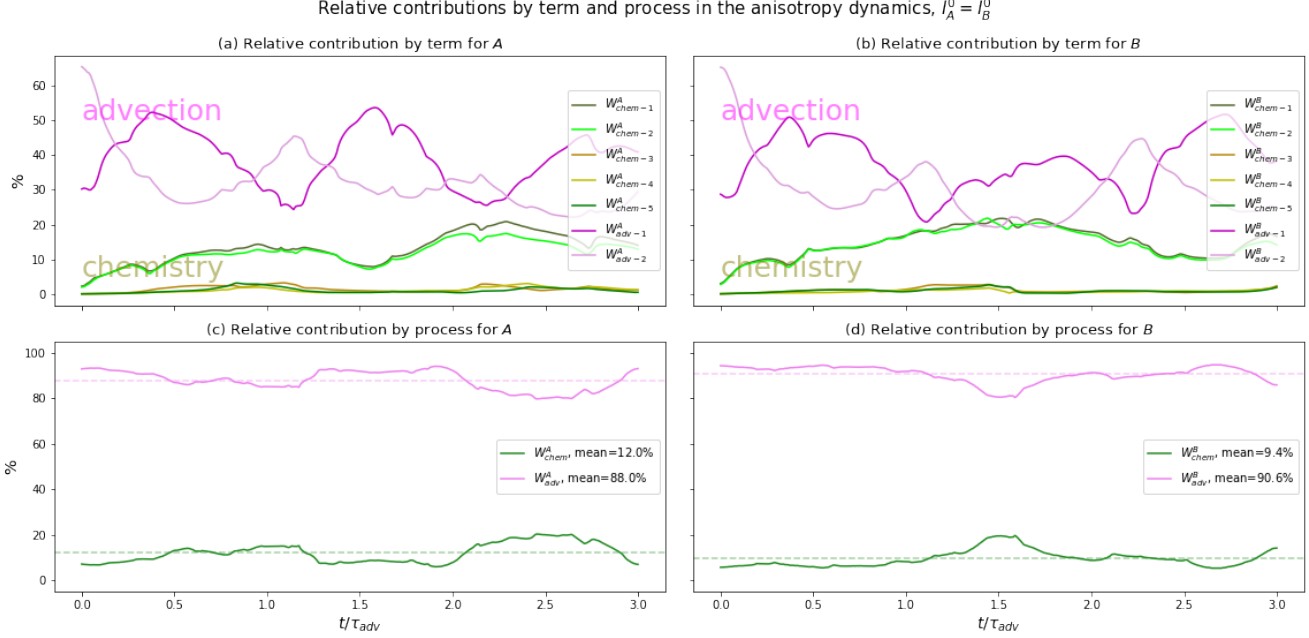

**Figure 10.** Numerical results for the case $l_0^A = l_0^B = 45\Delta x$. Time evolution for the relative contribution by term (resp. by process) involved in the anisotropy dynamics for species $A$ and $B$ on panels (a) and (b) (resp. panels (c) and (d)).

The results of the relative contributions presented in Fig. 10 (Fig. 11) for the equal (different) length-scale configuration are now discussed. Regarding the relative contribution by process experiment, the comparison between panels Fig. 10c (Fig. 10d) and Fig. 11c (Fig. 11d) indicates that when the initial length-scales are different, $l_0^A \neq l_0^B$, the chemistry has a more significant role ($W_{chem}$ is about 21%) compared when the length-scales are equal ($W_{chem}$ is about 10%) in the dynamics of the anisotropies. That difference was expected following the results obtained in the previous paragraph 3.3.1. Now focusing on the relative contribution by term on panels (a) and (b), it is noticeable that only the two terms, $W_{chem-1}^Z$ and $W_{chem-2}^Z$, have a significant role in the dynamics. The rest of the chemistry-related terms magnitudes are negligible. For equal initial length-scales, as the chemistry-related part of the anisotropy dynamics can be neglected, and as this part is mainly driven by $W_{chem-1}^Z$ and




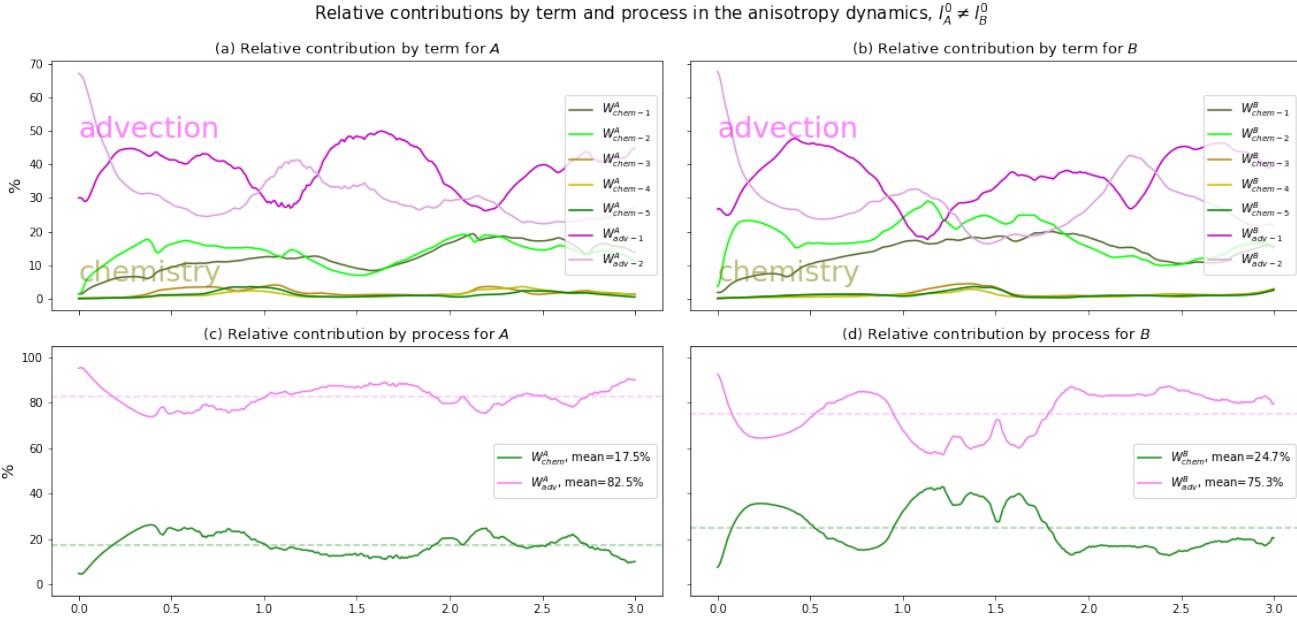

**Figure 11.** Numerical results for the case $l_0^A = 45\Delta x$ and $l_0^B = 66\Delta x$. Time evolution for the relative contribution by term (resp. by process) involved in the anisotropy dynamics for species $A$ and $B$ on panels (a) and (b) (resp. panels (c) and (d)).

$W_{chem-2}^Z$, this means an approximate compensation of the two terms. Eventually, this approximation simplifies to

$$\overline{\partial_x \tilde{\varepsilon}_A \partial_x \tilde{\varepsilon}_B} \simeq \frac{V_x^{AB}}{\sigma_x^A \sigma_x^B} \cdot \frac{2}{s_x^A + s_x^B}, \tag{28}$$

which is in accordance with the previous results of paragraph 3.3.1. However, this approximation becomes invalid in the heterogeneous case: the terms $W_{chem-1}^Z$ and $W_{chem-2}^Z$ no longer compensate each other as the gap between their corresponding curves increases in panels Fig. 11(c) and Fig. 11(d). In some other numerical trials (not shown here), this approximation was used regardless of the length-scales initial configuration, and the remaining open terms were set to zero. These trials produced incoherent forecasts for the anisotropy, pointing out the incapacity of the approximation to capture the true complexity of the

unknown terms. Subsequently, this approximation is no longer retained.

  Note that the open terms of the PKF dynamics Eq. (21) can be related to spatial derivatives of the cross-correlation Eq. (17) e.g. $\overline{\tilde{\varepsilon}_A \partial_x \tilde{\varepsilon}_B}(x) = (\partial_x \rho_{AB})(x,x)$ or $\overline{\partial_x \tilde{\varepsilon}_A \partial_x \tilde{\varepsilon}_B}(x) = (\partial_{xy} \rho_{AB})(x,x)$, leading to a closure of the PKF dynamics when the proxy $r_{AB}$ Eq. (20) is used in place of the true cross-correlation $\rho_{AB}$. However, numerical investigation of this closure did not lead to good results (not shown here).

From the detailed quantification of the relative contributions conducted here, it results that the advection contributes to 80% of the anisotropy dynamics while 20% are due to the chemistry.

  Next, a closure is proposed for the LV-CTM multivariate PKF dynamics.





### 3.3.3 Simplification of the PKF dynamics

Since the advection mainly leads the dynamics of the anisotropy, this suggests to remove the contribution of the chemistry in
Eq. (21f) and Eq. (21g), and leads to a close the PKF dynamics Eq. (21) as

$$\partial_t A + u\partial_x A = -A\partial_x u + k_1 A - k_2 AB - k_2 V_{AB} \tag{29a}$$

$$\partial_t B + u\partial_x B = -B\partial_x u - k_3 B + k_2 AB + k_2 V_{AB} \tag{29b}$$

$$\partial_t V_{AB} + u\partial_x V_{AB} = -2V_{AB}\partial_x u + V_{AB}(k_1 - k_2 B - k_3 + k_2 A) + k_2 V_A B - k_2 V_B A \tag{29c}$$

$$\partial_t V_A + u\partial_x V_A = -2V_A\partial_x u + 2[V_A(k_1 - k_2 B) - k_2 A V_{AB}] \tag{29d}$$

$$\partial_t V_B + u\partial_x V_B = -2V_B\partial_x u + 2[V_B(-k_3 + k_2 A) + k_2 B V_{AB}] \tag{29e}$$

$$\partial_t s_A = -u\partial_x s_A + 2s_A\partial_x u \tag{29f}$$

$$\partial_t s_B = -u\partial_x s_B + 2s_B\partial_x u \tag{29g}$$

The PKF dynamics for the multivariate LV-CTM being closed, it remains to detail the multivariate analysis step to be used
in the PKF assimilation cycle.

### 3.4 Extension of the PKF *analysis* step for multivariate assimilations

For multivariate statistics, the update equations (8) presented in section 2 have to be modified: they can be applied to update
the univariate error statistics (mean concentrations, variances, aspect-tensors) but do not indicate how to update the cross-
covariance fields. To apply the formulas Eqs. (8) in multivariate contexts, the subscript $l$ must be interpreted as a location and
the observed species, and $\mathbf{x}$ as any location on any species. For an observation at location $\mathbf{x}_l$ on the chemical species $Z_l$, the
cross-covariance field between two species $Z_1$ and $Z_2$ updates (see Appendix C):

$$V_{Z_1 Z_2}^a(\mathbf{x}) = V_{Z_1 Z_2}^f(\mathbf{x}) - \left(\sigma_{Z_2}^f(\mathbf{x})\rho_{Z_l,l}^{Z_2,f}(\mathbf{x})\sigma_{Z_1}^f(\mathbf{x})\rho_{Z_l,l}^{Z_1,f}(\mathbf{x})\right) \frac{V_{Z_l}^f(\mathbf{x}_l)}{V_{Z_l}^f(\mathbf{x}_l) + V_{Z_l}^o(\mathbf{x}_l)}, \tag{30}$$

where $\rho_{Z_l,l}^{Z_1,f}(\mathbf{x})$ is the cross-correlation function between $Z_l$ and $Z_1$ at location $\mathbf{x}_l$, defined by

$$\rho_{Z_l,l}^{Z_1,f}(\mathbf{x}) = \mathbb{E}\left[\varepsilon_{Z_l}^f(\mathbf{x}_l)\varepsilon_{Z_1}^f(\mathbf{x})\right] / \left(\sigma_{Z_l}^f(\mathbf{x}_l)\sigma_{Z_1}^f(\mathbf{x})\right). \tag{31}$$

Note that Eq. (30) also applies when one of the two chemical species $Z_1$ or $Z_2$ coincides with $Z_l$. This conduct to a new
formulation of the algorithm PKFO1 (alg. 1).

Next, the resulting multivariate PKF formulation is validated from a numerical experiment.

### 3.5 Numerical experiments: simple forecast and data assimilation over several cycles

In this section, two numerical experiments, labeled FCST and DA, are proposed to evaluate the multivariate formulation of
the PKF for the LV-CTM. Again, a large EnKF will be used as a reference to be compared with regarding the error statistics





---

**Algorithm 1** Sequential process building the analysis state and its error covariance matrix for the first-order PKF (PKFO1) with pseudo multivariate covariance model.

---

**Require:** Univariate fields of $\mathcal{X}_Z^f, \mathbf{s}_Z^f$ and $V_Z^f$ for all species $Z$. Covariance field $V_{Z_1 Z_2}^f$ of all pairs of species $Z_1$ and $Z_2$. Variance $V_{Z_l,l}^o$ of the species $Z_l$ and locations $\mathbf{x}_l$ of the $p$ observations to assimilate.

1: **for** $l = 1 : p$ **do**

2:      *0 - Initialization of the intermediate quantities*

3:      $\mathcal{Y}_{Z_l,l}^o = \mathcal{Y}_{Z_l}^o(\mathbf{x}_l), \mathcal{X}_{Z_l,l}^f = \mathcal{X}_{Z_l}^f(\mathbf{x}_l)$

4:      $V_{Z_l,l}^f = V_{Z_l,\mathbf{x}_l}^f, V_{Z_l,l}^o = V_{Z_l,\mathbf{x}_l}^o$

5:

6:      *1 - Computation of the analysis univariate statistics*

7:      **for** each species $Z$ **do**

8:          *a) Set the correlation function (auto or cross)*

9:          $\rho_{Z_l,l}^Z(\mathbf{x}) = \rho(V_{Z_l Z}^f, V_{Z_l}^f, V_Z^f, \mathbf{s}_{Z_l}^f, \mathbf{s}_Z^f)(\mathbf{x}_l, \mathbf{x})$

10:

11:          *b) Computation of the analysis state and its univariate error statistics*

12:          $\mathcal{X}_{Z,\mathbf{x}}^a = \mathcal{X}_{Z,\mathbf{x}}^f + \sigma_{Z,\mathbf{x}}^f \rho_{Z_l,l}^Z(\mathbf{x}) \frac{\sigma_{Z_l,l}^f}{V_{Z_l,l}^f + V_{Z_l,l}^o}\left(\mathcal{Y}_{Z_l,l}^o - \mathcal{X}_{Z_l,l}^f\right),$

13:          $V_{Z,\mathbf{x}}^a = V_{Z,\mathbf{x}}^f\left(1 - \left[\rho_{Z_l,l}^Z(\mathbf{x})\right]^2 \frac{V_{Z_l,l}^f}{V_{Z_l,l}^f + V_{Z_l,l}^o}\right)$

14:          $\mathbf{s}_{Z,\mathbf{x}}^a = \frac{V_{Z,\mathbf{x}}^a}{V_{Z,\mathbf{x}}^f}\mathbf{s}_{Z,\mathbf{x}}^f$

15:      **end for**

16:

17:      *2 - Computation of the analysis multivariate statistics*

18:      **for** each pair of species $(Z_i, Z_j, \text{ with } i < j)$ **do**

19:          *a) Set the cross-correlation functions*

20:          $\rho_{Z_l,l}^{Z_i}(\mathbf{x}) = \rho(V_{Z_l Z_i}^f, V_{Z_l}^f, V_{Z_i}^f, \mathbf{s}_{Z_l}^f, \mathbf{s}_{Z_i}^f)(\mathbf{x}_l, \mathbf{x})$

21:          $\rho_{Z_l,l}^{Z_j}(\mathbf{x}) = \rho(V_{Z_l Z_j}^f, V_{Z_l}^f, V_{Z_j}^f, \mathbf{s}_{Z_l}^f, \mathbf{s}_{Z_j}^f)(\mathbf{x}_l, \mathbf{x})$

22:

23:          *b) Compute the $Z_i Z_j$ analysis covariance field*

24:          $V_{Z_i Z_j}^a(\mathbf{x}) = V_{Z_i Z_j}^f(\mathbf{x}) - \left(\sigma_{Z_j}^f(\mathbf{x})\rho_{Z_l,l}^{Z_j,f}(\mathbf{x})\sigma_{Z_i}^f(\mathbf{x})\rho_{Z_l,l}^{Z_i,f}(\mathbf{x})\right)\frac{V_{Z_l}^f(\mathbf{x}_l)}{V_{Z_l}^f(\mathbf{x}_l)+V_{Z_l}^o(\mathbf{x}_l)}$

25:      **end for**

26:

27:      *3 - Update of the forecast state and its error statistics*

28:      **for** each species $Z$ **do**

29:          $\mathcal{X}_{Z,\mathbf{x}}^f \leftarrow \mathcal{X}_{Z,\mathbf{x}}^a$

30:          $V_{Z,\mathbf{x}}^f \leftarrow V_{Z,\mathbf{x}}^a$

31:          $\mathbf{s}_{Z,\mathbf{x}}^f \leftarrow \mathbf{s}_{Z,\mathbf{x}}^a$

32:      **end for**

33:

34:      **for** each pair of species $(Z_i, Z_j)$ **do**

35:          $V_{Z_i Z_j}^f(\mathbf{x}) \leftarrow V_{Z_i Z_j}^a(\mathbf{x})$

36:      **end for**

37: **end for**





produced. The first experiment, FCST, focuses on the on the forecast step alone. Therefore, the PKF dynamics (Eq. (29)) and the EnKF for equations (Eq. (14)) are forecasted. Then, in DA, 5 complete data assimilation cycles are performed to test the PKF capacity to produce multivariate analysis. DA only differs from FCST by the assimilations of observations, otherwise the configurations are identical. The next paragraphs details the settings of the experiments.

### 3.5.1 Settings of the numerical experiments

In both experiments, the EnKF relies on 6400 members. The total time of the simulation is $t_{max} = 5\tau_{adv}/3 \simeq 47.5$ hours ($\tau_{adv}$ is the characteristic time defined in section 2.4.2). A high resolution with $N_x = 723$ grid points is used. The settings of the wind field, chemical rates, initial concentrations, initial variances and covariance, time scheme, space grid etc. are identical to those used in section 3.2. The initial length-scale fields are homogeneously initialized at $l_0^A = l_0^B = 45\Delta x$.

For the data assimilation experiment, a network of 4 sensors regularly spaced on the right hand side of the domain is considered to generate observations of the chemical species $A$. Each $\tau_{adv}/3$ hours, observations are generated from an independant nature run and assimilated for both filters. The nature run is initialized with fields concentrations $A$ and $B$ set respectively to $1.2 + 0.12\zeta_A$ and $0.8 + 0.08\zeta_B$, where $\zeta_A$ and $\zeta_B$ are structured Gaussian random field of zero mean, standard-deviation 1 and length-scale $45\Delta x$ (*i.e.* sampled from $\mathbf{P}\left(1, (45\Delta x)^2\right)$ in Eq. (7)). The synthetic observations are considered uncorrelated (*i.e.* $\mathbf{R}$ is diagonal), and generated according to: $A^{obs}(x_l) = A_{NR}^f(x_l) + \sigma^{obs}\zeta$, where $\sigma^{obs} = 10\%$ is the observations standard-deviation, $\zeta$ is a sample from the standard Gaussian distribution, and $A_{NR}^f$ is the forecast of the nature run for location $x_l$. The model error is neglected in this experiment (*i.e.* $\mathbf{Q} = \mathbf{0}$ in Eq. 3b). For the PKF, the observations are assimilated using the PKF O1 algorithm.

### 3.5.2 Results

The results for the FCST experiment Fig. 12 are now discussed. The figure presents the state vector (panels a and b) and five error statistics (panels c-g) for the EnKF (dashed blue lines) and the PKF (solid red lines) at $t = 0.5t_{max}$ and $t = t_{max}$. The error statistics presented are, from panel (c) to (g), the two standard-deviations, the cross-correlation field and the two length-scales, rather than the raw PKF parameters. An horizontal grey line on each panel is here to represent the initial setting of the corresponding quantity.

The forecasts of the means match perfectly for both methods (panels a and b). Similarly to the univariate advection experiment (section 2.4.2), an accumulation of the tracers is observed in the low wind speeds region (center of the domain). The standard-deviations (panels c-d) observe a similar behaviour although the effects of the chemistry appear more clearly: the curves show some quite localized deformations, especially for the standard deviation of $A$. The cross-correlation field (panel e), specific to the multivariate case, is predicted with great accuracy by the PKF dynamics. It indicates that, starting from decorrelated error fields for $A$ and $B$, the chemistry dynamic has allowed non-zero cross-correlations to emerge by coupling the chemical species, in a non-linear fashion. With less exactitude than for the means, the filters coincide at estimating the standard-deviation as well as for the cross-correlation fields. The last two panels (f) and (g) which corresponds to the forecasts of the length-scales show a general accordance between the two methods, even though a difference can be observed in $A$'s case



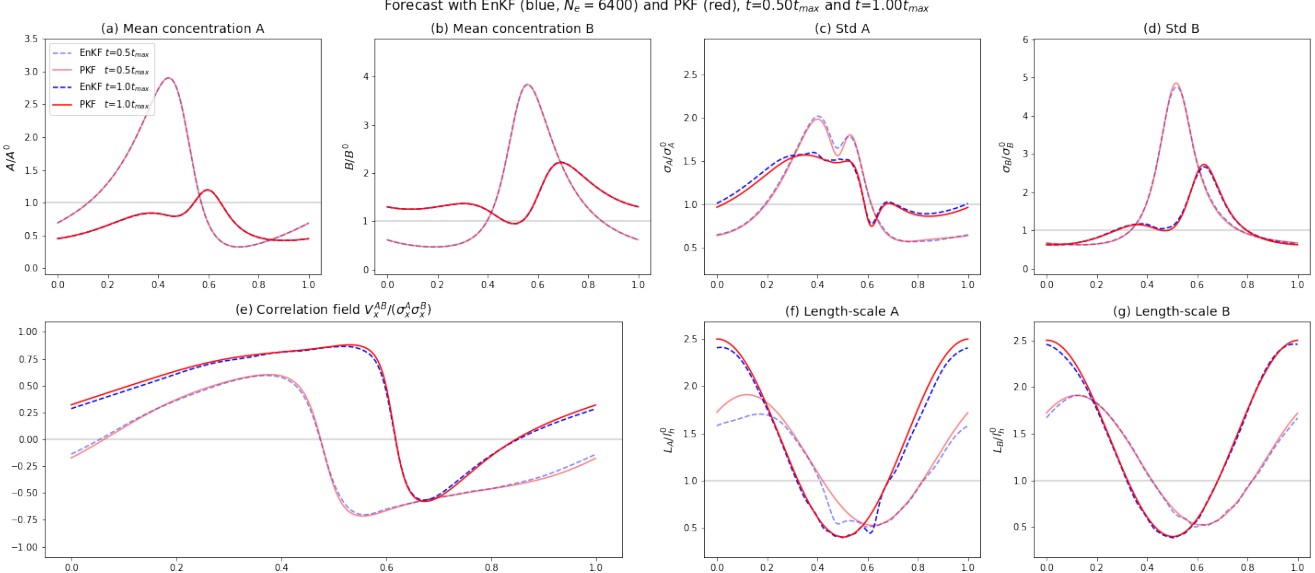

**Figure 12.** Results of the forecast numerical experiment. PKF errors statistics (solid red lines) and EnKF diagnosed error statistics (dashed blue lines) at times $t = [0.50, 1.00]t_{max}$. These times correspond approximately to $t$=23h45min and $t$=47h40min.

(f). This gap is due to the simplification of the anisotropy dynamics in the PKF formulation Eqs. (29), which does not permit to represent such behaviours. The equation of the anisotropy dynamics of $A$ in the original formulation of the PKF Eq. (21f) suggests an explanation to the spikes presented on the EnKF curves on panel (f) which are absent for the PKF. The terms labeled $T^A_{chem-3}$ and $T^A_{chem-4}$ indicate a forcing of the spatial derivatives of the variance $V_A$. Looking at panel (c), it appears that the variance of $A$ presents some strong spatial heterogeneity ($x = 0.45$ for $t = 0.5t_{max}$, and $x = 0.60$ for $t = t_{max}$), causing important magnitudes for $\partial_x V_A$ and thus for $T^A_{chem-3}$ and $T^A_{chem-4}$. This produces a local deformation on $A$'s length-scales which is effectively observed for the same times and locations on panel (f). However, these gaps between the EnKF and PKF curves are local and of a reasonable magnitude: overall, the PKF forecast for the anisotropy reproduces the EnKF results.

The outcome of the DA experiment Fig. 13 is now exposed. The results are presented similarly to the FCST experiment, except four vertical grey lines have been added to indicate the sensors locations. Also, time $t = t_{max}$ corresponds to a time for which synthetic observations for $A$ are generated, thus represented by black dots on panel (a).

For the DA experiment (Fig. 13), the resulting means on panel (a) and (b) are identical for the PKF and EnKF. This indicates similar forecasts and analysis for both methods during the five assimilation cycles. However, the corrections brought by the observations are not very significant given the neglected model error, the small amplitude of the forecast variance and the observation error. This configuration implied the generated observations to be very close to the forecasted concentrations, therefore the means are not significantly different than in the FCST experiment. The impact of the different analysis is more visible on the rest of the error statitics. For instance, the curves on panel (c) present important downspikes which result from the uncertainty reduction during the analysis. This reduction of the uncertainty is also visible, with a reduced amplitude, on



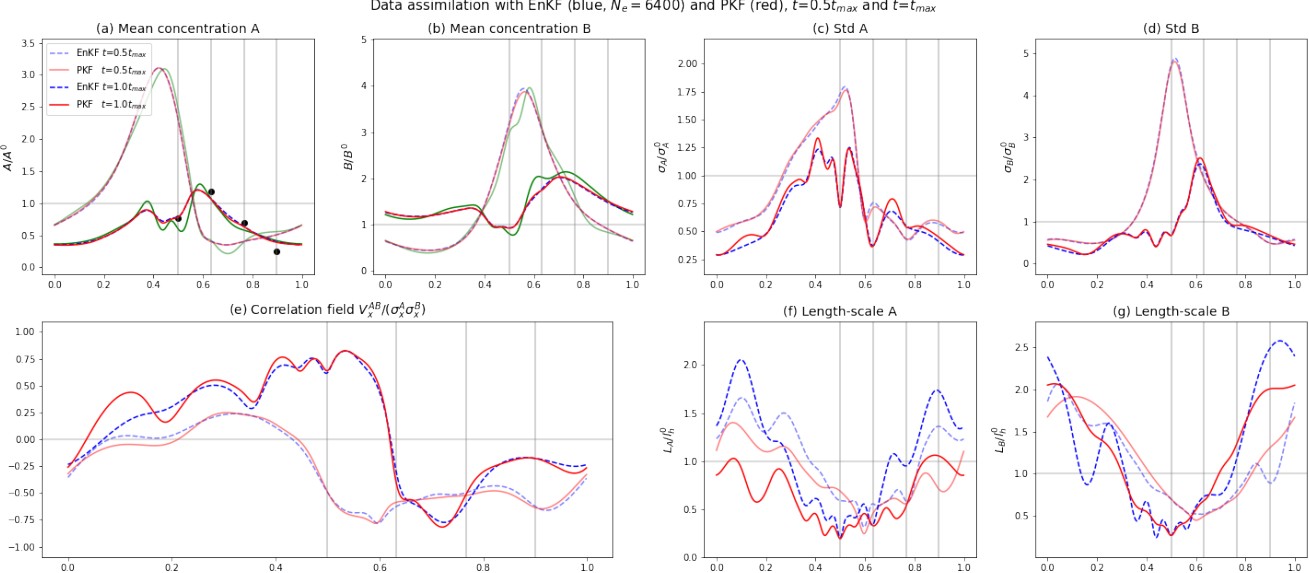

**Figure 13.** Results of the data assimiliation numerical experiment. Nature run (solid green lines), PKF errors statistics (solid red lines) and EnKF diagnosed error statistics (dashed blue lines) at times $t = [0.50, 1.00]t_{max}$. These times correspond approximately to $t$=23h45min and $t$=47h40min. At time $t = 0.5t_{max}$, two analysis steps have already been performed. At time $t = 1.00t_{max}$, the fifth analysis step is being realized, the generated observations are represented by black dots on panel (a). The vertical grey lines correspond to the sensors locations.

specie $B$ for which we do not have observations, as panel (d) shows. The ability to reduce the uncertainty of $B$ and to correct its concentration when $A$ is observed is the signature of the multivariate character of the analysis. The amplitude of the reduction of $\sigma_B$ and correction of $B$ is related to the strength of the cross-correlation at the moment of assimilation. The cross-correlation

635    field (e) is also impacted by the observation but it is less obvious to say in which manner. Looking at panel (f), an important gap between the PKF and EnKF for the length-scales of $A$ can be observed. It is caused by two reasons, the major one being the approximation in the anisotropy update formula Eq. (8c). This simplified formula is less accurate than its second order version Eq. (D1) (panel (e) of Fig. 13 from Pannekoucke (2021) demonstrates explicitly in which sense), but offers more robustness during numerical simulations (as the same paper has shown). The second reason is the reduction of the anisotropy dynamics

640    to the transport process in the PKF formulation, which has been detailed in paragraphs 3.3.1-3.3.3. Compared to the FCST experiment, the assimilation of observations has had the effect of reducing the length-scales.

In both of these experiments, the PKF has shown itself able to reproduce the results of a large ensemble Kalman Filter. Again, these qualitative results of the PKF were obtained at a low numerical cost: the equivalent of 3 time integrations of Eq. (14) compared to 6400 for the EnKF.

645    The simplified LV-CTM has allowed for a multivariate formulation of the PKF, validated in numerical experiments. In the next section, a more complex chemical model is considered to further test the PKF possibilities.




## 4 A more realistic chemical model: GRS

### 4.1 Description of the GRS model

To further explore of the PKF possibilities, a more complex chemical model is considered: the generic reaction set (Azzi et al., 1992; Haussaire and Bocquet, 2016). GRS describes the dynamics of a reduced number of chemical species or *pseudo*-species. Hence, six species are considered and interact as

$$ROC + h\nu \overset{k_1(t)}{\to} RP + ROC \tag{32a}$$

$$RP + NO \overset{k_2}{\to} NO_2 \tag{32b}$$

$$NO_2 + h\nu \overset{k_3(t)}{\to} NO + O_3 \tag{32c}$$

$$NO + O_3 \overset{k_4}{\to} NO_2 \tag{32d}$$

$$RP + RP \overset{k_5}{\to} RP \tag{32e}$$

$$RP + NO_2 \overset{2 \cdot k_6}{\to} S(N)GN \tag{32f}$$

where ROC, RP and S(N)GN respectivily mean *Reactive Organic Compound*, *Radical Pool* et *Stable (Non-) Gaseous Nitrogen product*. In this chemical model, additional processes such as photolysis rate variation, ground deposits or atmospheric emissions of certain pollutants are represented.

The system of equations of the GRS-CTM writes:

$$\partial_t[ROC] = -\partial_x\left(u \cdot [ROC]\right) - \lambda[ROC] + E_{ROC} \tag{33a}$$

$$\partial_t[RP] = -\partial_x\left(u \cdot [RP]\right) - \lambda[RP] + k_1(t)[ROC] - [RP]\left(k2[NO] + 2k_6[NO_2] + k_5[RP]\right) \tag{33b}$$

$$\partial_t[NO] = -\partial_x\left(u \cdot [NO]\right) - \lambda[NO] + E_{NO} + k_3(t)[NO_2] - [NO]\left(k_2[RP] + k_4[O_3]\right) \tag{33c}$$

$$\partial_t[NO_2] = -\partial_x\left(u \cdot [NO_2]\right) - \lambda[NO_2] + E_{NO_2} + k_4[NO][O_3] + k_2[NO][RP] - [NO_2]\left(k_3(t) + 2k_6[RP]\right) \tag{33d}$$

$$\partial_t[O_3] = -\partial_x\left(u \cdot [O_3]\right) - \lambda[O_3] + k_3(t)[NO_2] - k_4[NO][O_3] \tag{33e}$$

$$\partial_t[S(N)GN] = -\partial_x\left(u \cdot [S(N)GN]\right) - \lambda[S(N)GN] + 2k_6[NO_2][RP] \tag{33f}$$

where for a specie $Z$: $[Z](t,x)$ denotes the concentration field ; and for $Z \in \{ROC, NO, NO_2\}$, $E_Z(x) = E_Z^0 \mu(x)$ denotes the stationary emission field modulated by the smooth ocean/land mask $\mu(x) \in [0,1]$ shown in Fig. 14(b), and of maximum emission $E_Z^0$ whose value is given in Table 1 (right column). The ground deposition is represented by terms in $\lambda$, with a magnitude of 2% per day. Kinetic parameters and chemical reaction rates are set as follows: since Eq. (33a) and Eq. (33c) depends on the solar radiation, $k_1$ and $k_3$ evolve in time to represent the diurnal cycle while they are related by $k_1 = 0.152k_3$ (Fig. 14(c)); the other rates are constant and given in Table 1.

In a new numerical experiment, the PKF forecasts will be compared with those of an EnKF (of size 1600). There is no observations assimilations in this simulation. A brief overview of the PKF formulation for Eq. (33) is now exposed.



| $k_3(t)$ | $0.624 \exp\left(-\frac{|(t\equiv 24)-12|^3}{100}\right)$ | $k_1(t)$ | $0.00152k_3(t)$ |
|---|---|---|---|
| $k_2$ | 12.3 | $E^0_{ROC}$ | 0.0235 |
| $k_4$ | 0.275 | $E^0_{NO}$ | 0.243 |
| $k_5$ | 10.2 | $E^0_{NO_2}$ | 0.027 |
| $k_6$ | 0.12 | $\lambda$ | $0.02\text{day}^{-1}$ |

In $k_3$ definition, the symbol $\equiv$ corresponds to the modulo operator. Emission rates in ppbCday$^{-1}$ for ROC or ppbday$^{-1}$ for NO$_x$, and the kinetic rates in ppb$^{-1}$min$^{-1}$, except for $k_3$ and $k_1$ in min$^{-1}$.

**Table 1.** GRS settings

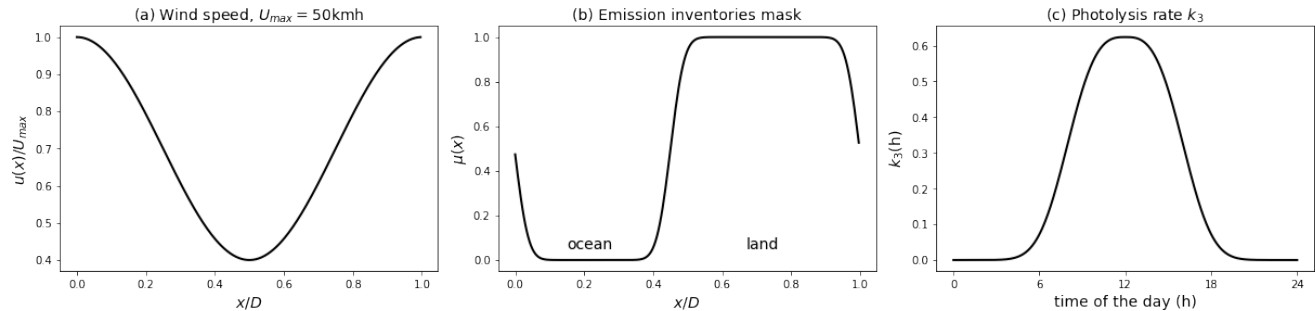

**Figure 14.** Settings of the GRS-CTM. (a): wind field, (b): emission field, (c): $k_3$ values along the day (min$^{-1}$).

## 4.2 The PKF for the GRS chemical transport model

Given the complexity of the set of equations Eq. (33), and the increased number of species in comparison to the LV-CTM Eq. (14), the equations of the PKF dynamics for the GRS-CTM are not presented in this article, but can be found in additional material[1]. In this context, the PKF system describes the dynamics of 33 pronostics parameters: 6 mean fields, 6 univariate vari-

ances fields, 6 anisotropy fields and 15 covariances fields (corresponding to the number of pairs of chemical species). In term of complexity, the PKF dynamics for the GRS-CTM is similar to the simplified LV-CTM: the transport part is the same, while the chemical part present the same kind of interactions between the chemical species. However, the stationary heterogeneous emissions, not present in LV-CTM, imply a forcing in the dynamics of the mean concentrations in GRS-CTM, but without effect on the uncertainty because the emissions are not stochastic here. Note that uncertainties on emission inventories can be

introduced in a PKF formulation *e.g.* as a source term in the variance dynamics and is related to the specification of boundary conditions in a PKF (Sabathier et al., 2022). Similarly to the LV-CTM, the dynamics of the anisotropy is closed by removing the terms due to the chemistry. Hence, latter, the dynamics of anisotropy in GRS-CTM is only due to the transport.

The settings and the results of the numerical experiment are now detailed.

---

[1]https://github.com/opannekoucke/pkf-multivariate





### 4.3  Numerical experiment: forecast

For the settings of this numerical experiments, the resolution of the grid has been reduced to $N_x = 241$ grid points, and the time step to $\Delta t = 10^{-4}$h to support the stiffness of the GRS equations. Some parameters remain unchanged: RK4 temporal scheme, finite differences to approximate spatial derivatives, choice of the wind field (Fig. 14a). The forecast starts at $t_0 = 00$h (midnight) and ends at $t = t_0 + 72$h.

Realistic heterogeneous initial concentration fields are constructed as follows. First, starting from zero concentrations, a

chemical equilibrium state is computed from a 4 weeks time integration of a 0D version of Eq. (33) where the transport has been switched off while the concentrations are forced by their respective emissions $E_{(\cdot)}^O$. The resulting concentrations are denoted by $[Z]_{0D}^{4\text{weeks}}$. Then, 1D concentration fields are constructed, set as constant and equal for each species to the final value of the 0D integration. The resulting homogeneous concentration fields are then independently perturbed so to produced heterogeneous concentration fields, more realistic than the homogeneous concentrations: for any species $Z$ of the 6 chemical

species, the resulting 1D perturbed field $[Z]^0(x) = [Z]_{0D}^{4\text{weeks}}(1 + 0.15 e(x))$ where $e = \mathbf{P}^{1/2}\zeta$ with $\mathbf{P}$ is an homogeneous Gaussian correlation version of Eq. (7) with variance 1 and constant length-scale $l_h = 12\Delta x$ ; and $\zeta$ is a sample of Gaussian random vector $\mathcal{N}(0, \mathbf{I}_{N_x})$. These perturbed 1D fields of concentrations correspond to the initial condition at $t_0 = 00h$ of the GRS-CTM simulations.

The initial condition for the PKF is set as follows. The mean state is given by the six 1D fields $[Z]^0(x)$. The multivariate

initial uncertainty is set as univariate (no cross-correlation) with a magnitude of $\sigma^0(Z) = 0.15[Z]_{0D}^{4\text{weeks}}$ for each one of the six species, with univariate homogeneous Gaussian correlation of length-scale $15\Delta x$ (60km), the length-scale are identical for all species.

For the validation, an ensemble of 1600 initial conditions has been populated, consistently from the PKF intial conditions, by adding univariate perturbations to the GRS-CTM initial condition. For each member $k$ of the ensemble and each field $Z$

that is to perturb, $[Z]_k^0(x) = [Z]^0(x) + 0.15[Z]_{0D}^{4\text{weeks}}e_k(x))$ where $e_k = \mathbf{P}^{1/2}\zeta_k$ with $\mathbf{P}$ is an homogeneous version of Eq. (7) with variance 1 and constant length-scale $l_h = 15\Delta x$ ; and $\zeta$ is a sample of Gaussian random vector $\mathcal{N}(0, \mathbf{I}_{N_x})$.

Fig. 15 shows the results of the experiments at two instants: at $t = 00h + 60h$, (slight transparency on the curves), and at $t = 00h + 66h$, (no transparency). These times corresponds to 12h00 and 18h00 of day 2. Each row of the six ones features the uncertainty for a specie $Z$ with respectively: the mean, the standard-deviation, the length-scale and a selection of four

cross-correlation functions with NO2, $\rho_Z^{NO2}$ ; that is the auto-correlation when $Z$ is $NO2$ itself. The choice of $NO2$ for the cross-correlation is arbitrary and other cross-correlations present the same behaviour (not shown). The statistics produced by the EnKF (resp. PKF) are represented using black dashed lines (resp. colored solid lines).

Regarding the behaviour of the error statistics, the impact of the chemistry appear: the chemical reactions led to non-zero cross-correlations visible on the right column (except panel (p) which corresponds to auto-correlations). Also, the roughness,

that was originally only present on the means, has been transferred (except for ROC) to the standard-deviations fields. The PKF equations for the dynamics of $V_{ROC}$ (not shown here) offer an explanation: $\partial_t V_{ROC}$ is only governed by decay and transport, and is not coupled with any means. Again, this illustrates the ability of the PKF to explain the physics of uncertainties. The





effect of the transport is also present: it produces spatial heterogeneities on the means (left column), standard-deviations (second column) and length-scales (third column).

Compared to the EnKF, the PKF offers a high quality forecast at a very low computational cost. The means (left column) are in perfect accordance in both methods. Slight differences can be observed regarding the standard-deviations fields (second column), but as established in 2.4.3, the EnKF diagnoses are biased by the numerical model error that is significant when using the low-resolution grid ($N_x = 241$ grid points in this simulation). The same argument applies to the length-scales (third column), although they may also be govern by some underlying chemical dynamics similar to those described for Fig. 12(f) in

section 3.5.2), not captured in the PKF formulation. Nevertheless, it does not prevent the PKF from estimating the auto and cross correlation functions (right column). The last column presents an important result: the cross-correlation functions estimations by the proxy are in great accordance with the EnKF. The proxy reproduces the variety of cross-correlation functions such as negative correlations, small amplitudes, asymmetric structures. Despite differences in length-scales estimations, the proxy shows itself robust and delivers satisfying modeled cross-correlation functions (at a qualitative level). Indirectly, it demonstrates

the capacity of the PKF to forecast all the cross-covariance fields.

     The present experiment validates the results obtained for the LV-CTM in the case of GRS-CTM: the PKF multivariate formulation results hold in this more complex chemical model. But it also makes appear some limitations of the multivariate formulation, for instance the rapidly growing number of parameters in the PKF dynamics.

     The results of this paper and the questions it raises for future developments of the PKF are discussed in the next section.

## 740    5    Discussion

In this work, we introduced a proxy for estimating the cross-covariances. However, some interrogations remains about its limitations: we did not questioned the positive definite character of the complete (auto and cross) covariance model, although it may not be an absolut necessity for the PKF applications. In the numerical experiments conducted here, it appeared that this proxy performed well at reconstructing the cross-correlation functions, but it has not been tested in other field of applications

such as geophysics and may be very specific to atmospheric chemistry. One could try to use this model for the shallow-water problem. Another questionable aspect is the extension of this model to the 2D or 3D case, which has to be verified.

     In the multivariate formulations (Lotka-Volterra and GRS) of the PKF dynamics, we limited the dynamics of the anisotropy to the advection process and unplugged the chemistry terms. This simplification lead to inaccurate forecasts of the anisotropy in the case where the chemical species have different length-scales.

Nonetheless, we were able to obtain high quality results (comparable to an EnKF of 6400 members) at a very low computational cost: putting aside the parallelisable property of an EnKF, the numerical cost of forecasting a PKF is equivalent to the one of the forecast of a dozen members in an EnKF, with high quality results. Plus, the PKF permits the understanding of the uncertainties dynamics: it offers equations that describes the time evolutions of variances, covariances and anisotropies. The impact of each process (advection, diffusion, chemistry) can be clearly identified in the dynamics of the error statistics,

allowing for a better comprehension of the overall problem. Difficult processes such as the injection of uncertainty in the sys-



tem by the emission inventory can be implemented easily in the PKF formulation just by acting on the variances dynamics. This readability is specific to the PKF and is not possible in other data assimilation methods. The PKF also reduce numerical costs by resuming the information contained in the forecast error covariance matrix of size $\mathcal{O}(N_x^2)$ to a few parameters of size $\mathcal{O}(N_x)$, reducing the need for high capacity storage. Finally, the PKF is less subject to numerical model error, when a slightly

diffusive or dispersive model might produces wrong estimations of the forecast error in an EnKF, as it is the error statistics that are directly being forecasted.

## 6    Conclusions

The goal of this work is to explore a multivariate formulation of the PKF for atmospheric chemistry needs.

To do so, a simplified chemical transport model is introduced in a 1D periodical domain. This simplified model allowed to
propose a proxy for the multivariate covariance to approximate the *cross*-covariances, which extends the univarite covariance model parameterized from the variance and the anisotropy tensor.

Then a multivariate PKF formulation has been proposed, which made appear a closure issue related to the chemical part, but not to the transport, and concerns the dynamics of the anisotropy. A detailed analysis of the effect of the chemistry on the dynamics of the anisotropy led to an analytical solution of the multivariate evolution of the uncertainty in a 1D harmonic
oscillator, which helps to understand the transfer of uncertainty from one species to another. Then the study of the relative contribution of the chemistry and of the transport to the trend of the anisotropy has been conducted, which appears to be mainly explained by the transport. Hence, a closed form has been considered by removing the terms related to the chemistry in the dynamics of the anisotropy.

Despite of this approximation, a validation test-bed using an ensemble method shown the that PKF dynamics is able to
predict the uncertainty dynamics for two chemical schemes based on Lotka-Volterra and GRS.

Moreover, several assimilation cycles have been conducted for the LV chemical scheme, showing the a multivariate PKF assimilation is possible, which is promising.

This work is a milestone in the development of a multivariate assimilation based on the PKF and applied to air quality, and feeds the reflection on the ongoing univariate implementation of the PKF approach in the operational transport model
MOCAGE at Meteo-France. In particular, the cost of the current multivariate PKF formulation scales as the square of number of chemical species which appears as a limitation, at least if all the chemical species are considered in the multivariate uncertainty prediction. Hence, it would be interesting to test a PKF formulation on a reduced chemical scheme of interest for the data assimilation.

Moreover, while this contributions focused on air quality, it contributes to improve our understanding of multivariate statis-
tics *e.g.* with the analytical solution of the 1D harmonic oscillator. It would be interesting to extend this multivariate PKF formulation to other geophysical applications *e.g.* the numerical weather prediction. Compared with air quality where the chemical reactions are point-wise, geophysical equations make appear local interactions that have to be study in view of the PKF approach *e.g.* the geostrophic balance in the barotropic model.





*Code and data availability.* The code developed and used to generate the experiments is available under https://github.com/opannekoucke/
pkf-multivariate

## Appendix A: Lotka-Volterra chemical model

We consider for chemical species $A, B, X$ and $Y$ governed by the chemical reactions:

$$X + A \quad \overset{k_1}{\to} \quad 2A, \tag{A1}$$

$$A + B \quad \overset{k_2}{\to} \quad 2B, \tag{A2}$$

$$B \quad \overset{k_3}{\to} \quad Y. \tag{A3}$$

The kinetic of the reaction, deduced from the mass action law for reaction rate writes:

$$\frac{d[A]}{dt} = k_1[X][A] - k_2[A][B] \tag{A4a}$$

$$\frac{d[B]}{dt} = k_2[A][B] - k_3[B] \tag{A4b}$$

where $[\cdot]$ denotes the concentration. When the concentrations of $X$ and $Y$ are constant (that is in excess), the system simplifies
as:

$$\frac{d[A]}{dt} = k_1[A] - k_2[A][B] \tag{A5a}$$

$$\frac{d[B]}{dt} = k_2[A][B] - k_3[B] \tag{A5b}$$

which is a Lotka-Volterra system.

## Appendix B: Dynamics of the error statistics for the Harmonic Oscillator

The harmonic oscillator equations writes:

$$\partial_t A = -kB, \tag{B1a}$$

$$\partial_t B = kA, \tag{B1b}$$

with $A = A(t, x)$ and $B = B(t, x)$ being functions of time and 1D space. As this problem is linear, the dynamic is identical
for the errors,

$$\partial_t \varepsilon_A = -k \varepsilon_B, \tag{B2a}$$

$$\partial_t \varepsilon_B = k \varepsilon_A. \tag{B2b}$$



Their analytical solution is given by:

$$\varepsilon_A(t,x) = \cos(kt)\varepsilon_A(t,0) - \sin(kt)\varepsilon_B(t,0), \tag{B3a}$$

$$\varepsilon_B(t,x) = \sin(kt)\varepsilon_A(t,0) + \cos(kt)\varepsilon_B(t,0). \tag{B3b}$$

We consider the case where the initial error are uncorrelated $V_{AB}^0 = \mathbb{E}\left[\varepsilon_A^0 \varepsilon_B^0\right] = 0$ and the initial variance and length-scale fields are homogeneous at initial time, *i.e.* $\partial_x V_A^0 = \partial_x V_B^0 = \partial_x g_A^0 = \partial_x g_B^0 = 0$.

From the analytical solution for the errors Eq. (B3), we deduce solutions for the error statistics.

$$V_A(t,x) = \mathbb{E}\left[\left(\varepsilon_A(t,x)\right)^2\right] \tag{B4a}$$

$$= \cos^2(kt)\mathbb{E}\left[\varepsilon_A^2\right](t,0) - 2\cos(kt)\sin(kt)\mathbb{E}\left[\varepsilon_A\varepsilon_B\right](t,0) + \sin^2(kt)\mathbb{E}\left[\varepsilon_B^2\right](t,0) \tag{B4b}$$

$$= \cos^2(kt)V_A^0 - 2\cos(kt)\sin(kt)\underbrace{V_{AB}^0}_{=0} + \sin^2(kt)V_B^0 \tag{B4c}$$

$$= \cos^2(kt)V_A^0 + \sin^2(kt)V_B^0 \tag{B4d}$$

$$\tag{B4e}$$

Following the same process, we deduce that $V_B(t,x) = \sin^2(kt)V_A^0 + \cos^2(kt)V_B^0$ and $V_{AB}(t,x) = \cos(kt)\sin(kt)(V_A^0 - V_0^B)$. We can now determine the dynamics of the metric tensors:

$$g_A(t,x) = \mathbb{E}\left[\left(\partial_x\left(\frac{\varepsilon_A}{\sqrt{V_A}}\right)\right)^2\right](t,x) \tag{B5a}$$

$$= \mathbb{E}\left[\left(\frac{\sqrt{V_A}\partial_x\varepsilon_A - \varepsilon_A\partial_x V_A}{V_A}\right)^2\right](t,x) \tag{B5b}$$

As we consider homogeneous fields, we have that $\partial_x V_A = 0$, simplifying the expression to

$$g_A(t,x) = \frac{1}{V_A}\mathbb{E}\left[(\partial_x\varepsilon_A)^2\right](t,x) \tag{B6a}$$

$$= \frac{1}{V_A(t,x)}\mathbb{E}\left[\cos^2(kt)\left(\partial_x\varepsilon_A^0\right)^2 - 2\cos(kt)\sin(kt)\partial_x\varepsilon_A^0\partial_x\varepsilon_B^0 + \sin^2(kt)\left(\partial_x\varepsilon_B^0\right)^2\right](x) \tag{B6b}$$

Again, under the condition of homogeneous initial fields, $\mathbb{E}\left[\left(\partial_x\varepsilon_A^0\right)^2\right]$ simplifies to $V_A^0 g_A^0$ and $\mathbb{E}\left[\left(\partial_x\varepsilon_B^0\right)^2\right] = V_B^0 g_B^0$. The independance of $\varepsilon_A^0$ and $\varepsilon_B^0$ also implies $\mathbb{E}\left[\partial_x\varepsilon_A^0\partial_x\varepsilon_B^0\right] = 0$. Finally, we obtain that:

$$g_A(t,x) = \frac{1}{V_A(t,x)}\left[\cos^2(kt)V_A^0 g_A^0 + \sin^2(kt)V_B^0 g_B^0\right]. \tag{B7}$$





We can also deduce an analytical solution for the term $\mathbb{E}\left[\partial_x \tilde{\varepsilon}_A \partial_x \tilde{\varepsilon}_B\right]$.

$$\mathbb{E}\left[\partial_x \tilde{\varepsilon}_A \partial_x \tilde{\varepsilon}_B\right](t,x) = \mathbb{E}\left[\left(\partial_x \frac{\varepsilon_A}{\sqrt{V_A}}\right)\partial_x\left(\frac{\varepsilon_B}{\sqrt{V_B}}\right)\right](t,x) \tag{B8a}$$

$$= \frac{1}{(\sqrt{V_A}\sqrt{V_B})(t,x)}\mathbb{E}\left[\partial_x \varepsilon_A \partial_x \varepsilon_B\right](t,x) \tag{B8b}$$

$$= \frac{1}{\sigma_A(t)\sigma_B(t)}\mathbb{E}\left[\cos(kt)\sin(kt)\left(\left(\partial_x \varepsilon_A^0\right)^2 - \left(\partial_x \varepsilon_B^0\right)^2\right) + \partial_x \varepsilon_A^0 \partial_x \varepsilon_B^0\left(\cos^2(kt) - \sin^2(kt)\right)\right](t,x) \tag{B8c}$$

$$= \frac{1}{\sigma_A(t)\sigma_B(t)}\left(\cos(kt)\sin(kt)\left(\underbrace{\mathbb{E}\left[\left(\partial_x \varepsilon_A^0\right)^2\right]}_{V_A^0 g_B^0} - \underbrace{\mathbb{E}\left[\left(\partial_x \varepsilon_B^0\right)^2\right]}_{V_B^0 g_B^0}\right) + \underbrace{\mathbb{E}\left[\partial_x \varepsilon_A^0 \partial_x \varepsilon_B^0\right]}_{=0}\left(\cos^2(kt) - \sin^2(kt)\right)\right)(t,x) \tag{B8d}$$

$$= \frac{\cos(kt)\sin(kt)}{(\sigma_A \sigma_B)(t,\mathbf{x})}\left(V_A^0 g_B^0 - V_B^0 g_B^0\right). \tag{B8e}$$

Note that we could have derived analytical solutions in the case of heterogeneous initial fields, but for the sake of simplicity we choose to consider only the homogeneous case. However, obtaining analytical solution when the initial error fields are correlated seems more difficult.

## Appendix C: Cross-covariance analysis formula demonstration

By introducing the true state and the error fields $\mathcal{X}^a = \mathcal{X}^t + \varepsilon^a$, $\mathcal{X}^f + \varepsilon^f$ and $\mathcal{Y}^o(\mathbf{x}_l) = \mathcal{X}^t(\mathbf{x}_l) + \varepsilon^o(\mathbf{x}_l)$, the analysis equation (8aa) becomes:

$$\varepsilon^a(\mathbf{x}) = \varepsilon^f(\mathbf{x}) + \sigma^f(\mathbf{x})\rho_{\mathbf{x}_l}^f(\mathbf{x})\frac{\sigma^f(\mathbf{x}_l)}{V^f(\mathbf{x}_l) + V^o(\mathbf{x}_l)}\left(\varepsilon^o(\mathbf{x}_l) - \varepsilon^f(\mathbf{x}_l)\right) \tag{C1}$$

which can be adapted to the multivariate case:

$$\varepsilon_{Z_1}^a(\mathbf{x}) = \varepsilon_{Z_1}^f(\mathbf{x}) + \sigma_{Z_1}^f(\mathbf{x})\rho_{Z_l,l}^{Z_1;f}(\mathbf{x})\frac{\sigma_{Z_l}^f(\mathbf{x}_l)}{V_{Z_l}^f(\mathbf{x}_l) + V_{Z_l}^o(\mathbf{x}_l)}\left(\varepsilon_{Z_l}^o(\mathbf{x}_l) - \varepsilon_{Z_l}^f(\mathbf{x}_l)\right) \tag{C2}$$

where $Z_l$ is the chemical species that is observed, $Z_1$ can be any chemical species, and $\rho_{Z_l,l}^{Z_1;f}(\mathbf{x}) = \mathbb{E}\left[\varepsilon_{Z_l}^f(\mathbf{x}_l)\varepsilon_{Z_1}^f(\mathbf{x})\right]/\left(\sigma_{Z_l}^f(\mathbf{x}_l)\sigma_{Z_1}^f(\mathbf{x})\right)$ is the forecast *cross*-correlation function between $Z_l$ and $Z_1$ at location $\mathbf{x}_l$. Writing the same equation for another chemical $Z_2$

$$\varepsilon_{Z_2}^a(\mathbf{x}) = \varepsilon_{Z_2}^f(\mathbf{x}) + \sigma_{Z_2}^f(\mathbf{x})\rho_{Z_l,l}^{Z_2;f}(\mathbf{x})\frac{\sigma_{Z_l}^f(\mathbf{x}_l)}{V_{Z_l}^f(\mathbf{x}_l) + V_{Z_l}^o(\mathbf{x}_l)}\left(\varepsilon_{Z_l}^o(\mathbf{x}_l) - \varepsilon_{Z_l}^f(\mathbf{x}_l)\right) \tag{C3}$$



and using the definition of the analysis error covariance field $V_{Z_1 Z_2}^a(\mathbf{x}) = \mathbb{E}\left[\varepsilon_{Z_1}^a(\mathbf{x})\varepsilon_{Z_2}^a(\mathbf{x})\right]$ leads to

$$V_{Z_1 Z_2}^a(\mathbf{x}) = \underbrace{\mathbb{E}\left[\varepsilon_{Z_1}^f(\mathbf{x})\varepsilon_{Z_2}^f(\mathbf{x})\right]}_{=V_{Z_1 Z_2}^f(\mathbf{x})} + \frac{\sigma_{Z_l}^f(\mathbf{x}_l)}{V_{Z_l}^f(\mathbf{x}_l) + V_{Z_l}^o(\mathbf{x}_l)}\mathbb{E}\left[\left(\sigma_{Z_2}^f(\mathbf{x})\rho_{Z_l,l}^{Z_2;f}(\mathbf{x})\varepsilon_{Z_1}^f(\mathbf{x}) + \sigma_{Z_1}^f(\mathbf{x})\rho_{Z_l,l}^{Z_1;f}(\mathbf{x})\varepsilon_{Z_2}^f(\mathbf{x})\right)\left(\varepsilon_{Z_l}^o(\mathbf{x}_l) - \varepsilon_{Z_l}^f(\mathbf{x}_l)\right)\right]$$
$$+ \frac{\left(\sigma_{Z_l}^f(\mathbf{x}_l)\right)^2}{\left(V_{Z_l}^f(\mathbf{x}_l) + V_{Z_l}^o(\mathbf{x}_l)\right)^2}\sigma_{Z_1}^f(\mathbf{x})\rho_{Z_l,l}^{Z_1;f}(\mathbf{x})\sigma_{Z_2}^f(\mathbf{x})\rho_{Z_l,l}^{Z_2;f}(\mathbf{x})\mathbb{E}\left[\left(\varepsilon_{Z_l}^o(\mathbf{x}_l) - \varepsilon_{Z_l}^f(\mathbf{x}_l)\right)^2\right] \tag{C4a}$$

Then, using the definition of the cross-correlation function $\mathbb{E}\left[\varepsilon_{Z_l}^f(\mathbf{x}_l)\varepsilon_{Z_1}^f(\mathbf{x})\right] = \sigma_{Z_l}^f(\mathbf{x}_l)\sigma_{Z_1}^f(\mathbf{x})\rho_{Z_l,l}^{Z_1;f}(\mathbf{x})$, the independance between the forecast and observation errors $\mathbb{E}\left[\varepsilon_{Z_l}^f(\mathbf{x}_l)\varepsilon_{Z_1}^o(\mathbf{x}_l)\right] = 0$, and the definitions of the error variance $V_{Z_l}^o(\mathbf{x}_l) = \mathbb{E}\left[\left(\varepsilon_{Z_l}^o(\mathbf{x}_l)\right)^2\right]$ and forecast error $V_{Z_l}^f(\mathbf{x}_l) = \mathbb{E}\left[\left(\varepsilon_{Z_l}^f(\mathbf{x}_l)\right)^2\right]$, we obtain that:

$$V_{Z_1 Z_2}^a(\mathbf{x}) = V_{Z_1 Z_2}^f(\mathbf{x}) - \frac{\sigma_{Z_l}^f(\mathbf{x}_l)}{V_{Z_l}^f(\mathbf{x}_l) + V_{Z_l}^o(\mathbf{x}_l)}\left(\sigma_{Z_2}^f(\mathbf{x})\rho_{Z_l,l}^{Z_2;f}(\mathbf{x})\sigma_{Z_l}^f(\mathbf{x}_l)\sigma_{Z_1}^f(\mathbf{x})\rho_{Z_l,l}^{Z_1;f}(\mathbf{x}) + \sigma_{Z_1}^f(\mathbf{x})\rho_{Z_l,l}^{Z_1;f}(\mathbf{x})\sigma_{Z_l}^f(\mathbf{x}_l)\sigma_{Z_2}^f(\mathbf{x})\rho_{Z_l,l}^{Z_2;f}(\mathbf{x})\right)$$
$$+ \frac{V_{Z_l}^f(\mathbf{x}_l)}{\left(V_{Z_l}^f(\mathbf{x}_l) + V_{Z_l}^o(\mathbf{x}_l)\right)^2}\sigma_{Z_1}^f(\mathbf{x})\rho_{Z_l,l}^{Z_1;f}(\mathbf{x})\sigma_{Z_2}^f(\mathbf{x})\rho_{Z_l,l}^{Z_2;f}(\mathbf{x})\left(V_{Z_l}^o(\mathbf{x}_l) + V_{Z_l}^f(\mathbf{x}_l)\right) \tag{C5a}$$

$$= V_{Z_1 Z_2}^f(\mathbf{x}) - \frac{V_{Z_l}^f(\mathbf{x}_l)}{V_{Z_l}^f(\mathbf{x}_l) + V_{Z_l}^o(\mathbf{x}_l)}2\left(\sigma_{Z_2}^f(\mathbf{x})\rho_{Z_l,l}^{Z_2;f}(\mathbf{x})\sigma_{Z_1}^f(\mathbf{x})\rho_{Z_l,l}^{Z_1;f}(\mathbf{x})\right)$$
$$+ \frac{V_{Z_l}^f(\mathbf{x}_l)}{V_{Z_l}^f(\mathbf{x}_l) + V_{Z_l}^o(\mathbf{x}_l)}\sigma_{Z_1}^f(\mathbf{x})\rho_{Z_l,l}^{Z_1;f}(\mathbf{x})\sigma_{Z_2}^f(\mathbf{x})\rho_{Z_l,l}^{Z_2;f}(\mathbf{x}) \tag{C5b}$$

$$= V_{Z_1 Z_2}^f(\mathbf{x}) - \left(\sigma_{Z_2}^f(\mathbf{x})\rho_{Z_l,l}^{Z_2;f}(\mathbf{x})\sigma_{Z_1}^f(\mathbf{x})\rho_{Z_l,l}^{Z_1;f}(\mathbf{x})\right)\frac{V_{Z_l}^f(\mathbf{x}_l)}{V_{Z_l}^f(\mathbf{x}_l) + V_{Z_l}^o(\mathbf{x}_l)}. \tag{C5c}$$

## Appendix D: Second order update formula for the anisotropy

In the alternative version PKFO2 of the PKF analysis algorithm PKFO1, the update equation for the metric tensor is (see
Pannekoucke (2021) for details)

$$\mathbf{g}^a(\mathbf{x}) = \frac{V^a(\mathbf{x})}{V^f(\mathbf{x})}\mathbf{s}^f(\mathbf{x}) - \frac{1}{V^f(\mathbf{x})V^a(\mathbf{x})}\left[\nabla V^f(\nabla V^f)\right](\mathbf{x})-$$
$$\frac{1}{V^a(\mathbf{x})}\frac{V^f(\mathbf{x}_l)}{V^f(\mathbf{x}_l) + V^o(\mathbf{x}_l)}\left[\nabla(\sigma^f\rho_{\mathbf{x}_l}^f)\left(\nabla(\sigma^f\rho_{\mathbf{x}_l}^f)\right)^{\mathrm{T}}\right](\mathbf{x}) - \frac{1}{4(V^a(\mathbf{x}))^2}\left[\nabla V^a(\nabla V^a)^{\mathrm{T}}\right](\mathbf{x}) \tag{D1}$$

Then, the analysis anisotropy $\mathbf{s}^a$ is obtained using $\mathbf{s}^a(\mathbf{x}) = (\mathbf{g}^a(\mathbf{x}))^{-1}$.

*Author contributions.* AP and OP explored the multivariate extension of the PKF and designed the experiments. A part of the work has been co-supervised with VG during the master internship of AP.



*Competing interests.* The authors declare that they have no conflict of interest.

*Acknowledgements.* We thank the Toulouse Paul Sabatier University, and the doctoral school SDU2E (Sciences de l'Univers, de l'Environnement et de l'Espace) which support Antoine Perrot's thesis. AP and OP would like to thank Richard Ménard for interesting discussions.

*Financial supporting* This work was supported by the French national programe LEFE/INSU grant "Multivariate Parametric Kalman Filter" (MPKF).





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



**Figure 15.** Multivariate forecasts for the GRS-CTM, PKF outputs (colored lines) and EnKF diagnoses (black dashed lines) for times $t = 00$h $+\{60, 66\}$h. As we consider a simulation that starts at midnight of day 0, $t = 00$h $+60$h corresponds to midday of day 2, and $t = 00$h $+66$h to 18h00 of day 2. From left to right, the columns correspond to the forecasts of: the mean concentration, the standard-deviation, the length-scales (normalized by $\Delta x$), and the correlation functions (*auto* and *cross*) with NO2 at locations $x = [0.1, 0.36, 0.63, 0.9]D$, for each of the six species (rows).