# Peer review of "Toward a multivariate formulation of the PKF assimilation: application to a simplified chemical transport model"

_EGUsphere, 2022_

## Referee Comment (RC2)

Review of

"Toward a multivariate formulation of the PKF assimilation: application to a simplified chemical transport model"

by A. Perrot, O. Pannekoucke, and V. Guidard.

General:

This is an interesting and original paper showing that a fairly sophisticated implementation of a Kalman filter can be done using parameters to define the local form of the evolving state error covariance instead of using a costly large ensemble. The method is exhibited using a simple chemistry transport model, with convincing success in this example. It suggests that an extension of the same technique might work also for a more complex model that includes the dynamical variables of the flow itself. At each time, and at each point in space, the covariance amongst the variables is approximately modeled by, for example, a multivariate Gaussian covariance, and it is the parameters prescribing this covariance which the PKF aims to evolve. The authors note that this becomes more burdensome in proportion to the square of the number of variables described by the local covariance, so it will be of great interest to see, in future, whether their technique remains competitive with ensemble methods in cases of greater dynamical complexity. But the paper is generally well written and convincingly argued, so I am happy to recommend that it be published after the correction of several very minor points listed by line numbers below.

Minor points:

L15 "..to the quality.."

L24: "..correct other concentrations and reduce .."

L38,39: "..needs the introduction of filtering.."

L40: "..to set correlations to zero.."

L53: What is Reynold decomposition – is there a reference you could add?

L115, L116, : "diagnoses" (plural)

L138: "..of a covariance model..", "..build a heterogeneous.."

L145: "non-obvious" (hyphenate?)

L148: "..approximation to reproduce.."

L198: "designed"

L268: replace "resumed" by "reduced"

L294: "scales"

L295: "..to attributing.."

L306: "dash-dotted"

Caption to Fig. 5: "clockwise" (not "clockwisely")

L394: "cross-correlations"

L410: "excluded"

L412: "overestimation of the true.."

L413:' replace "points of percentage" by "percentage points"

L126: "guarantee"

L441: "independent of anisotropy"

L452: "rehabilitated to quantify.."

L458: "removed to focus.."

L491: "there are no.."

L496: " ..to non-constant.."

L498: "stationary"

L499: "..anisotropy is equal.."

L500: "..stationarity.."

L510: ".. each of the terms.."

L523: ''..in the anisotropy.."

L525: " ..as they lead to.."

L575: "This leads to.."

L581: "..focuses on the forecast.."

L591: "independent"

L603: "A horizontal.."

L630: "statistics."

L647: Expand the acronym "GRS" in the main heading 4.

L675: "..observation assimilation in this.."

L680: "In terms.."

L682: "..presents"

L685: "..dynamics, and is.." (insert comma?)

L737: "..makes apparent some.."

L741: "..remain.."

L742: "we have not questioned.."

L743: "absolute"

L748: "This simplification led.."

L753: "describe.."

L774—777: It might be better to combine these lines into a single paragraph.

L787: "..studied.."

L840: "..we chose.."

L844: It is not clear what equation number you meant by "(8aa)"

L875: "..programme.."

---

## Author Comment (AC1)

Reply to Annika Vogel

First of all, we would like to thank Annika Vogel for her review and for giving us the opportunity to improve our paper. We added our acknowledgements to Annika Vogel for her evaluation of our contribution in the new manuscript.

Now, we organized the answer to the comments as follows. First, we list some changes afford to the manuscript then detail our answers to the questions raised by the referee.

**List of changes for the revision**

**Minor changes**

Errors in the label of some figures in the text have been corrected:
L491 Fig.8-(d) -> Fig. 8-(f)
L493 Fig.9-(d) -> Fig. 9-(f)

The figures have been re-rendered to improve their quality (size of legend, title,..). Because of the sampling noise inherent to the ensemble estimation, the values they show can have changed, without modifying the meaning or the robustness of the results. For instance, in Fig. 7, the averages where 22,8% for l_A=l_B while it is 23,1% now. As another example, the curves in Fig. 10 or 11 are not strictly the same as for the first version of the manuscript while we recognize the same patterns and conclusions.

**Differences between the two versions of the manuscript**

To facilitate the comparison between the two version of the manuscript, a companion version of the manuscript lists all the modifications where old (new) statements are in red (blue). But the line numbers will refer to the revised version of the manuscript (not to the companion version).

**Answer to the question of the referees**

We copied your commentary in italics below, we reply in normal blue font.

General comments:

*1) The submitted manuscript "Toward a multivariate formulation of the PKF assimilation: application to a simplified chemical transport model" by Perrot et aline contributes to the developments of a parametric Kalman Filter (PKF) in which the error statistics of a geophysical system are represented in form of a few parameters of the statistics. Specifically, the main contribution of this paper is the extension of previously published PKF formulations to multivariate problems in which cross-covariances between the individual prognostic fields occur. This is a very important step towards the application to real problems like complex chemical transport models (CTMs) and makes the manuscript highly valuable to the scientific community.*

Thank you for your motivating remarks.

*2) The quality of the research and the way of its presentation is good, but the manuscript is too long and overloaded. It appears that the authors aimed at putting too much content into one manuscript.*

*I would suggest division it into two (or even three) manuscripts, each specifically focusing on one aspect, eg:*

- *General aspects of multivariate PKF, including theoretical validation and limits (until end of Sect.2)*
- *Application to simplified chemistry, including proxy for cross-correlation function, contribution of individual terms and closure (mainly Sect.3 and maybe Sect.4)*
- *Maybe: application to more complex chemistry (Sect.4)*

Thanks to the referee comments, we improved the quality of the manuscript while preferring to keep it as a whole. From our estimation, the resulting manuscript in two-column is less than 30 pages (28 pages when latex is compiled using the npg template without the "manuscript" option).

*3) Besides that, some parts presented in the manuscript could be shortened significantly by referring to previous literature and focusing on the new aspects of this work (especially in Sec.2, see specific comments). The manuscript also contains a number of inconsistencies in the notation and some grammar/wording mistakes (see technical corrections) which makes it sometimes difficult for the reader to follow the details.*

This has been corrected point by point following your recommendations. Thank you.

*Specific comments:*

1) *Different quantities are used for error statistics in different equations and plots. Eg. in Sec. 2.3, variance V and metric tensor g are used in Equation (8) whereas standard deviation and length scale are used in Fig.2. Additionally, the aspect tensor s is used in Equation (21) whereas the metric tensor is used in Equation (24). It would increase the readability significantly if the authors would stick to one quantity thought the manuscript were possible, or at least within the same evaluation (eg. to increase the consistency between Equation (21),(24) and (25) in Sec.3.3). In addition, the correlation length scale (eg. in Fig.2 and LINE 247) was not defined and its connection to the other correlation quantities (metric and aspect tensor) remains unclear.*

This has been modified along the manuscript which is now focused on the aspect tensor (see e.g. equations 8, 25), that is directly connected to the length-scale in 1D domain. Moreover, the length-scale is now defined after Eq.(6).

*2) Section 2: The main new scientific contribution of this section appears to be the comparison to the enKF with two different spatial resolutions showing that the PKF is able to produce reasonable statistics already at coarse resolutions. Thus, the section can be significantly shortened by focusing on the new aspects and the most essential information required for those:*

> *2.1) The main content of Sec.2.2 and 2.3 was previously formulated in Pannekoucke2021. I suggest shortening the introduction of the PKF univariate equations by referring to this paper and just providing equations which are essential for the new aspect (i.e. numerical limits in Sec.2.4.3)*

Since section 2 is devoted to recalling the context of the PKF and the definition we need for the understanding of the manuscript, we find it difficult to delete sections 2.2 and 2.3, and prefer to keep them, for self-consistency of the manuscript.

However, in the new version of the manuscript, we simplified the introduction of the PKF dynamics for the advection and suppressed the previously dedicated section 2.4.1. Now Sec. 2.4.1 refers to the numerical validation with the EnKF.

*2.2) Section 2.4.2 seems to contain mainly results were already demonstrated in previous publications. Eg. LINE 257: The ability of the PKF to produce high quality univariate forecasts of error statistics was already shown in Pannekoucke2016,2018,2021. Although the description presents the main aspects and advantages of the PKF in a well-formulated way, it appears to be more suitable for a review or textbook-like article than as part of a scientific manuscript. Here, the description can be shortened significantly, focusing on new aspects.*

Compared with previous works, one of the contribution of  Sec. 2.4.2 (now the section 2.4.1) is to validate the correlation functions provided by the PKF, compared by the one estimated from the EnKF.

This now clearly appears in the manuscript : « Compared to previous studies that focused only on the comparison of variance and anisotropy error statistics, here we have shown the ability to reproduce complex heterogeneous correlation functions using the PKF formulation in 1D domain. » Line 316-318

This section introduces the numerical framework that is used throughout the manuscript as well as the estimation of the variance and anisotropy of an ensemble. Therefore, removing this description here would have required introducing it later in the manuscript without saving space.

Moreover, the section presents the behavior of the error statistics is case where the dynamics is a conservative equation (that was not the case of previous published works). This behavior is complex since it makes appear variations of the magnitude of the mean and of the error variance because of the heterogeneity of the wind and of the conservation. We chose this configuration, with an heterogeneous wind, because it offers a simple but rich framework for the exploration of CTM uncertainty dynamics. The description helps to interpret the results that will come in the multivariate setting, and for which the complexity increases because of the coupling between chemical species. For these reasons we consider that the section is important for the reader that is not used to the PKF, and for the self-consistency of the manuscript, while we agree that scientists with a strong background in PKF will find this part less interesting for themselves – however this is still the introduction of the PKF.

*2.3) The main conclusion of Sec.2.4.3 is that the PKF is able to produce reasonable error statistics also for coarse spatial resolutions (LINE 302f, LINE 307f). No validation with true statistics is available for this setup and the conclusion is based on comparison of comparison of the low resolution PKF with high- and low resolution enKF. In this setup, the agreement between low resolution PKF and high resolution enKF does not necessarily proof good performance because both resolutions could be insufficient. The easiest way to indicate convergence of the methods, and thus the accuracy of the solution, would be to make sure that the solution of the PKF remains the same for high resolution simulations. This would indicate the convergence of the PKF to a solution which is well approximated by the low resolution simulation. Ideally, the convergence of the enKF to the same solution could also be indicated by performing even more high resolution simulations. But given the computational efforts, it might be sufficient to verify that the high resolution PKF solution agrees with the low resolution PKF and the high resolution enKF solutions.*

We computed the high resolution of the PKF, and compared it to the low resolution results. We observed that there is no difference at eye from the low and the high resolution simulation, at a

quantitative level, the relative error of the low vs. high solution (the high resolution being considered as the reference here) is less than 1 % in relative error (computed from the L2 norm).

This is now mentioned in the manuscript: "A PKF at high resolution has been computed (not shown here) and has been found equivalent to the PKF computed at low resolution, with a relative error at the end of the forecast window lower than 0.2% for the mean, 0.3% for the standard-deviation, and 0.05% for the length-scale ; where the relative error of fields has been computed as $\|PKF_{LR} - PKF_{HR}\|/\|PKF_{HR}\|$, with $\|\cdot\|$ the L2 norm." in L309-312

> 2.4) *Given the length of Section 2.4, I would suggest making it a new main Section (->Sec.3, if not significantly shortened according to other comments above. This comment is also related to the main general comment of dividing the manuscript.) In addition, the title might be misleading, as it appears to include only the advection process as part of the forecast step and not the full PKF with forecast and analysis step. Maybe something like: "PKF for advection equation of passive tracer" would be a more appropriate section title.*

We have retained but reduced Sec. 2.4 and modified the title as suggested.

3) *Section 3: This section is much too long. Following the general comments (above), I would suggest taking this section as a paper on its own, which would be of appropriate length (17 pages +introduction, conclusions etc). In any way, Sec.3.3+3.4 as well as Sec.3.5 could be separate sections each, i.e. dividing Sec.3 into three sections, eg: 1) Sec.3.1+3.2, 2) Sec.3.3+3.4, 3) Sec.3.5.*

We chose to refactor section 3 into 3 subsections, by merging *1) Sec.3.1+3.2, 2) Sec.3.3+3.4, 3) Sec.3.5, as proposed.*

4) *In Sec.3.3.2 and Sec.3.5.2 it remains unclear how much the results can be generalized are subject to the specific setup of the experiments. For example, it would be interesting to see if the advection terms remain dominant under different conditions like weaker wind or accelerated chemistry. This becomes also important for verifying the neglection of chemistry part in the anisotropy for the GRC CTM in Sec.4.2.*

We agree on the importance of these points, and we have added them in the conclusion of the report to guide the remaining investigations on the subject, which we have not conducted here.

5) *Section 5: The discussion section only partly includes an actual discussion.*
5.1) *The first paragraph of the discussion refers only to a specific part of the study, not to the complete work. I suggest moving it to the referring Sec.3.2.2 (maybe as new subsection if necessary).*

The beginning of the paragraph was mentioned in sec. 3.2.2; while the end has been moved in the conclusion.

5.2) *The rest of the section is a conclusion rather than a discussion and should be moved to the conclusion section.*

It has been moved in the conclusion that has been rephrased so to include specific the points that was not already present e.g. the extension to 2D/3D domains.

6) *Appendix D: This appendix provides no added value for this paper because the equation is not used and can be found in Pannekoucke2021 for reference. Remove appendix and refer only to the equation in Pannekoucke2021 paper in Sect. 3.5.2 (LINE 638)*

We removed the appendix and referred to Pannekoucke2021 paper for the equation.

Technical corrections:

General technical corrections (at multiple locations in the manuscript):

1. *Suggest replacing "modelized" by "modelled" and "modelizes" by eg "models" or similar (LINE 2, LINE 239)*
   We preferred to let the version in 'z'.

2. *Inconsistent typing w.r.t. hyphen. The manuscript composition guidelines suggest the form without hyphen (eg. "auto correlations" and "cross correlations" in LINE 21, "forecast error" in LINE 117, "length scale" in LINE 247,…)*

   Thank you very much. However, we did not find this recommandation in https://www.nonlinear-processes-in-geophysics.net/submission.html#english, where it is seems to only concern : adverb ending in –ly, Latin phrases or abbreviated units. Will check this with the proof-editing services after acceptance of the manuscript.

3. *Replace "validated from" eg with "validated by" or "validated w.r.t."* (LINE 66, LINE 188)
   This has been corrected, thank you.

4. *The formulation "so to" should be replaced eg by "to" or "in order to" (LINE 74, LINE 148, LINE 356, LINE 436, LINE 449, LINE 452, LINE 458, LINE 698)*
   This has been corrected, thank you.

5. *There are inconsistent indications of locations in different variables. Eg. subscript like $V\_x$ vs. in brackets like $g(x)$ in Equation (7) whereas $V(x)$ was used in LINE 118 (same in LINE 191 vs LINE 195, and Equation (20)). Suggest sticking to common indication (either as subscript or in brackets) for all variables, or to point out specifically the difference between the two types of variables (eg. discrete vs continuous?)*
   This has been corrected, thank you.

6. *Suggest avoiding double use of brackets, if possible (eg. $P(V,s)(x,y)$ in Equation (7) and $rho(g^f)\ (x\_L,x)$ in LINE 169)*
   We preferred to keep the notation, thank you.

7. *Figure captions need to be extended in order to describe the figure sufficiently, that it can be understood independent of the text.*
   - *1: missing information that this is a predefined and stationary wind field and description of axes incline normalization.*
   - *6: missing description of individual lines, unclear: cross-correlations to which species at which location x=0.5?*
   - *14: analog to Fig.1 for wind field and emissions inventory mask*
   This has been corrected. Thank you.

8. (Related to point 7): Label sizes need to be increased, especially for axes and legends (all figures).
   This has been modified. Thank you.

9. *(Related to point 7): Purely technical figure descriptions should be removed from the text, and put in the figure caption instead (eg. LINE 306 "cyan dash-dotted lines", LINE 379, LINE 398, …, including LINE 623-624, LINE 713-715, LINE 716f).*
   This has been corrected. Thank you.

10. Often, subsections are finished by a sentence introducing the following subsection. This hinders the flow of reading. I would suggest removing these sentences and, were necessary, motivating/introducing the new subsection in its beginning (LINE 242-243, LINE 275f, LINE 309f, LINE 446f, LINE 509f, LINE 552, LINE 563, LINE 577, LINE 645f, LINE 675, LINE 688, LINE 739)
*This has been modified. Thank you.*

11. When referring to figures in the text, the authors often only indicate the subplot panel and not the actual Figure number, eg "(panel a)". Although the Figure number was mentioned before in the text, it is standard to refer to subfigures by eg "(Fig.1a)" (compare also manuscript preparation guidelines) which also makes it easier for the reader to follow the argumentation. (LINE 245, LINE 247, and many more…)
This has been modified along the manuscript

12. The word "paragraph" should be replaced by "Sect." according to the manuscript preparation guidelines (LINE 257, LINE 439, LINE 449, LINE 452, LINE 457, LINE 529, LINE 534)
This has been corrected, thank you.

13. The naming convention of (cross)(co)variances is sometimes confusion. I would suggest using different names or at least clearly highlighting the differences, eg V_A(x) variances (between same species, same location =diagonal elements of P_A), P_A(x,y) (auto-)covariances (same species, different locations), V_AB(x) cross-variances (different species, same location =diagonal elements of P_AB), P_AB(x,y) cross-covariance (different species, different locations), or similar. Eg. in LINE 351, V_AB(x) is named cross-covariance without mentioning that is refers to the same location, and in LINE 440, V_AB is named covariance although is refers to different species.
We checked along the manuscript and do the appropriate modifications using the following terminology: V_A == Variance, V_AB == single-point cross-variance, P_AB == two-point cross-covariance. Hence, now, V_AB is a function of $x$ alone and V_AB(x,y) is replaced by P_AB(x,y) everywhere.

14. The two species indicators are sometimes written as lower and sometimes as upper index, eg. V_AB vs V^AB in LINE 351 vs LINE 353. Again if these are different quantitates, it should be clarified in the text, if not, please check the whole document for a consistent notation.
This has been corrected. Thank you.

15. Replace "independant" by "independent" and "independance" by "independence" (LINE 441, LINE 591, LINE 831, LINE 855)
This is now corrected, thank you.

Content-related technical corrections:

*1) LINE 137: Add reference to Weaver&Courtier2001: https://doi.org/10.1002/qj.49712757518*

Weaver and Courtier 2001 introduced the use of the diffusion equation, but the setting of the diffusion coefficient that is the purpose of this line is not addressed in WC01. So we added the reference to WC01 before we discussed the setting of the diffusion tensors from the anisotropy.

*2) Equation (8)+LINE 170-171: add reference to derivation of equations, Pannekoucke2021:* https://doi.org/10.1080/16000870.2021.1926660
*The reference has been added.*

*3) LINE 3: suggest adding "has previously been" to make clear that this is not part of the present work*
*This has been modified.*

*4) LINE 18-19: state-of-the-art CTMs are much more complex than transport and chemical reactions (eg. diffusion, emissions, deposition, interaction with clouds, …). I suggest reformulating the sentence to make clear that transport and chemistry are some of multiple processes, which are however considered dominant for most applications.*
The sentence has been completed to mention the complexity of a CTM with an explicit mention to the diffusion, emissions, desposition and interaction with clouds.

*5) LINE 39ff: The sentence beginning with "In air quality,…" makes a jump in the chronology of the text. Based on the previous sentence, it is not clear to what the word "them" is referring. Please reformulate.*
This has been modified, as follows: "In air quality, it may be preferable to set  to zero the ensemble estimation of the multivariate correlation, so to avoid polluting the resulting analysis state". (L47-48)

*6) LINE 44: The context of the word "but" is not clear in this sentence. Suggest replacing by something like "…a numerical model, which are often computed in parallel at lower resolution.", if this fits the statement.*
This has been modified as proposed. Thank you.

*7) LINE 85: Sparse observations and modelling errors are not the only reasons for the unknown true state. I would suggest reformulating, for example adding that all available information (observations and model forecasts) contain errors.*
This as been rephrased as "Because of the spatio-temporal sparsity of observations, as well as modeling, prediction and measurement errors, the exact actual state at a time $t=t\_q$, $\X^t\_q$, is unknown." (L97-98)

*8) LINE 87: The formulation "estimation of X^t_q coming from the past" is unclear and unspecific. Does it refer to the forecast state?*
Yes, it refers to the forecast state, this has been modified. *Thank you*

9) LINE 94: The Kalman Filer also assumes independent errors between observations and forecast.
*This has been modified. Thank you.*

*10) LINE 118: The definition of eps^f (x) is inconsistent with Sec.2.1 were it was a discrete vector. If a continuous formulation is used here, this should be introduced accordingly. If not, the transposed notation should also be used here.*
The use of the continuous / discrete versions of the quantities is now better introduced in the beginning of Section 2.1 with : « Thereafter, $\X$ can be seen either as a collection of continuous fields with dynamics given by Eq.(1) or a discrete vector of dynamics the discretized version of Eq. (1). ». Hence, the definition of the variance field as it is specified by using eps^f(x) is now well defined.

*11) Equation (4)+Equation (7): The norm is not defined. Suggest a short note on the norm and the meaning of its lower index, maybe with reference to literature if needed.*

The norm is now defined as follows: "where $\| \cdot \|_g$ stands for the Euclidean norm associated with a metric g and defined from $\|x\|_{2\,g} = x_T\,gx$." L 137

*12) LINE 122+130: Suggest adding "at each grid location x" to make clear that g is a tensor at each location.*

We added that "There is one local metric tensor at each grid location $\x$" L138-139

*13) Equation (5): A note on the meaning of x_i and x_j (indication of derivatives into two directions?) is missing.*

We explained that this notation refers to the coordinate functions as follows: "where $\x_i$'s are the coordinate functions associated with the coordinate system $\x$." L142-143

*14) LINE 137-139: This sentence is too long and repetitive. Suggest reformulating, eg. something like: "This covariance model is used in variation DA to generate heterogeneous covariances were correlation functions vary between grid points."*

*We replaced the sentence by your proposition. Thank you. (see L154-155)*

*15) LINE 147: formulation remains unclear: "leads to sum up the statistical content into a set of parameters". Reformulate for clarification.*

We rephrased as follows: "Hence, approximating a covariance matrix, as the forecast-error covariance at a given time, by a covariance model is reduced to the knowledge of a set of parameters". L162-163

*16) LINE 165+Equation (8): Although it is referred to a single observation here, I would suggest adding the indication of the observation location for the observation variance "V^o(x_L)" to be consistent to the other quantities at observation location (eg. "V^f(x_L)").*

This has been added. Thank you.

*17) LINE 169: Suggest adding "is the correlation function between the observation location and each model gridpoint x".*

This has been modified. Thank you

*18) LINE 186: The formulation "to predict the uncertainty dynamics, the latter being estimated from an ensemble method introduced to provide a reference." is quite complicated and long. Suggest reformulation, eg something like "to predict the uncertainty dynamics compared to a reference ensemble estimation (enKF)."*

This has been modified. Thank you.

*19) LINE 291-292: The reasoning of the statement "the dispersive term influences both, the variance and the length scale" remains unclear because Equation (13) only refers to the mean state. Maybe is could be described a bit more how the authors come to this statement.*

We detailed as follows: "The reason is that Eq.(13) being linear, it also governs the error field, as the one predicted by the EnKF, and for which the magnitude of the dispersion is more intense as the error correlation length-scale is short. In this simulation, the scale of the mean state is large (of the order of $D$), so the effect of the dispersion is much less intense than for the errors whose typical scale of oscillations is $l_h$ (of order $D/10$). This justifies why the dispersion does not affect the prediction of the mean state -- the estimation for the means coinciding for the two methods on Fig2-(a) --, while it acts on the EnKF predictions of the variance and of the length-scale, related to the error dynamics."L295-301

*20) LINE 296ff: The statement of the sentence starting with "Therefore, as with the PKF..." remains unclear. How does the fact that error statistics are forecasts equivalently to state forecasts in the PKF relate to the sensitivity of the enKF to model errors? This seems to be two different aspects. Please reformulate or clarify.*

We rephrased this sentence and moved it at the end of the discussion mentioned added it the last point 19): "In this simulation, the PKF is not influenced by the dispersion because the spatial scale of the variance and of the length-scale is large (order of $D$). This points out the sensitivity of the EnKF to numerical model error." L301-302

*21) Equation (16),(17): The notation is confusion w.r.t. P_AB(x,y) and V_AB(x,y). Both are defined in the text as "two-point cross covariance." If the same quantity is meant, the same variable should be used, if not, the different should be made clearer.*

P_AB is the cross-covariance matrix. This is now mentioned in the manuscript following the modification made from answer to your General technical corrections 13).

*22) LINE 398: Generation of "ensemble estimated cross-correlation" unclear. Is the cross-correlation model applied to each ensemble realization?*

Here we mean that the computation of the proxy r_AB of the cross-correlation rho_AB is computed from Eq.(20) by using the estimation of the statistics needed for the relation Eq.(20). This is made more clear now:

"To assess the skill of the proxy, Fig. 6 shows the functions $r_{AB}(x_l,\cdot)$ (computed from Eq.(20) with the ensemble-estimated parameters $\widehat{\mathcal{P}}(t)=(\widehat{V_A}, \widehat{V_B}, \widehat{V_{AB}}, \widehat{s_A}, \widehat{s_B})(t)$), compared with the ensemble estimated cross-correlation $\rho_{AB}(x_l,\cdot)$." L405-406

The proxy for the cross-correlation is not used to sample the ensemble since the initial errors are decorrelated following Eq.(18).

*23) LINE 470,Fig.8: How is the open term calculated? Eg. from the truth or the ensemble mean?*

It is calculated from the ensemble mean, as it is done for the other statistics (e.g. the variance Eq. (11)). The detail of the computation of the open term has been introduced in Eq.(23)

*24) LINE 482: For clarification, I would suggest noting that these are analytical expressions, eg "evolution of the statistics analytically" or "an analytical evolution of the statistics"*

This has been modified. Thank you.

*25) Equation (24f): Inconsistent notation. Up to now the overbar was used to indicate the expectation, whereas the E[ . ] notation was used here. Please stick to one notation for the entire manuscript.*

This has been modified. Thank you.

*26) Equation (26),(27),Fig.10: The different normalization of the weights by term in Equation (26) and by process in Equation (27) might lead to confusion when looking at Fig.10. For example, the relative contribution of the two advection terms seams to be only slightly higher than the chemistry terms in Fig.10a (~55% vs 45%), but advection is highly dominant in Fig.10c (~80% vs 20%). I would suggest noticing the different normalization in the text or maybe even consider using a common normalization for both, if that makes sense.*

The use of a different normalization is now indicated: "Note that the normalization is different between Eq.(27) and Eq.(28)." L523-524

*27) LINE 546ff: The discussion of different approaches for closure is spited into Sec.3.3.2 (LINE 546-551) and Sec.3.3.3. I would suggest moving LINE 546-551 into Sec.3.3.3 and renaming this*

*section eg "closure of the PKF dynamics".*
We moved as proposed and renamed the section 3.3.3. Thank you.

*28) LINE 568: The formulation beginning with "the subscript l must be …" is slightly confusing. Suggest reformulation for clarification. x: element w.r.t. any species at any location, x_L: observation of a species Z_L at observation location?*
This is now modified. Thank you.
"To apply the formulas Eq.(8) in multivariate contexts, the $x_l$ must refer to the observation of a species $Z_l$ at observation location, while $x$ refers to any species at any location." L569-570

*29) LINE 573: Is there a reason for having the second species index Z_1 as superscript in rho whereas is it written as subscript for all other variables? I would suggest putting it as subscript for consistency reasons.*
This has been modified and corrected where ever the cross-correlation appeared (*e.g.* in appendix C).

*30) Algorithm 1: Inconsistent syntax for loops. Eg. line 1 should be "for each observation l do" to be consistent with the other (or the other way around)*
This has been modified. Thank you.

*31) LINE 595: Is each observation sampled independently for each time or are they temporally correlated? In addition to describing in the text, it might be useful to add the time index to make this clear in the equation.*
The time index has been added. Thank you.

*32) LINE 623: It remains unclear if only one assimilation of the four observations is performed at time t=t_max or if several assimilation cycles are performed during the simulation. Please add this information.*
There are five assimilation cycles during the simulation. The total simulation window Is [0,t_max] where t_max = 5tau_adv/3, and an assimilation is performed after each time integration of tau_adv/3 .Hence, at t=t_max, five assimilations have been performed.
While it has been indicated in the beginning of section 3.5.1, we recalled the detail of the DA experiment here, so the sentence has been rephrased as:
"The outcome of the DA experiment Fig. 13 is now exposed, where five assimilation cycles are done over the period $[0, t_{max}]$ (one assimilation after each $\tau_{adv}/3$ time integration, with $t_{max}=5\tau_{adv}/3$)" (see L564-565)

*33) LINE 686f: Was any investigation done if the dominating impact of dynamics vs chemistry also holds for the GRS-CTM? (compare specific comment 5a)*
Since the PKF was able to reproduce the ensemble estimation, we did not investigate the dominating impact of the dynamics vs chemistry that has been detailed for the LV-CTM case. However, following the answer to point 4) of your specific comment, it has been added in the conclusion as an interesting experiment to consider in real CTM.

*34) LINE 722: Context, the description in the previous sentences appears to describe the general behavior. A conclusion of the performance of the PKF requires mentioning the fact that the PKF is able to reproduce all features described above. It also remains unclear if this statement only refer to chemistry or also to transport. I would suggest adding a related sentence and moving into the next paragraph (eg LINE 725ff), if that fits the content, and reformulating accordingly.*
The discussion about the particular form of the dynamics of ROC appeared as a digrassion at this step while it is important to highlight the benefice of the PKF. To simplified, we chose to move the explanation of the V_ROC in a note ("Note that the specific behavior of the ROC .." LINE 739-

743). Now the paragraph better addresses the respective contribution of the chemistry and of the transport. The conclusion of the performance have been removed toward the conclusion section .

*35) LINE 728,Fig.15: It looks like the PKF produces the same length-scales for all species. It this is the case, it would be interesting to mention and explain.*
Yes, the length-scale fields are the same for all chemical species because they follow the same dynamics (only the transport is considered for the length-scale evolution, not the chemistry) and start from the same initial homogeneous length-scale value (here l_h=12 Delta x).
This is made clearer with the sentence "Since the PKF formulation considered here is closed by removing the contribution of the chemistry on the length-scale dynamics (following the simplification discussed in Sec.3.2.4), the length-scale dynamics is the same for all species. " LINE 730-732

*36) LINE 734: Suggest replacing the word "Indirectly" by a more specific formulation. Does this refer to the other cross-correlations, which are also well captured by the PKF but not shown here?*
The sentence has been rephrased in two ones: "This has been observed for other cross-correlation functions (not shown here). It demonstrates the capacity of the PKF to forecast the cross-covariance fields." L 737-738

*37) LINE 759ff: I don't see a connection of this statement to the content of the paper. While not being wrong, it seems to appear without any explanation. Therefore, I would suggest removing it here.*
It has been removed.

*38) LINE 762: Sec.6 also includes a short summary (first part of this section). Therefore, I would rename the section "Summary and conclusions"*
The title of the section is a default standard in the NPG template used.

*39) LINE 767-773: The paragraph deals with the first experiment with simplified chemistry (eg the evaluation of transport vs chemistry). This needs to be mentioned in order to put the conclusions into context. I would suggest reformulating the sentence in LINE 774-777 accordingly and moving it to the beginning of this paragraph.*
The paragraph is not clear and we rephrased the conclusion for the introduction of the three experiments and their results.

*40) LINE 779: Formulation "feeds the reflection on" is unclear. Does it mean that this work is an important step in extending the univariate implementation to complex operational CTMs like MOCAGE? Reformulate.*
This has been rephrased as: "and is an important step in extending the univariate PKF implementation to complex operational CTMs like the operational transport model MOCAGE at Meteo-France" L 778-780

*41) LINE 780: Sentence starting with "In particular" seems to refer to a different aspect, which is actually a drawback of the method. This should be made more clear in this sentence.*
This has been rephrased as: "The work also highlight a drawback of the PKF: the cost of the current multivariate PKF formulation scales as the square of number of chemical species which appears as a limitation, at least if all the chemical species are considered in the multivariate uncertainty prediction. Hence, it would be interesting to test a PKF formulation on a reduced chemical scheme of interest for the data assimilation." L 780-783

*42) Equation (B3),(B4): The expectations of eps_A^2, eps_B^2 and eps_A\*eps_B denote the boundary condition at time t for x=0. Instead, it should be the initial condition V_A^0 = E[eps_A^2] (0,x) were eps_A(0,x) = eps_A^0, right?*
Yes, sorry for the typos. It has been corrected. Moreover we added the definition of the upper-script ^0. Thank you.

*43) Equation (B5b): If I'm not mistaken, there is a square root missing for V_A in the second term of the numerator: "- eps_A d_x sqrt(V_A)"*
The typos has been corrected. Thank you.

*44) LINE 830: The assumption of homogeneous initial fields remains unclear here. Doesn't E[(d_x eps_A^0)^2] = V_A^0 g_A^0 follow directly from Equation (B6a) evaluated at t=0 ??*
This has been rephrased as: "Then, at t=0, E[..]" since the assumption of homogeneity has been introduced at this step. (see L 830)

*45) Equation (B8b): The homogenous assumption is used in this step.*

The assumption has been added. Thank you.

*Individual purely technical corrections:*

1. LINE 14: put reference in brackets "(Kalman, 1960)" *-- Done*
2. LINE 18: put reference in brackets: "(Josse et aline , 2004)" *-- Done*
3. LINE 20: suggest replacing "features" eg. by "contains" or "includes" (if this fits the statement) – *Done, replaced by contains.*
4. LINE 24: remove final "s" from "others" *-- Done*
5. LINE 35: "On the other hand" should only be used when following "On the one hand". Suggest replacing eg. by "At the same time" or "But", "However", … *-- Done*
6. LINE 38: suggest replacing "needs to introduce" eg. by "requires the introduction of" *-- Done*
7. LINE 54: replace the word "leveraged". Meaning unclear. *-- Done: replaced by "is based on"*
8. LINE 55: "an other" -> "another" *-- Done*
9. LINE 71: grammar, replace "before to conclude" with eg. "before concluding remarks" or similar *-- Done*
10. LINE 90+104: wrong symbol for $X^a_q$ *-- Done*
11. LINE 116: wording, replace "recalled here for the forecast-errors covariance matrix" eg by "applied to forecast-error covariance matrices" or "used for the description of the forecast-error covariance matrix", or similar. *-- Done*
12. LINE 112: remove "," before "that" – *Done at line 122*
13. LINE 158: wording "sketch", replace eg with "In practice, this step consists…" if fitting the statement. – *Done*
14. LINE 175: Suggest less metaphoric formulation replacing "To put some flesh on the bone" – *Done*
15. LINE 185: grammar, replace "In what follows" eg with "In the following" – *Done*
16. LINE 240: inconsistent units for tau_adv [s] vs 1/u [s/m]. – *Done, it is a typos: tau is D/u.*
17. Fig. 2 caption: "low resolution forecast" might be confusing here because the different resolutions were not mentioned yet. Suggest putting it into brackets here. – *Done*
18. LINE 264-265: The explanation of correlation anisotropy beginning with "e.g. in panel (e) were the …" is unnecessary. Suggest removing it. – *We preferred to keep the formulation as it is because it explains what is meant by anisotropy here.*
19. LINE 266: wording, suggest replacing "covariance error" by eg "(main parameters of the) error covariance" to avoid confusion with the uncertainty of the covariance estimate. – *Done*
20. LINE 269: bracket "(with O being … "proportional to")" unnecessary, suggest removing. –

*We preferred to keep the definition of the notation, from our experiment of previous article feedbacks.*

21. *LINE 278: referring to the general technical correction 10, Sect.2.4.3 could be introduced eg by something like "As described in Sect.2.4.2, the experiments show a gab between …" (just a suggestion)*
    *We preferred to keep as initially proposed. Thank you.*

22. *LINE 299: Connection to previous sentence unclear (may be due to unclear statement, see content comment-related technical comment about previous sentence). Suggest reformulating, maybe eg "This is demonstrated by comparing the PKF statistics to a high resolution forecast of the EnKF, ..."*
    The sentence has been modified as proposed.

23. *LINE 323: Formulation "non-linearly" unclear. Is something like "non-linear reactive chemical species" or "non-linearly reacting chemical species" meant? Suggest rewording.* –
    This has been modified as proposed.

24. *Fig. 5 caption: complicated formulation "with one orbit by level of purple transparency magnitude". Maybe it can be replaced by something like "purple curves with different transparencies".* – This has been modified as proposed.

25. *LINE 351: typo, replace "Morover" by "Moreover"* – *Done*

26. *LINE 354: it might be useful to note that all parameters are a function of model space (not only V_AB)* – This has been modified as proposed

27. LINE 372: I guess, "computation of the cross-covariance" refers to the enKF. If so, I would suggest replacing eg. by "ensemble cross-covariance" or "sample cross-covariance" to emphasis the calculation from enKF. – This has been modified as proposed

28. LINE 388: "as function of" – *Done*

29. LINE 389: "an interpolation" – *Done*

30. Fig. 7: Suggest y-axis ranging from 0% to 100% to avoid the visual impression that the relative error almost vanished to zero at certain times. *We preferred to keep as initially proposed. Thank you.*

31. *LINE 410: "are excluded"* – *Done*

32. *LINE 412: Complicated and unclear formulation. What is meant with "the true value of the averages"? Is "by an amount of 8 points of percent" equivalent to just writing "by 8%"?* – This has been modified as proposed, by writing 8%.

33. LINE 414: Unscientific formulation, suggest reformulation (assuming that sufficient literature search has been performed): eg "According to our/the authors knowledge, no proxy of cross-correlations similar to Equation (20) has been introduced up to now." – *Done*

34. *LINE 736-738: This paragraph is a conclusion which should be moved to the conclusion section.* – The idea detailed in the paragraph being in the conclusion, the paragraph has been removed.

35. Equation (21): The order of terms is inconsistent between the individual subequations. In Equation (21a)-(21e), the transport term is on the left hand side, while the T_adv are on the right hand side in Equation (21f)-(21g). Suggest putting on the same side for all subequations. – This has been modified as proposed

36. LINE 436-438: Sentence about the notation of terms starting with "Hence, each term…" is unnecessary. Suggest removing. – It has been removed.

37. LINE 441: remove "s" from "fields" – *Done*

38. LINE 450: remove "," after "dynamics" – *Done*

39. LINE 453: suggest adding "in Sec.3.3.3", eg something like "simplified dynamics of the anisotropy are used in Sec.3.3.3 to close the PKF dynamics" – *Done*

40. LINE 457-458: remove the two "," after T_adv and T_chem, respectively. – *Done*

41. LINE 460: "dynamics in Equation (21)" – *Done*

42. LINE 460: remove "," after "transport" – *Done*

43. LINE 492: Suggest replacing "at the opposite" with eg "in contrast" – *Done*

44. LINE 506: Unspecific formulation "makes appear a swing". Reformulate. – *Done*
45. LINE 508: Unclear formulation "along each specie". Meaning a different magnitude of uncertainty (=stdev?) for each of the two species? – *Done – it has been rephrased.*
46. LINE 512: Formulation, replace "What follows aim" eg by something like "The following section aims at..." or "In the following, we aim at..." – *Done*
47. LINE 513: Wording, suggest replacing "among" with eg "with respect to" – *Done*
48. LINE 537f: Suggest adding "can be neglected compared to the advection part (Fig.10c,d)" and "by W_chem-1 and W_chem-2 (Fig.10b,d)" to support the relation between statement and plot. – *Done*
49. Equation (28), LINE 540: Suggest removing Equation (28) and referring to Equation (25) instead of writing the same equation again. – *Done*
50. LINE 555: "Equation (21g), which leads to a closure of the PKF dynamics" – *Done*
51. LINE 566: "Equation (8) presented in 2" – *Done*
52. LINE 569: typo: "location $x_L$ of the chemical species" – *Done*
53. LINE 573: suggest "is the forecast cross-correlation function" – *Done*
54. LINE 585: suggest replacing "settings" by "setup" – *Done*
55. LINE 600: I would suggest replacing "Fig.12 are now discussed" by eg "are shown in Fig.12". – *Done*
56. LINE 611: Wording, suggest replacing "With less exactitude" eg with "While being less accurate" or "With less accuracy" – *Done*
57. LINE 612f: Removing "The last two panels (f) and (g) which correspond to" and adding reference to figure in brackets "of the length scales (Fig.12f,g) show a general…" would increase the readability of the sentence. – *Done*
58. LINE 630: For readability, it is most important to name the field rather than the subplot in the text. Add name of field and refer to subplot in brackets, eg "For instance, the standard deviation of species A (Fig.13c) shows important …" – *Done*
59. LINE 632: similarly to above, I would suggest adding "specie B (Fig.13d) for which ..." and remove last part of sentence "as panel (d) shows". – *Done*
60. LINE 640: "which has been detailed in paragraphs 3.3.1-3.3.3" could be shortened to "(compare Sec.3.3)" – just a suggestion. – *Done*
61. LINE 680: "terms" – *Done*
62. LINE 697: replace ", set as" by ", by setting as" or "defined to be" – *Done*
63. LINE 698: replace "produced" by "produce" – *Done*
64. LINE 705: remove "one" -> "for each of the six" – *Done*
65. LINE 713: suggest removing "of the six ones" because it provides no additional information. – *Done*
66. LINE 718f: Missing "s " in "appears". The rather long sentence could also be shortened significantly eg to something like "The impact of chemistry leads to non-zero cross-correlations between all pairs of species (Fig.15, right column, except the auto-correlation in Fig.15p)." – *Done*
67. LINE 719: The word "roughness" is quite unspecific. Suggest replacing by eg "small-scale spatial variation" if that fits the content. – *Done*
68. LINE 727: missing " 2.4.3" – *Done*
69. LINE 730: remove additional bracket ")" after "Sec.3.5.2" – *Done*
70. LINE 753: remove final "s" from "describes" – *Done*
71. LINE 757: missing "s" in "reduces" – *Done*
72. LINE 787: replace "study" by eg "studied" or "investigated" – *Done*
73. LINE 792: typo, "We consider four chemical species, …" – *Done*
74. LINE 799: meaning of "(that is in excess)" unclear. Reformulate or remove. – *Done*
75. LINE 815: double use of word "initial". Remove. – *It has been rephrased.*
76. LINE 818: remove additional ")" – *Done*
77. LINE 822: empty subequation. Remove. – *Done*

78. LINE 824: remove final "s" in "tensors" – *Done*
79. LINE 843: replace "X^f+eps_f" with "X^f = X^t + eps_f"? – *Done*
80. LINE 844: remove double "a" -> "Equation (8a)" – *Done*
81. LINE 856f: V^o is the observation error variance. Reformulate, eg. "observation and forecast error variances V_ZL^o (x_L) = ...., V_ZL^f (x_L) + ..." or similar. – *Done*

[revised manuscript text omitted]

---

## Author Comment (AC2)

First of all, we would like to thank the referee for his/her review and for giving us the opportunity to improve our paper. We added our acknowledgements to the referee in the new manuscript.

Now, we organized the answer to the comments as follows. First, we list some changes afford to the manuscript then detail our answers to the questions raised by the referee.

**List of changes for the revision**

*Minor changes*

Errors in the label of some figures in the text have been corrected:
L491 Fig.8-(d) -> Fig. 8-(f)
L493 Fig.9-(d) -> Fig. 9-(f)

The figures have been re-rendered to improve their quality (size of legend, title,..). Because of the sampling noise inherent to the ensemble estimation, the values they show can have changed, without modifying the meaning or the robustness of the results. For instance, in Fig. 7, the averages where 22,8% for l_A=l_B while it is 23,1% now. As another example, the curves in Fig. 10 or 11 are not strictly the same as for the first version of the manuscript while we recognize the same patterns and conclusions.

*Differences between the two versions of the manuscript*

To facilitate the comparison between the two version of the manuscript, a companion version of the manuscript lists all the modifications where old (new) statements are in red (blue). But the line numbers will refer to the revised version of the manuscript (not to the companion version).

**Answer to the question of the referees**

We copied your commentary in italics below, we reply in normal blue font.

General feedback:

*"This is an interesting and original paper showing that a fairly sophisticated implementation of a Kalman filter can be done using parameters to define the local form of the evolving state error covariance instead of using a costly large ensemble. The method is exhibited using a simple chemistry transport model, with convincing success in this example. It suggests that an extension of the same technique might work also for a more complex model that includes the dynamical variables of the flow itself. At each time, and at each point in space, the covariance amongst the variables is approximately modeled by, for example, a multivariate Gaussian covariance, and it is the parameters prescribing this covariance which the PKF aims to evolve. The authors note that this becomes more burdensome in proportion to the square of the number of variables described by the local covariance, so it will be of great interest to see, in future, whether their technique remains competitive with ensemble methods in cases of greater dynamical complexity. But the paper is generally well written and convincingly argued, so I am happy to recommend that it be published*

*after the correction of several very minor points listed by line numbers below."*

*We would like to thank the referee for her/his interest for the work and for underlining the originality of the contribution. As she/he mentioned, the next step will be to see if it can be extended for real chemical schemes which contains much more chemical species, a work that would interesting to join with an adaptation of the chemical scheme itself.*

Minor points:

*L15 "..to the quality.."*
  *This has been corrected, thank you.*
*L24: "..correct other concentrations and reduce .."*
  *This has been corrected, thank you.*
*L38,39: "..needs the introduction of filtering.."*
  *This has been corrected, thank you.*
*L40: "..to set correlations to zero.."*
  *This has been corrected, thank you.*
*L53: What is Reynold decomposition – is there a reference you could add?*
  *This corresponds to the  Reynolds averaging technique. We added a reference to the Chapter 4 of the book of Lesieur (2007) who details it.*
*L115, L116, : "diagnoses" (plural)*
  *This has been corrected, thank you.*
*L138: "..of a covariance model..", "..build a heterogeneous.."*
  *This has been corrected, thank you.*
*L145: "non-obvious" (hyphenate?)*
  *This has been corrected, thank you.*
*L148: "..approximation to reproduce.."*
  *This has been corrected, thank you.*
*L198: "designed"*
  *This has been corrected, thank you.*
*L268: replace "resumed" by "reduced"*
  *This has been corrected, thank you.*
*L294: "scales"*
  *This has been corrected, thank you.*
*L295: "..to attributing.."*
  *This has been corrected, thank you.*
*L306: "dash-dotted"*
  *This has been corrected, thank you.*
*Caption to Fig. 5: "clockwise" (not "clockwisely")*
  *This has been corrected, thank you.*
*L394: "cross-correlations"*
  *This has been corrected, thank you.*
*L410: "excluded"*
  *This has been corrected, thank you.*
*L412: "overestimation of the true.."*
  *This has been corrected, thank you.*
*L413:' replace "points of percentage" by "percentage points"*
  *This has been corrected, thank you.*
*L126: "guarantee"*
  *This has been corrected, thank you.*
*L441: "independent of anisotropy"*
  *This has been corrected, thank you.*

*L452: "rehabilitated to quantify.."*

*This has been corrected, thank you.*

*L458: "removed to focus.."*

*This has been corrected, thank you.*

*L491: "there are no.."*

*We chose to replace "there is no oscillations" by "there is no oscillation" (singular for oscillation).*

*L496: " ..to non-constant.."*

*This has been corrected, thank you.*

*L498: "stationary"*

*This has been corrected, thank you.*

*L499: "..anisotropy is equal.."*

*This has been corrected, thank you.*

*L500: "..stationarity.."*

*This has been corrected, thank you.*

*L510: ".. each of the terms.."*

*This has been corrected, thank you.*

*L523: ''..in the anisotropy.."*

*This has been corrected, thank you.*

*L525: " ..as they lead to.."*

*This has been corrected, thank you.*

*L575: "This leads to.."*

*This has been corrected, thank you.*

*L581: "..focuses on the forecast.."*

*This has been corrected, thank you.*

*L591: "independent"*

*This has been corrected, thank you.*

*L603: "A horizontal.."*

*This has been corrected, thank you.*

*L630: "statistics."*

*This has been corrected, thank you.*

*L647: Expand the acronym "GRS" in the main heading 4.*

*This has been corrected, thank you.*

*L675: "..observation assimilation in this.."*

*This has been corrected, thank you.*

*L680: "In terms.."*

*This has been corrected, thank you.*

*L682: "..presents"*

*This has been corrected, thank you.*

*L685: "..dynamics, and is.." (insert comma?)*

*This has been modified, thank you.*

*L737: "..makes apparent some.."*

*This has been corrected, thank you.*

*L741: "..remain.."*

*This has been corrected, thank you.*

*L742: "we have not questioned.."*

*This has been corrected, thank you.*

*L743: "absolute"*

*This has been corrected, thank you.*

*L748: "This simplification led.."*

*This has been corrected, thank you.*

*L753: "describe.."*

*This has been corrected, thank you.*

*L774—777: It might be better to combine these lines into a single paragraph.*

*This has been modified, thank you.*

*L787: "..studied.."*

*This has been corrected, thank you.*

*L840: "..we chose.."*

*This has been corrected, thank you.*

*L844: It is not clear what equation number you meant by "(8aa)"*

*It was Eq.(8a). This has been corrected, thank you.*

*L875: "..programme.."*

*This has been corrected, thank you.*

[revised manuscript text omitted]

$$RP + RP \stackrel{k_5}{\rightarrow} RP \tag{32e}$$

$$RP + NO_2 \stackrel{2 \cdot k_6}{\rightarrow} S(N)GN \tag{32f}$$

where ROC, RP and S(N)GN respectively mean *Reactive Organic Compound*, *Radical Pool* et *Stable (Non-) Gaseous Nitrogen product*. In this chemical model, additional processes such as photolysis rate variation, ground deposits or atmospheric emissions of certain pollutants are represented.

The system of equations of the GRS-CTM writes:

$$\partial_t [ROC] = -\partial_x \left( u \cdot [ROC] \right) - \lambda [ROC] + E_{ROC} \tag{33a}$$

$$\partial_t [RP] = -\partial_x \left( u \cdot [RP] \right) - \lambda [RP] + k_1(t)[ROC] - [RP] \left( k2[NO] + 2k_6[NO_2] + k_5[RP] \right) \tag{33b}$$

$$\partial_t [NO] = -\partial_x \left( u \cdot [NO] \right) - \lambda [NO] + E_{NO} + k_3(t)[NO_2] - [NO] \left( k_2[RP] + k_4[O_3] \right) \tag{33c}$$

$$\partial_t [NO_2] = -\partial_x \left( u \cdot [NO_2] \right) - \lambda [NO_2] + E_{NO_2} + k_4[NO][O_3] + k_2[NO][RP] - [NO_2] \left( k_3(t) + 2k_6[RP] \right) \tag{33d}$$

$$\partial_t [O_3] = -\partial_x \left( u \cdot [O_3] \right) - \lambda [O_3] + k_3(t)[NO_2] - k_4[NO][O_3] \tag{33e}$$

$$\partial_t [S(N)GN] = -\partial_x \left( u \cdot [S(N)GN] \right) - \lambda [S(N)GN] + 2k_6[NO_2][RP] \tag{33f}$$

where for a specie $Z$: $[Z](t,x)$ denotes the concentration field ; and for $Z \in \{ROC, NO, NO_2\}$, $E_Z(x) = E_Z^0 \mu(x)$ denotes the stationary emission field modulated by the smooth ocean/land mask $\mu(x) \in [0,1]$ shown in Fig. 14(b), and of maximum emission $E_Z^0$ whose value is given in Table 1 (right column). The ground deposition is represented by terms in $\lambda$, with a magnitude of 2% per day. Kinetic parameters and chemical reaction rates are set as follows: since Eq. (33a) and Eq. (33c) depends on the solar radiation, $k_1$ and $k_3$ evolve in time to represent the diurnal cycle while they are related by $k_1 = 0.152k_3$ (Fig. 14(c)); the other rates are constant and given in Table 1.

In a new numerical experiment, the PKF forecasts will be compared with those of an EnKF (of size 1600). There is no  observation assimilation in this simulation.

| | | | | |
|---|---|---|---|---|
| $k_3(t)$ | $0.624\exp\left(-\frac{\lfloor(t\equiv24)-12\rfloor^3}{100}\right)$ | $k_1(t)$ | $0.00152k_3(t)$ |
| $k_2$ | 12.3 | $E^0_{ROC}$ | 0.0235 |
| $k_4$ | 0.275 | $E^0_{NO}$ | 0.243 |
| $k_5$ | 10.2 | $E^0_{NO_2}$ | 0.027 |
| $k_6$ | 0.12 | $\lambda$ | $0.02\text{day}^{-1}$ |

In $k_3$ definition, the symbol $\equiv$ corresponds to the modulo operator. Emission rates in ppbCday$^{-1}$ for ROC or ppbday$^{-1}$ for NO$_x$, and the kinetic rates in ppb$^{-1}$min$^{-1}$, except for $k_3$ and $k_1$ in min$^{-1}$.

**Table 1.** GRS settings

[Figure]

**Figure 14.** Settings of the GRS-CTM. (a): , with the predefined heterogeneous and stationary wind field , (bpanel a) : and emission field, inventories mask (epanel b): $k_3$ values along ; and with the day 
[revised manuscript text omitted]

---

## Author Comment (AC3)

First of all, we would like to thank the referee for his/her review and for giving us the opportunity to improve our paper. We added our acknowledgements to the referee in the new manuscript.

Now, we organized the answer to the comments as follows. First, we list some changes afford to the manuscript then detail our answers to the questions raised by the referee.

**List of changes for the revision**

*Minor changes*

Errors in the label of some figures in the text have been corrected:
L491 Fig.8-(d) -> Fig. 8-(f)
L493 Fig.9-(d) -> Fig. 9-(f)

The figures have been re-rendered to improve their quality (size of legend, title,..). Because of the sampling noise inherent to the ensemble estimation, the values they show can have changed, without modifying the meaning or the robustness of the results. For instance, in Fig. 7, the averages where 22,8% for l_A=l_B while it is 23,1% now. As another example, the curves in Fig. 10 or 11 are not strictly the same as for the first version of the manuscript while we recognize the same patterns and conclusions.

*Differences between the two versions of the manuscript*

To facilitate the comparison between the two version of the manuscript, a companion version of the manuscript lists all the modifications where old (new) statements are in red (blue). But the line numbers will refer to the revised version of the manuscript (not to the companion version).

**Answer to the question of the referees**

We copied your commentary in italics below, we reply in normal blue font.

General feedback:

1. *"The manuscript describes an application of "parametric" Kalman Filter to data assimilation for chemical modeling. I believe it can be significantly shortened by providing just one example of the application of PKF."*

This is not an application, but an investigation of the feasibility of the PKF in multivariate chemistry which still corresponds to research but not operational routine, as suggested by the title we chose "Toward a multivariate formulation of the PKF assimilation: application to a simplified chemical transport model".

2. *"The assimilation method relies on evolving covariance error statistics based on prognostic equations rather than obtaining them from ensemble of states as in more traditional ensemble KFs. Increased complexity of forecast model equations corresponds to the increased complexity of equations for error covariances and increased number of simplifying, maybe arbitrary assumptions."*

In this contribution we detailed the contribution of the transport vs. the chemistry so to avoid arbitrary assumptions. We agree that this is an academic approach that will have to be confirmed in a real model, may be following the same methodology as the one we detailed here based on the PKF dynamics.

Concerning the complexity, we agree it is a limit of the present formulation of the PKF, as it is mentioned in the conclusion of the manuscript. However, even in real applications, only a few chemical species are often assimilated -- as discussed in the next point 3 about the real-time operational CAMS 2.40 ensemble forecast system – which could limit the complexity of the PKF. However, other simplification can be interesting as the one considered by the recent work of Voshtani et al. (2022a,b) (also discussed in the next point 3).

3. *"Chemical model forecasts do not depend on chemical parameterizations alone but also on evolving physical state (meteorology). Therefore, it is difficult to imagine how a proposed system of modeling error covariances using chemical parameterizations alone can favorably compare with a traditional system that relies on an ensemble of simulations forming a basis to obtain estimates of error covariances of species. I don't believe that a proposed system of assimilation using PKF will be successful in application to real-world data assimilation for air quality such as with model MOCAGE that the authors mention. It would be valuable if a positive result can be demonstrated. Otherwise, the proposed application of "parametric" KF is just a curiosity that can be described in a significantly reduced manuscript.*

*I believe that the manuscript should be much revised and its publication subject to a demonstration that the application of PKF in a realistic DA scenario provides benefits that are comparable to those using a traditional EnKF approach."*

We agree with the referee that in real atmosphere, chemistry influence the meteorology, as it is also true for the biosphere, or the ocean,.. that would be represented in Earth modelling systems.

However, this is not the case for Chemical Transport Models which only use a wind that comes from a global or a regional model, whose computation is made before, and without retroaction on atmospheric dynamics (offline computation of the weather). Hence, the chemistry has no influence on the meteorology in CTMs.

This is more clearly detailed in the revised version of the manuscript: "However, in CTMs chemistry do not influence the meteorology, which is of course a crude approximation of the true atmosphere." L19-20

We agree that a CTM is not consistent with the real atmospheric dynamics, but this approach has been shown useful in air quality, and is used in several operational or research centres e.g. CHIMERE (LSCE, France) or CMAQ-DDM (EPA, USA), while two-way systems also exist e.g. WRF-CHIMERE or WRF-CMAQ. Some operational centres have, or are going to develop, in-line models where the meteorology is integrated in-step with the chemistry e.g. GEM-MACH (ECCC, Canada). But again, depending on the version or the options of the forecast, the coupling to meteorology can be one-way (chemistry does not influence the meteorology) or two-way.

Note that the operational air quality ensemble of forecast (CAMS2.40, https://atmosphere.copernicus.eu/cams-european-air-quality-ensemble-forecasts-welcomes-two-new-state-art-models) daily products an ensemble of 11 models so to cover the following 4 days, by assimilating ozone, nitrogen dioxide, sulphur dioxide, and fine particulate matter PM2.5 and PM10. These forecasts are performed from specific emissions datasets (product of CAMS) and are driven by ECMWF's high resolution weather forecasts.

The models included in the consortium are 11: CHIMERE from INERIS (France), EMEP from

MET Norway (Norway), EURAD-IM from Jülich IEK (Germany), LOTOS-EUROS from KNMI and TNO (Netherlands), MATCH from SMHI (Sweden ), MOCAGE from Meteo-France (France), SILAM from FMI (Finland), DEHM from AARHUS UNIVERSITY (Denmark), GEM-AQ from IEP-NRI (Poland), MONARCH from BSC (Spain) and MINNI from ENEA (Italy) .

Among the 11 models, 10 are pure CTMs (offline meteorology given by ECMWF weather forecast with out influence of the chemistry on the meteorology), and one is online with a nudging to the meteorology.

We improved the description of the state of the art by adding a reference to  CAMS2.40 in the manuscript "The advantage of a CTMs is that it allows air quality prediction at a low numerical cost, and is used in several operational centers. For instance, the CAMS regional air quality production, which daily forecast an ensemble of 11 members that covers the following 4 days, is performed from the integration of 11 models from which 10 are CTMs." L20-23

For MOCAGE, the assimilation is univariate (the assimilation of a chemical species has no influence on the others), with homogeneous spatial correlations, and background error variance are proportional to the concentration of the chemical species to assimilate (a very reduced set of chemical species compared to the hundred of chemical species that count the chemical schemes). The configuration of the assimilation of MOCAGE is now detailed as an example of forecast systems in CAMS 2.40:
"Note that in operational applications, chemical species are often assimilated separately e.g. in CAMS 2.40, the univariate 3DVar system of MOCAGE is used for the assimilation of ozone, nitrogen dioxide, sulphur dioxide, and fine particulate matter PM2.5 and PM10 (following a configuration similar to the one used for MACII, detailed by Marécal et al. (2015))." in L28-31

So, the research we are doing tends to improve this state of the art in assimilation for CTM and the numerous limitations that are encountered (constant length-scale, background error variance set as proportional to concentrations, ..), and in particular by the exploration of the multivariate assimilation.

Furthermore, we consider that the PKF, as a theoretical tool, can help to improve our knowledge and use of ensemble method, as it is mentioned in the manuscript about the loss of variance observed in EnKF method for CTM, and that can be explained from the PKF perspective (see eg Ménard et al. 2021). This makes the PKF appear not as a curiosity but as a complementary tool for research and for real applications. Note that as a real application of the PKF we can mention the recent work of Voshtani et al. (2022a,b) who considered as simplified version of the PKF where only the variance is updated (from the representors approach) and predicted (as a transport + a inflation that mimics the model error), with a  stationary and homogeneous horizontal correlations (constant length-scale per model level), that they called Parametric Variance KF (PvKF) ; and apply their algorithm on assimilation of methane from GOSAT observations with success. We indicated this recent work in the manuscript:

[revised manuscript text omitted]
 further explore To explore the ability of the PKF possibilities, to apply to a more complex chemical model is considered: scheme, an intermediate chemical model is now introduced, the generic reaction set (Azzi et al., 1992; Haussaire and Bocquet, 2016) -(GRS), then used to validate the PKF forecast.

**4.1 Description of the GRS model**

GRS describes the dynamics of a reduced number of chemical species or *pseudo*-species. Hence, six species are considered and interact as

$$ROC + h\nu \overset{k_1(t)}{\to} RP + ROC \tag{32a}$$

$$RP + NO \overset{k_2}{\to} NO_2 \tag{32b}$$

$$NO_2 + h\nu \overset{k_3(t)}{\to} NO + O_3 \tag{32c}$$

$$NO + O_3 \overset{k_4}{\to} NO_2 \tag{32d}$$

$$RP + RP \overset{k_5}{\to} RP \tag{32e}$$

$$RP + NO_2 \overset{2 \cdot k_6}{\to} S(N)GN \tag{32f}$$

where ROC, RP and S(N)GN respectively mean *Reactive Organic Compound*, *Radical Pool* et *Stable (Non-) Gaseous Nitrogen product*. In this chemical model, additional processes such as photolysis rate variation, ground deposits or atmospheric emissions of certain pollutants are represented.

The system of equations of the GRS-CTM writes:

$$\partial_t[ROC] = -\partial_x(u \cdot [ROC]) - \lambda[ROC] + E_{ROC} \tag{33a}$$

$$\partial_t[RP] = -\partial_x(u \cdot [RP]) - \lambda[RP] + k_1(t)[ROC] - [RP](k2[NO] + 2k_6[NO_2] + k_5[RP]) \tag{33b}$$

$$\partial_t[NO] = -\partial_x(u \cdot [NO]) - \lambda[NO] + E_{NO} + k_3(t)[NO_2] - [NO](k_2[RP] + k_4[O_3]) \tag{33c}$$

$$\partial_t[NO_2] = -\partial_x(u \cdot [NO_2]) - \lambda[NO_2] + E_{NO_2} + k_4[NO][O_3] + k_2[NO][RP] - [NO_2](k_3(t) + 2k_6[RP]) \tag{33d}$$

$$\partial_t[O_3] = -\partial_x(u \cdot [O_3]) - \lambda[O_3] + k_3(t)[NO_2] - k_4[NO][O_3] \tag{33e}$$

$$\partial_t[S(N)GN] = -\partial_x(u \cdot [S(N)GN]) - \lambda[S(N)GN] + 2k_6[NO_2][RP] \tag{33f}$$

where for a specie $Z$: $[Z](t,x)$ denotes the concentration field ; and for $Z \in \{ROC, NO, NO_2\}$, $E_Z(x) = E_Z^0 \mu(x)$ denotes the stationary emission field modulated by the smooth ocean/land mask $\mu(x) \in [0,1]$ shown in Fig. 14(b), and of maximum emission $E_Z^0$ whose value is given in Table 1 (right column). The ground deposition is represented by terms in $\lambda$, with a magnitude of 2% per day. Kinetic parameters and chemical reaction rates are set as follows: since Eq. (33a) and Eq. (33c) depends on the solar radiation, $k_1$ and $k_3$ evolve in time to represent the diurnal cycle while they are related by $k_1 = 0.152k_3$ (Fig. 14(c)); the other rates are constant and given in Table 1.

In a new numerical experiment, the PKF forecasts will be compared with those of an EnKF (of size 1600). There is no observations assimilations observation assimilation in this simulation. A brief overview of the PKF formulation for is now exposed.

| $k_3(t)$ | $0.624\exp\left(-\frac{\lvert(t\equiv24)-12\rvert^3}{100}\right)$ | $k_1(t)$ | $0.00152k_3(t)$ |
|---|---|---|---|
| $k_2$ | 12.3 | $E^0_{ROC}$ | 0.0235 |
| $k_4$ | 0.275 | $E^0_{NO}$ | 0.243 |
| $k_5$ | 10.2 | $E^0_{NO_2}$ | 0.027 |
| $k_6$ | 0.12 | $\lambda$ | $0.02\text{day}^{-1}$ |

In $k_3$ definition, the symbol $\equiv$ corresponds to the modulo operator. Emission rates in ppbCday$^{-1}$ for ROC or ppbday$^{-1}$ for NO$_x$, and the kinetic rates in ppb$^{-1}$min$^{-1}$, except for $k_3$ and $k_1$ in min$^{-1}$.

**Table 1.** GRS settings

[Figure]

**Figure 14.** Settings of the GRS-CTM. (a): , with the predefined heterogeneous and stationary wind field , (bpanel a) : and emission field, inventories mask (epanel b): $k_3$ values along ; and with the day 
[revised manuscript text omitted]

---

## Referee Report (RR1)

I find the responses to my inquiries lacking.

1. I suggested that the manuscript should be shortened because it tests patience of the reader with multiple examples that seem repetitious. That has not been addressed.

2. In my opinion the authors don't present "an application to a simplified chemical transport model" but present a concept of using PKF to propagating error covariances. There is big difference between application in a CTM model and illustrating a concept. Reference to CAMS ensemble that the authors mention in the reply is farfetched. In opinion it is hard to find any commonalities between CAMS ensemble and concept that authors present.

3. Reply to my 3$^{rd}$ inquiry is long but, in my opinion, entirely misses the point. Whether the model has interactive meteorology and chemistry has no relevance to the data assimilation approach discussed in the manuscript. In a common application the spread of an ensemble of chemical model realizations may come from varying meteorological states, be it wind in the simplest case for off-line model (u,v) plus state variables (T,q,…) in an online model. Varying meteorology will contribute to spread of the chemical ensemble because it will affect concentrations of the species. As noted in my review, I don't believe that the approach presented will lead to an efficient data assimilation. To be convinced I would welcome an application that the authors proposed using MOCAGE. Otherwise, as I pointed in my review a theoretical basis for a concept may exist but the concept itself will remain just a curiosity without any prospect for a real-world application.

In the future, please be concise and to the point.

Based on the above, I believe that authors have not provided satisfactory replies to my inquires for me to recommend the publication in Nonlinear Processes in Geophysics.

---

## Referee Report (RR2)

Based on the 1st review phase (interactive discussion), the authors performed several modifications to the manuscript which improve its quality and clearness significantly. However, some aspects are still to be corrected from my point of view. I propose to accept the manuscript with minor corrections.

1. Main comment:

The total length of the manuscript was a point of discussion in the 1st review phase. I completely agree with the authors argumentation that each part of the manuscript is relevant. Indeed, all parts of the manuscript are important and contain interesting results. However, it's a combination of the actual length on paper and the amount of new content and results that makes the manuscript appearing very long. My main concern is that readers get discouraged or lost.

My recommendation is to move less important or "assisting" parts in the appendix - being still easily accessible by the interested reader but reducing the length of the main text. I would suggest one or more of the following parts (or also others depending on the authors argumentation):

A) Sec 2.4 - univariate experiments: The univariate experiments in Sec.2.4 provide a good preparation for the subsequent parts especially for unexperienced readers, but the results are mainly an intermediate step for the interpretation of the multivariate results. Ideally this aspect could be addressed in a preceding publication, but could also be moved to the appendix of this paper. - this would shorten the main text by about 5 pages (in its current version)

B) Sec.3.2 - LV-CTM formulation: While the formulation of the PKF dynamics for the LV-CTM (Sec.3.2.1) and the multivariate PKF analysis (Sec.3.2.5) should stay in the main text, Sec.3.2.2 (evaluation of chemistry alone) and Sec.3.2.3 (contribution of individual terms) are interesting, yet less important parts which are evaluating the enKF results preparing the closure of the PKF equations. Thus, I recommend moving Sec.3.2.2 and 3.2.3 in the Appendix and referring to them in the actual formulation of the PKF closure in Sec.3.2.4 (e.g. adding reference to Sec.3.2.3 in l.555, ...). - this would shorten the main text by about 5 pages (in it's current version)

C) Sec.3.3 - LV-CTM experiments: Algorithm 1 provides a good summary, but is rather long. It could also be moved to the appendix because all important steps are described in the manuscript. - this would shorten the main text by about 1 page (in its current version)

From my point of view, B) has more priority to be moved into the appendix than A) unless the authors argue differently.

2. Minor technical corrections:
- related to the 1st review, reviewer1, technical correction 10: There are still some introducing sentences of next sections at end of sections. I suggest removing or moving them to beginning of new section (eg: before Sec.2.4, 2.4.1 ,3.1.3, 4.2)
- related to the 1st review, reviewer1, technical correction 9: Remove technical figure description "blue dashed lines" at l.314 and similarly l.419

- l.43: Suggest reformulating the new sentence starting with "to zero" by moving these two words to the end: "...to set the ensemble estimation of the multivariate correlation to zero, ..."
- l.194: In the current version the title of Sec.2.4 was not changed as described in the 1st review, reply to reviewer1, general correction 2.4
- l.106: Remove "that"
- l.295-297: The complex sentence structure makes it difficult to follow. Do you want to say something like: Because the equation for the mean (Eq.(13)) is linear, the error field is given by an equivalent equation. And for these equation, the shorter the correlation length scale the larger is the error magnitude. ??

---

## Author Response (AR2)

**Reply to Zoltan Toth for the Editor Decision and Comments**

First of all, we would like to thank the Editor for his comments and for giving us the opportunity to improve our paper. We added our acknowledgements to the editor in the new manuscript.

Now, we organized the answer to the comments as follows. First, we list some changes afford to the manuscript then detail our answers to the questions raised by the Editor in his decision and in the comments of Jan. 25th.

**List of changes for the revision**

**Minor changes**

- Fig. 2 and Fig. A1 'y' label has been modified using \chi in place of A.
- There was a typos in Eq.(D2) that has been corrected replacing \sum_{k=1}^N_e by \sum_{k=1}^{N_e}

**Differences between the two versions of the manuscript**

To facilitate the comparison between the two version of the manuscript, a companion version of the manuscript lists all the modifications where old (new) statements are in red (blue). But the line numbers will refer to the revised version of the manuscript (not to the companion version).

**Answer to the comments**

We copied your commentary in italics below, we reply in normal blue font.

**I) Comment associated with the Editor decision:**

*1) The first and most important point is that both reviewers still find the manuscript excessively long (see Rev. 1's main comment, and Rev. 3's 1st comment). To address this, I encourage you to consider find Rev. 1's suggestion to move parts of the main text into Appendixes. Following her suggestion would shorten the main text significantly.*

We followed the recommendations proposed by Rev. 1 to move details in Appendixes, but kept a reduced version of the introduction of the univariate testbed at the end of the section on the background on the parametric formulation. The reason is that a large part of the material would have been moved in the multivariate part (details of the numerical setting e.g. time and space numerical scheme, illustration of the stationary wind used in Fig.1, description of the estimation of variance and anisotropy,..) and also because the description of the PKF dynamics helps to appreciate the effect of the chemistry in the multivariate framework (comparison between Fig. 2 and Fig. 6). See the detailed answer to Rev. 1

With these modifications, the manuscript has been significantly shorten.

*2) Another important point, first outlined in Rev. 3's 1st review (see 3rd par. in her/his first review) is further elaborated in the second review of Rev 3 (see her/his 2nd and 3rd comments in second review). [..] To address these points, you may consider corresponding changes wherever they are appropriate, including the Abstract, Introduction, main text, and/or Conclusions.*

*2.1) First, the subject is not specified clearly for your work - i.e., you may need to clearly state that uncertainties in chemical variables only due to chemistry, but not due to the dynamical evolution of the state are assessed in your work.*

This is now clearly stated :
- in the abstract "This contribution focuses on the situation where the uncertainty is due to the chemistry but not due to the uncertainty of the weather"(l 4-5)
- in the introduction "This contributions only focuses on the uncertainty dynamics due to the chemistry without accounting for the part of the uncertainty of the weather e.g. we do not take into account the uncertainty of the wind that transports the chemical species." (l 92-93)
- in the conclusion "While a significant portion of the air quality uncertainty is due to meteorology (e.g. the uncertainty of the wind used for the transport), the present work focuses on the situation where the uncertainty in chemical variables is due solely to chemistry as it evolves along a given meteorological situation." (l620-622)

*2.2) And second, the context and significance of the work need to be clarified accordingly - you may want to point out that in any practical application of the methodology you developed and tested, one must also address uncertainties in chemical variables due to uncertainties in the meteorological state variables. Introducing any ideas about how this could possibly be done is not a prerequisite of publication, but would certainly strengthen your report.*

The conclusion now addresses this point, and introduces how it would be interesting to take into account of the wind uncertainty in the framework of the parametric Kalman filter:
"In addition, since we have focused on the uncertainty due to chemistry, it would be interesting to address the part of the uncertainty due to meteorology. For a CTM like MOCAGE, this could be done by considering an ensemble of weather forecasts with each member used as a forcing for a single CTM forecast. However, this solution would lead to multiple CTM forecasts, which would be expensive. Therefore, from the perspective of using a PKF (applied to a CTM), a less expensive solution would be to consider a single PKF forecast where the wind is uncertain (stochastic advection wind), with the wind uncertainty characterized by the variance and anisotropy tensor estimated from the weather forecast ensemble.  The challenge will be to find an appropriate closure for the unknown terms in the dynamics, including the cross-correlation between the wind error and chemical species, with the help of this contribution on multivariate statistics."(l 654-661)

**II) Editorial comments on egusphere-2022-928, 25 Jan. 2023:**

*1) I found your responses to General comment 2 (G2), G2.1, Technical comment 1 (T1), and T6 from Rev. 1, where you state your preference to disregard reviewer suggestions lacking any reason or explanation. In your response to the next round of reviews, if you disregard any suggestion, can you please provide your reasons, beyond stating your preference? This is especially important in case of major recommendations, like the one highlighted in my next comment.*

Indeed, we should have explained better why we thought it was better to delete the parts that may have caused problems for Rev. 1 and Rev. 3. There were three reasons for this:

The first is that the PKF is new and that from experience it is not yet possible to rely solely on references -- even when they exist -- because readers want the contributions to be self-sufficient. The second is that the multivariate framework chosen is complex since it takes into account both chemistry and transport. However, in order to understand which part of the uncertainty is linked to transport and which to chemistry, the passage through the univariate framework seemed to us unavoidable. Finally, we considered that the details provided contributed to the rigor of the work done, to avoid arbitrariness in the proposals we made.

At this stage, we did not think of moving what might have bothered the two referees to the appendix, as now proposed by Rev. 1. With this transfer to the appendix, the interested reader can follow the detail while making the manuscript much more readable.

*2) Rev. 1's G2 and Rev. 3's 1st general comments are aligned - they both criticize the excessive length of your manuscript. I sympathize with these comments. On the other hand, Rev 2 has not raised this as an issue. As we wait for the 2nd round of reviews, should Rev. 1 and 3 insist on this major recommendation, your options would include complying, at least partially, and/or explaining why keeping all material together would be beneficial.*

Following the recommendations of Rev1, the length of the manuscript has been significantly reduced now (as more explained in the answer to Editor decision point 1).

3*) Rev. 1, Specific comment 4 - wonder if the related additions are best placed in the Conclusions, or in Sections 3.3.2 and 3.5.2, where they may aid a more nuanced interpretation of the results?*

We introduced the nuance in the conclude section 3.3.2 (without change to the Conclusion where it is still mentioned): "It would be interesting to assess the robustness of the results, including whether the advection terms remain dominant under different conditions, such as weaker winds or accelerated chemistry, from a set of operational CTM predictions." (l 519-520)

4) Rev 1, Technical comment T2: is "autocorrelation" and "cross-correlation" the standard version of spelling?

We now use the terminology found in Derber and Bouttier 1999 who used "autocorrelation" and "cross-correlation", that is now better introduced in the manuscript "In multivariate covariance modeling applied in meteorology, these correlations are respectively denoted by autocorrelations and cross-correlations (Derber and Bouttier, 1999)." (l32-33)

5) Rev. 1, T7: what do you mean by your addition " prediction" related error? Do you want to refer to chaotically amplifying initial error in forecasts? If so, please clarify

We replaced "prediction" by "chaotically amplification of initial error in forecast" which is more clear now (see l 113)

6) Rev. 1, Technical correction TC38 - I find Rev. 1's suggestion reasonable. Much of Section 5 reads as a long summary.  My understanding is that NPG has no strict rules as to the title of various sections.

We used the token \conclusion of the Latex template, that is now replaced by a section called "Summary and conclusions".

First of all, we would like to thank Annika Vogel for her review and for giving us the opportunity to improve our paper.

Now, we organized the answer to the comments as follows. First, we list some changes afford to the manuscript then detail our answers to the questions raised by the referee.

**List of changes for the revision**

*Minor changes*

- Fig. 2 and Fig. A1 'y' label has been modified using \chi in place of A.
- There was a typos in Eq.(D2) that has been corrected replacing \sum_{k=1}^N_e by \sum_{k=1}^{N_e}

*Differences between the two versions of the manuscript*

To facilitate the comparison between the two version of the manuscript, a companion version of the manuscript lists all the modifications where old (new) statements are in red (blue). But the line numbers will refer to the revised version of the manuscript (not to the companion version).

**Answer to the question of the referees**

We copied your commentary in italics below, we reply in normal blue font.

*1. Main comment:*

*The total length of the manuscript was a point of discussion in the 1st review phase. I completely*

*agree with the authors argumentation that each part of the manuscript is relevant. Indeed, all parts*

*of the manuscript are important and contain interesting results. However, it's a combination of the*

*actual length on paper and the amount of new content and results that makes the manuscript*

*appearing very long. My main concern is that readers get discouraged or lost.*

*My recommendation is to move less important or "assisting" parts in the appendix - being still easily*

*accessible by the interested reader but reducing the length of the main text. I would suggest one or*

*more of the following parts (or also others depending on the authors argumentation):*

*A) Sec 2.4 - univariate experiments: The univariate experiments in Sec.2.4 provide a good*

*preparation for the subsequent parts especially for unexperienced readers, but the results are*

*mainly an intermediate step for the interpretation of the multivariate results. Ideally this aspect*

*could be addressed in a preceding publication, but could also be moved to the appendix of this*

*paper. - this would shorten the main text by about 5 pages (in its current version)*

*B) Sec.3.2 - LV-CTM formulation: While the formulation of the PKF dynamics for the LV-CTM*

*(Sec.3.2.1) and the multivariate PKF analysis (Sec.3.2.5) should stay in the main text, Sec.3.2.2*

*(evaluation of chemistry alone) and Sec.3.2.3 (contribution of individual terms) are interesting, yet less important parts which are evaluating the enKF results preparing the closure of the PKF equations. Thus, I recommend moving Sec.3.2.2 and 3.2.3 in the Appendix and referring to them in the actual formulation of the PKF closure in Sec.3.2.4 (e.g. adding reference to Sec.3.2.3 in l.555, ...).*

*- this would shorten the main text by about 5 pages (in it's current version)*

*C) Sec.3.3 - LV-CTM experiments: Algorithm 1 provides a good summary, but is rather long. It could also be moved to the appendix because all important steps are described in the manuscript. - this would shorten the main text by about 1 page (in its current version)*

*From my point of view, B) has more priority to be moved into the appendix than A) unless the authors argue differently.*

B) and C) have been moved entirely, and now correspond to Appendix D for B) and included in Appendix F for C) (algo 1).

For A) the description of the part related to the validation of the correlation functions (end of the former section 2.4.1) as well as the explanation of the significant difference between PKF and EnKF by the model error (former section 2.4.2) have been moved to follow the recommendations – (corresponding to Appendix A and B of the new version of the manuscript). However, we have chosen to keep the univariate example with the description of equations 9 and 10 (by deleting the title of sub-section 2.4.1) for the following reasons:

a) first the example introduces the numerical framework which should have been introduced anyway in the multivariate experiments, which would not have decreased the size of the paper, and this concerns the whole paragraphs "The numerical framework.." followed by the paragraph "For this experiment. " of l246-259). Indeed, these paragraphs introduce the details of the numerical resolution and the description of the wind field used (such as Fig.1).

b) then the example also introduces the diagnostics of ensemble estimation for mean, variance and anisotropy (paragraph "To assess..." l232-246)... which again should have been moved to the section on multivariate, so without impact on the size of the paper.

c) finally this example allows the understanding of the dynamics of uncertainties uniquely related to the conservation equation Eq.(9), which induces an evolution of the variance and span length fields (Fig. 2 of the new version), and thus prepares the reader for the interpretation of the multivariate framework (Fig. 6 of the new version) where one understands better what is the effect of chemistry in the propagation of uncertainties (visible by comparing Fig. 6(c) and Fig.2(b)). In particular, a clear reference to Fig. 2 has been added in results of Sec. 3.3.2.

Thus, compared to moving the whole section to an appendix, as you propose, and taking into account the displacement of paragraphs that this move would have induced (introduction of the numerical descriptions as mentioned in point a) and b) to be introduced in the multivariate part), the size of the paper corresponding to the introduction of the univariate part weighs only for approximately one and a half page in the current format (single column one row over two).

*2. Minor technical corrections:*

*a) related to the 1st review, reviewer1, technical correction 10: There are still some introducing sentences of next sections at end of sections. I suggest removing or moving them to beginning of new section (eg: before Sec.2.4, 2.4.1 ,3.1.3, 4.2)*

It has been at the beginning of Sec. 2.4 , 3.1.3, 4.2.

Section 2.4.1 title has been removed after the transformation Sec. 2.4 as discussed in main comment (move of point A) ).

*b) related to the 1st review, reviewer1, technical correction 9: Remove technical figure description "blue dashed lines" at l.314 and similarly l.419*

It has been removed now.

*c) l.43: Suggest reformulating the new sentence starting with "to zero" by moving these two words to the end: "...to set the ensemble estimation of the multivariate correlation to zero, …"*

It has been modified.

*d) l.194: In the current version the title of Sec.2.4 was not changed as described in the 1st review, reply to reviewer1, general correction 2.4*

Sorry, this is done now, thanks!

*e) l.106: Remove "that"*

We removed the word and added a comma before the word "and".

*f) l.295-297: The complex sentence structure makes it difficult to follow. Do you want to say something like: Because the equation for the mean (Eq.(13)) is linear, the error field is given by an equivalent equation. And for these equation, the shorter the correlation length scale the larger is the error magnitude. ??*

We rephrase the paragraph as follows (in the new version where Eq.(13) now stands for Eq.(A1) ):

"This can be understood as follows. Since Eq. (A1) is linear, it is the dynamics of the mean and of the errors in the numerical experiment. But the typical scale of the mean and of the error are different: in this simulation, the spatial scale of the mean state is large, of the order of D, while the spatial scale of the errors is of order $lh \approx D/16$, where $16 \approx 241/15$ ; this implies that the magnitude of the negative phase shift due to the dispersive term is larger for the error than for the mean (see e.g. KdV Eq. (1.19) in Whitham (1999), p.9)." (l694-698)

First of all, we would like to thank the referee for her/his review and for giving us the opportunity to improve our paper.

Now, we organized the answer to the comments as follows. First, we list some changes afford to the manuscript then detail our answers to the questions raised by the referee.

**List of changes for the revision**

*Minor changes*

- Fig. 2 and Fig. A1 'y' label has been modified using \chi in place of A.
- There was a typos in Eq.(D2) that has been corrected replacing \sum_{k=1}^N_e by \sum_{k=1}^{N_e}

*Differences between the two versions of the manuscript*

To facilitate the comparison between the two version of the manuscript, a companion version of the manuscript lists all the modifications where old (new) statements are in red (blue). But the line numbers will refer to the revised version of the manuscript (not to the companion version).

**Answer to the question of the referees**

We copied your commentary in italics below, we reply in normal blue font.

General comments:

*1. I suggested that the manuscript should be shortened because it tests patience of the*

*reader with multiple examples that seem repetitious. That has not been addressed.*

Following the advice of Rev. 1 we moved a large part of the details in appendices, that shorten the new version of the manuscript.

*2. In my opinion the authors don't present "an application to a simplified chemical*

*transport model" but present a concept of using PKF to propagating error covariances.*

*There is big difference between application in a CTM model and illustrating a concept.*

*Reference to CAMS ensemble that the authors mention in the reply is farfetched. In*

*opinion it is hard to find any commonalities between CAMS ensemble and concept*

*that authors present.*

In the new version of the manuscript we clarified the introduction of CAMS and the assimilation in MOCAGE :

We precise that CAMS is a multi-model ensemble ("..which daily forecast a *multi-model* ensemble of 11 members.." l 25), where each model has its own assimilation system ("Note that each member of the ensemble relies on its own data assimilation system for providing its surface analysis, while all models process the same set of surface observations, and all model forecasts are based on the same meteorological forcings from ECMWF high resolution weather forecasts." l26-29) and we precise that CAMS ensemble is not used within an EnKF ("In particular, members of the CAMS multi-model ensemble are not used within an EnKF to provide its own assimilation system." l29-30), all this to avoid any kind of confusion with the method and the framework we consider. Note also that we added a reference to the scientific description of CAMS (see confluence website reference in footnote 1, p2).

CAMS has been introduce to precise the state of the art, with a focus on the CTM MOCAGE and it 3DVar assimilation system that relies on simplifications we would like to avoid by using a PKF. This is now made more clear in the manuscript:

First we precise that the specification of the background error variance as a percentage of the first guess is very different from the forecast error variance that occurs in EnKF where it is deduced from the ensemble ("Note also that simplifications are often introduced to represent a flow dependency of the background term e.g. in several studies using MOCAGE, the 3DVar background error standard deviations are specified as a percentage of the first guess field (Amraoui et al., 2020; Aabaribaoune et al., 2021; Peiro et al., 2018) – which is very different from the forecast error variance in an EnKF that results from the ensemble estimation and the dynamics of the uncertainty." l39-43).

Then, we precise what would be the interest of using a PKF for the MOCAGE assimilation ("Compared to specifying the background variance as a percentage of the first guess, as mentioned above for the MOCAGE assimilation, the PKF could provide a flow dependence more consistent with the KF theoretical framework, but without the numerical cost of using an ensemble as with an EnKF." l82-85).

Since the MOCAGE assimilation and forecast system, as it is used in CAMS, is a 3DVar with background error variance specified as a percent of the guess and a forecast base on the deterministic high-resolution forecast of the ECMWF, we chose a testbed that reproduces a part of this configuration to tackle the multivariate framework: two chemical species in non-linear interaction, transported by a given heterogeneous wind over a 1D domain, with no uncertainty considered on the wind. Because of this it is a simplified CTM that share numerous important property of MOCAGE. This is now clearly stated in the manuscript with a clear reference to the MOCAGE framework: "To explore a multivariate formulation of the PKF, a simplified chemical transport model is introduced that mimics the MOCAGE framework. This simplified CTM contains the essential features of what can be found in a more realistic CTM, that is advection, multiple chemical species and non-linearities." (l298-300).

*3. Reply to my 3rd inquiry is long but, in my opinion, entirely misses the point. Whether the model has interactive meteorology and chemistry has no relevance to the data assimilation approach discussed in the manuscript. In a common application the spread of an ensemble of chemical model realizations may come from varying meteorological states, be it wind in the simplest case for off-line model (u,v) plus state variables (T,q,...) in an online model. Varying meteorology will contribute to spread of the chemical ensemble because it will affect concentrations of the species. As noted in my review, I don't believe that the approach presented will lead to an efficient data assimilation. To be convinced I would welcome an application that the authors proposed using MOCAGE. Otherwise, as I pointed in my review a theoretical basis for*

*a concept may exist but the concept itself will remain just a curiosity without any prospect for a real-world application.*

As detailed in the answer of point 2), the 3DVar assimilation of MOCAGE (as used in the CAMS operational ensemble where MOCAGE is one of the state of the art model), do not take into account the variability due to the meteorology. We want to improve the assimilation of MOCAGE using a PKF, while keeping a low numerical cost for the improvement. However this contribution focuses on the multivariate issue, and the uncertainty due to the chemistry as it is now clearly stated in the abstract (l 4-5), in the introduction (l92-93) and in the conclusion (l620-622) (see our detailed answer to the Editor decision point 2.1) which is about this focus).

Note that the specification of the background variance in the 3DVar of MOCAGE is a way to take into account a part of the uncertainty due to the meteorology (since it is flow dependent through the first guess), but this is far from the uncertainty due to the meteorology as it would follows from the considering the Kalman framework that would be based on en ensemble of weather forecast. After considering the multivariate issue, we are agree that introducing the uncertainty due to the meteorology is an important issue. However, this point can not be addressed without any clue on how to consider the multivariate statistics e.g. the cross-correlation of between the uncertainty on the wind and on the chemistry. The present work contributes to gives some ideas about how to do that.

In particular, in the conclusion, we indicate how we think interesting to take into account the spread due to the meteorology in a way consistent with the PKF approach:
"In addition, since we have focused on the uncertainty due to chemistry, it would be interesting to address the part of the uncertainty due to meteorology. For a CTM like MOCAGE, this could be done by considering an ensemble of weather forecasts with each member used as a forcing for a single CTM forecast. However, this solution would lead to multiple CTM forecasts, which would be expensive. Therefore, from the perspective of using a PKF (applied to a CTM), a less expensive solution would be to consider a single PKF forecast where the wind is uncertain (stochastic advection wind), with the wind uncertainty characterized by the variance and anisotropy tensor estimated from the weather forecast ensemble. The challenge will be to find an appropriate closure for the unknown terms in the dynamics, including the cross-correlation between the wind error and chemical species, with the help of this contribution on multivariate statistics."(l 654-661)

---

## Author Response (AR3)

**Answer to referee comments for the final version of the manuscript**

Again we would like to thanks the referees for their fruitful comments that have improve the quality of the manuscript.

**Answer to Annika Vogel (referee #1)**

Thank you very much for your final review of the manuscript.

*- L.33 & l.606: "cross correlation" -> "cross-correlation"*
This has been corrected.
*- L.317: The spelling of "auto covariance" is inconsistent with the spelling of "autocorrelation" which was introduced in l.33.*
This has beend corrected.
*- L.292 structure of sentence: remove 1st "of". I.e., "The numerical experiment studies of the time propagation…"*
This has beend corrected
*- Ll.272-275: Generally ok, but maybe you could formulate more clearly that the PKF is performing better here? I.e., This difference is due to errors in the enKF, rather than errors in the PKF.*
The sentence you proposed has been included in the final version of the manuscript (see L673).
*- Maybe you want to add a referent to the Algorithm for the multivariate PKF (Alg.1) in the conclusions? This would increase the visibility of the Alg. although being in the Appendix.*
This has been introduced by rewording L646-648 (in the final manuscript version) of the conclusion as follows "Moreover, a multivariate formulation of the PKF analysis step has been introduced, given by Algorithm 1, and several assimilation cycles.."

**Answer to Anonymous Referee #3**

*The authors responded to reviewers' comments. They shortened the manuscript and clarified limitations of the assimilation approach. It appears that this approach would be most suitable to a box chemical model rather than to real-world air quality prediction. I am supporting publication of the manuscript in the current form.*

Thank you very much for your final review of the manuscript. We do not understand your comment concerning the "box model" terminology that refers to 0D domain  for us ( i.e. the dynamics of chemical species in a parcel of fluid) while our contribution concerns a 1D domain with a transport plus a set of chemical reactions, which is a step forward from box models, and intermediate to operational models of air quality in the real world. We agree that assuming the dynamics of the anisotropy independent of the chemistry can make think to a box model where grid-points would be independent from one to the other ; but this is not strictly the case here since we considered non-trivial spatial correlation that implies a spatial dependency of the forecast and of the analysis error along the grid, where the forecast error anisotropy changes because of the heterogeneity of the wind that has been considered. Note however that this simplification of the anisotropy dynamics has been supported by the 1D simulations but it has to be re-evaluated for real systems, as it is mentioned in the conclusion of the manuscript (see l653-654 "To go further, it will be interesting to see if the advection terms remain dominant under different conditions like weaker wind or accelerated chemistry from an ensemble of forecasts of operational CTMs".)